# Bidirectional histone monoaminylation dynamics regulate neural rhythmicity

Qingfei Zheng[1,16,17], Benjamin H. Weekley[2,17], David A. Vinson[2,17], Shuai Zhao[3,17], Ryan M. Bastle[2,17], Robert E. Thompson[4], Stephanie Stransky[5], Aarthi Ramakrishnan[2], Ashley M. Cunningham[2], Sohini Dutta[2], Jennifer C. Chan[2], Giuseppina Di Salvo[2,6], Min Chen[2], Nan Zhang[1], Jinghua Wu[1,16], Sasha L. Fulton[2], Lingchun Kong[2], Haifeng Wang[3], Baichao Zhang[3], Lauren Vostal[7,8], Akhil Upad[7,8], Lauren Dierdorff[2], Li Shen[2], Henrik Molina[9], Simone Sidoli[5], Tom W. Muir[4], Haitao Li[3,10,11]✉, Yael David[7,8,12,13]✉ & Ian Maze[2,14,15]✉

Histone H3 monoaminylations at Gln5 represent an important family of epigenetic marks in brain that have critical roles in permissive gene expression[1–3]. We previously demonstrated that serotonylation[4–10] and dopaminylation[9,11–13] of Gln5 of histone H3 (H3Q5ser and H3Q5dop, respectively) are catalysed by transglutaminase 2 (TG2), and alter both local and global chromatin states. Here we found that TG2 additionally functions as an eraser and exchanger of H3 monoaminylations, including H3Q5 histaminylation (H3Q5his), which displays diurnally rhythmic expression in brain and contributes to circadian gene expression and behaviour. We found that H3Q5his, in contrast to H3Q5ser, inhibits the binding of WDR5, a core member of histone H3 Lys4 (H3K4) methyltransferase complexes, thereby antagonizing methyltransferase activities on H3K4. Taken together, these data elucidate a mechanism through which a single chromatin regulatory enzyme has the ability to sense chemical microenvironments to affect the epigenetic states of cells, the dynamics of which have critical roles in the regulation of neural rhythmicity.

Post-translational modifications (PTMs) of histones have emerged as key regulatory mechanisms contributing to diverse DNA-templated processes[14]. Well-studied PTMs, such as methylation, acetylation and ubiquitination, are dynamically regulated by site-specific writer and eraser enzymes, and can be recognized by reader proteins that facilitate cellular responses[15,16]. Furthermore, numerous small-molecule metabolites can directly react with substrate proteins to form site-specific adducts, or can be indirectly added to amino acid side chains through enzymatic processes[17–19]. Together, these PTMs can impact the three-dimensional architecture of chromatin and alter transcriptional landscapes to mediate cell-fate and plasticity[14–19].

We recently reported on the discovery of a new class of histone PTM, whereby monoamine neurotransmitters, such as serotonin and dopamine, can be transamidated to glutamine residues (termed serotonylation and dopaminylation, respectively)[4–13,20]. We determined that histone H3Q5 is a primary site of modification and demonstrated that H3 monoaminylations have important roles in neural transcriptional programming. We found that H3Q5ser acts as a permissive mark, enhancing the recruitment of the transcription factor complex TFIID

to the active mark H3K4 tri-methylation (H3K4me3), and attenuating demethylation of H3K4 through inhibition of KDM5 and LSD1 demethylases[4,5]. We also characterized dopaminylation at this same site (H3Q5dop) in brain, and found that the accumulation of neural H3Q5dop during abstinence from drug abuse promotes persistent transcriptional programs that precipitate relapse vulnerability[11–13]. We identified that these monoaminylations are catalysed by TG2 (encoded by *TGM2*), a $Ca^{2+}$-dependent enzyme that exhibits multiple functions in cells[21,22]. However, the enzymatic regulatory mechanisms through which H3 monoaminylation adducts are removed, or exchanged, to allow for neural transcriptional plasticity remained poorly understood.

## TG2 is an H3 monoaminylation writer and eraser in cells

To investigate the regulatory mechanism of H3 monoaminylation dynamics in cells, we used a 5-propargylated tryptamine (5-PT) chemical probe to track histone serotonylation reactions through Cu(I)-catalysed azide-alkyne cycloaddition (CuAAC)[4,23]. We verified that TG2 is the primary writer of H3 serotonylation in cells by generating

[1]Department of Radiation Oncology, College of Medicine and Center for Cancer Metabolism, James Comprehensive Cancer Center, The Ohio State University, Columbus, OH, USA. [2]Nash Family Department of Neuroscience, Friedman Brain Institute, Icahn School of Medicine at Mount Sinai, New York, NY, USA. [3]State Key Laboratory of Molecular Oncology, MOE Key Laboratory of Protein Sciences, Beijing Frontier Research Center for Biological Structure, School of Basic Medical Sciences, Tsinghua University, Beijing, China. [4]Department of Chemistry, Princeton University, Princeton, NJ, USA. [5]Department of Biochemistry, Albert Einstein College of Medicine, New York, NY, USA. [6]Department of Psychiatry and Neuropsychology, School for Mental Health and Neuroscience (MHeNs), Maastricht University, Maastricht, The Netherlands. [7]Chemical Biology Program, Memorial Sloan Kettering Cancer Center, New York, NY, USA. [8]Tri-Institutional PhD Program in Chemical Biology, New York, NY, USA. [9]The Rockefeller University Proteomics Resource Center, The Rockefeller University, New York, NY, USA. [10]SXMU-TM Collaborative Innovation Center for Frontier Medicine, Shanxi Medical University, Taiyuan, China. [11]Tsinghua-Peking Center for Life Sciences, Beijing, China. [12]Department of Physiology, Biophysics and Systems Biology, Weill Cornell Medicine, New York, NY, USA. [13]Department of Pharmacology, Weill Cornell Medicine, New York, NY, USA. [14]Department of Pharmacological Sciences, Icahn School of Medicine at Mount Sinai, New York, NY, USA. [15]Howard Hughes Medical Institute, Icahn School of Medicine at Mount Sinai, New York, NY, USA. [16]Present address: Department of Medicinal Chemistry and Molecular Pharmacology, College of Pharmacy, Purdue University, West Lafayette, IN, USA. [17]These authors contributed equally: Qingfei Zheng, Benjamin H. Weekley, David A. Vinson, Shuai Zhao, Ryan M. Bastle. ✉e-mail: lht@tsinghua.edu.cn; davidshy@mskcc.org; ian.maze@mssm.edu

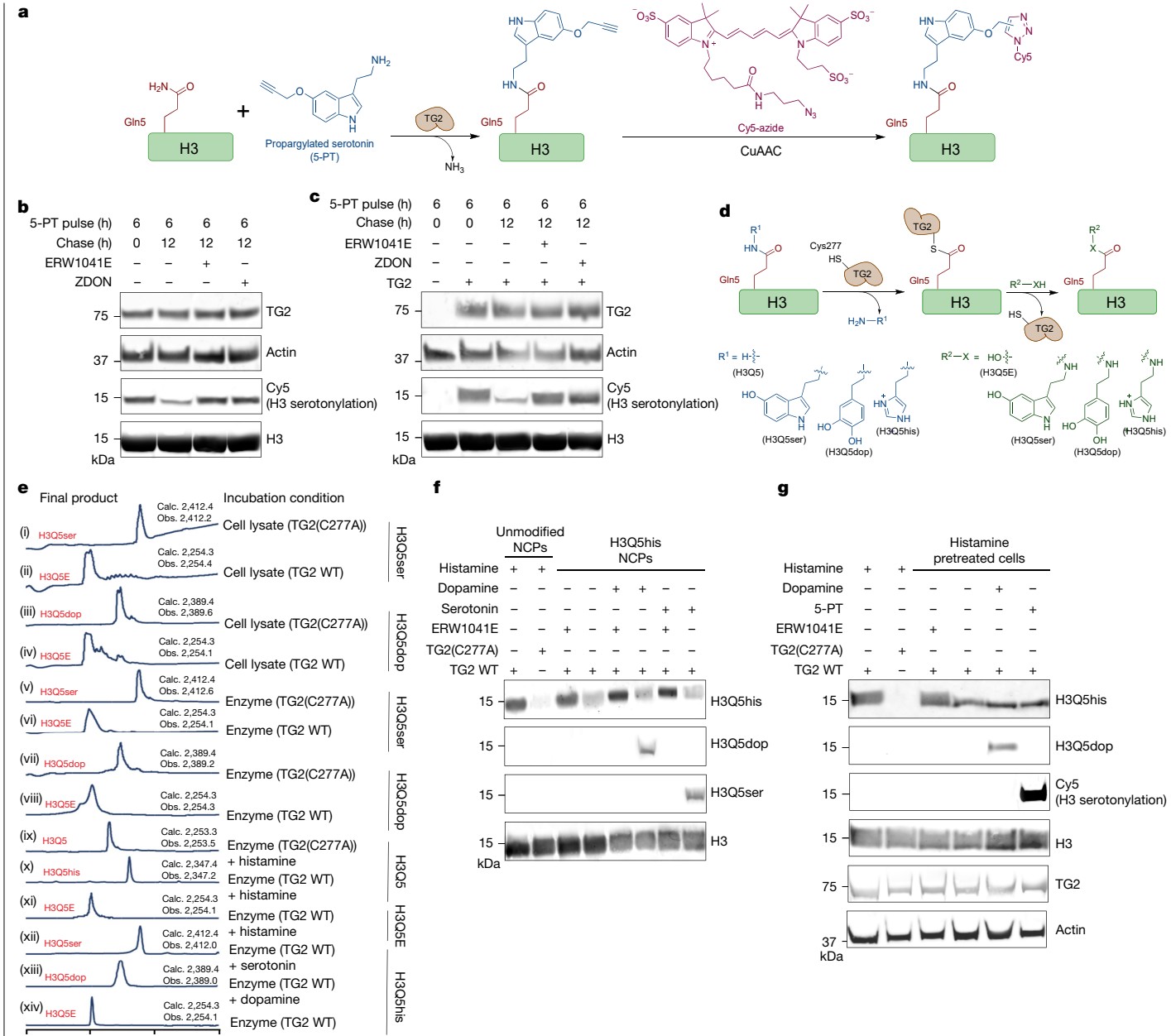

**Fig. 1 | TG2 is a writer, eraser and exchanger of H3 monoaminylations.**
**a**, TG2-mediated covalent modification of H3Q5 by 5-PT, and its visualization by CuAAC-mediated Cy5 conjugation. **b**, 5-PT pulse–chase in HeLa cells treated with or without TG2 inhibitors (ERW1041E or ZDON) after removal of 5-PT. Western blotting was performed for Cy5 (that is, H3 serotonylation) and TG2. **c**, 5-PT pulse–chase experiment in HEK293T cells transfected with WT TG2 and treated with or without TG2 inhibitors after removal of 5-PT. Western blotting was performed for Cy5 and TG2. **d**, Suggested mechanism for TG2-mediated H3 monoaminylation writing, erasing and exchange. **e**, LC–MS analysis of modified H3 peptides (as well as a deamidated H3Q5E peptide standard) (xi) after incubation with cellular lysates expressing WT TG2 versus TG2(C277A) (i–iv) or recombinant WT TG2 versus TG2(C277A) (v–x). H3Q5his peptide was incubated with WT TG2 in the presence or absence of replacement monoamine donors (xii (serotonin) versus xiii (dopamine)), demonstrating WT TG2-mediated deamidation of H3 monoaminylations in the absence of replacement donors

(xiv), and exchange of H3 monoaminylations in the presence of replacement donors. Calculated (calc.) versus observed (obs.) masses are provided. High-performance LC (HPLC) UV traces, $\lambda = 214$ nm. **f**, WT TG2, but not TG2(C277A), transamidated histamine to H3Q5 on NCPs. NCPs premodified by histamine at H3Q5 could be deamidated by WT TG2, which was inhibited by treatment with ERW1041E. WT TG2 exchanged H3Q5his on NCPs in the presence of replacement donors, resulting in the establishment of H3Q5ser or H3Q5dop. NCP, nucleosome core particle. **g**, WT TG2, but not TG2(C277A), transamidated histamine to H3Q5 in HEK293T cells. H3Q5his-premodified histones could be deamidated by WT TG2 in cellulo, which was inhibited by treatment with ERW1041E. WT TG2 exchanged H3Q5his in cellulo in the presence of replacement donors, such as 5-PT (Cy5) or dopamine. H3 and actin were used as loading controls for western blotting. All of the experiments were repeated three times. Uncropped blots are shown in Supplementary Fig. 1.

a CRISPR-mediated *TGM2*$^{-/-}$ HeLa cell line (Extended Data Fig. 1a) and treating them with 5-PT. Next, histones were isolated, clicked with CuAAC-mediated cyanine 5 (Cy5) and imaged for fluorescence (Fig. 1a). Indeed, *TGM2*$^{-/-}$ cells could not catalyse H3 transglutamination; however, H3 serotonylation in these cells was fully restored by overexpression of wild-type (WT)[4], but not catalytically inactive (TG2(C277A)) TG2 (Extended Data Fig. 1b). We next performed a pulse–chase experiment to determine the stability of H3 serotonylation in WT HeLa cells.

Cells were first treated with 5-PT, washed and then incubated in 5-PT-free medium. Histones were then isolated, clicked with Cy5 and imaged for fluorescence. Our results demonstrated that H3 serotonylation was induced in the presence of 5-PT and decreased after 5-PT removal, but not in cells that were treated with TG2 inhibitors (ERW1041E[24] or ZDON[25]) during the chase (Fig. 1b and Extended Data Fig. 1c). We confirmed these results in HEK293T cells, which do not endogenously express TG2[4], by overexpressing WT TG2 and performing a similar pulse–chase experiment to that described above (Fig. 1c).

## TG2 is an H3 monoaminylation exchanger

One potential mechanism of serotonin erasing by TG2 involves a nucleophilic attack of Cys277 on the γ-carboxamide of H3Q5, forming a thioester intermediate (Fig. 1d). A second nucleophilic attack by another monoamine, or water molecule, would then result in the formation of a new monoaminylation adduct, or monoamine removal, respectively. To test this, we synthesized H3 N-terminal tail peptides that represent substrates and product standards for H3-unmodified, H3Q5ser, H3Q5dop and H3Q5E (the predicted deamidated product) (Extended Data Fig. 2a–d). H3Q5ser or H3Q5dop peptides were incubated with lysates from HEK293T cells expressing WT TG2 or TG2(C277A); liquid chromatography–mass spectrometry (LC–MS) analysis of the reactions revealed that peptides incubated with lysates containing WT TG2, but not TG2(C277A), underwent stoichiometric removal/deamidation of the adduct (Fig. 1e (xi, i–iv)). Using purified recombinant WT TG2 or TG2(C277A) (Extended Data Fig. 2f) with H3Q5ser or H3Q5dop peptides provided consistent results (Fig. 1e (v–viii)). Co-immunoprecipitation (co-IP) experiments on recombinant H3 confirmed this enzymatic mechanism by capturing the putative TG2–H3 thioester complex with WT TG2 but not TG2(C277A) (Extended Data Fig. 2g).

In addition to serotonin and dopamine, another monoamine that is involved in diverse physiological processes, ranging from local immune signalling to functions of the gut and neuromodulation, is histamine[26]. Given its importance, we next examined whether histamine also serves as a metabolic donor for histone H3Q5 monoaminylation (that is, H3Q5his). We incubated WT H3 peptides with recombinant TG2 (WT or C277A) in the presence of histamine, followed by LC–MS (or LC–MS/MS) analyses (Fig. 1e (ix–x) and Extended Data Fig. 3a–d). We found that histamine can be stoichiometrically added to H3 peptides at Gln5 (Extended Data Fig. 3c), consistent with previous reports that histamine is a preferred donor for TG2-mediated transamidation of other substrates[27]. Synthetic H3Q5his peptides (Extended Data Fig. 2e) were found to be converted to H3Q5E by WT TG2 in the absence of free monoamines (Fig. 1e (xiv)), and to H3Q5ser (Fig. 1e (xii)) or H3Q5dop (Fig. 1e (xiii)) in the presence of the corresponding monoamine donor. Importantly, we identified endogenous H3Q5his by LC–MS/MS in histamine-treated HEK293T cells expressing WT TG2, but not TG2(C277A) (Extended Data Fig. 3e).

## TG2 regulates nucleosomal H3 monoaminylations

To investigate the dynamics of H3Q5his and its regulation by TG2 in a physiologically relevant context, we performed in vitro competition assays using reconstituted nucleosome core particles (NCPs). We examined NCPs treated with TG2 and histamine, as well as after monoamine exchange, and found that TG2 efficiently catalyses H3Q5his on NCPs, which can be enzymatically exchanged in the presence of another monoamine donor, yet not when a TG2-specific inhibitor was present (Fig. 1f).

To track H3Q5his in cells, we generated polyclonal antibodies that selectively recognize H3Q5his (Extended Data Figs. 4a–e (H3Q5his total) and 4f–h (H3K4me3Q5his)). Treating HEK293T cells expressing WT TG2 or TG2(C277A) with histamine revealed the accumulation of H3Q5his in WT TG2-overexpressing but not TG2(C277A)-overexpressing cells

(Fig. 1g). We then performed a pulse–chase experiment where histamine was added to the medium to allow for H3Q5his establishment, followed by a chase with media containing 5-PT or dopamine. In the presence of WT TG2, but not TG2-C277A, H3Q5his was efficiently exchanged to the corresponding monoamine present in the medium (Fig. 1g).

## H3Q5his antagonizes WDR5 binding

H3Q5ser influences the deposition, maintenance and functional readouts of another key epigenetic modification, H3K4me3[4,5,28]. We therefore next assessed the biochemical impact of exchanging H3Q5ser for H3Q5his on this mark. We performed isothermal titration calorimetry (ITC) to compare the binding affinities of six well-established H3(K4me3) reader domains to H3(K4me3) versus H3(K4me3)Q5ser versus H3(K4me3)Q5his peptides. Among the reader domains tested, only WDR5-WD40—a core member of H3K4 methyltransferase complexes[29] (MLL1–4 and SETD1A/B; Fig. 2a)—was found to display differential binding to H3. H3Q5ser displayed slightly favoured interactions ($K_D = 14.01\ \mu M$) versus unmodified H3 ($K_D = 20.20\ \mu M$), and H3Q5his reduced H3 binding by around fivefold ($K_D = 108.81\ \mu M$) (Extended Data Fig. 5a–f and Supplementary Table 1). Previous research indicated that H3Q5ser can subtly potentiate the binding of WDR5[4,5,28] to promote H3K4 methylation. As such, we examined whether H3Q5his alters the catalytic activities of WDR5-containing MLL–SETD1 complexes, based on our previous observation that H3Q5ser does not greatly affect H3K4 methyltransferase activities of MLL1 in vitro[5] (validated in Extended Data Fig. 6a). We performed matrix-assisted laser desorption/ionization time-of-flight (MALDI-TOF) MS and found that H3Q5his robustly attenuated MLL1 complex processivity to add H3K4 methylation (Fig. 2b). We confirmed these results using LC–MS/MS to monitor the establishment of H3K4 methylation by all enzymatically active H3K4 methyltransferase complexes—MLL1–4 and SETD1A/B. Using recombinant complexes incubated with H3 peptides, either unmodified or Q5his, we found that H3Q5his significantly decreased the activities of all of the complexes examined (Fig. 2c and Extended Data Fig. 6b–g).

The MLL–SETD1 complexes bind to the H3 tail through interactions with WDR5, which can facilitate H3K4 methyltransferase activities[30]. We therefore performed peptide immunoprecipitations (IPs)—using H3 unmodified versus Q5his versus Q5ser—against full-length recombinant WDR5 (the only member of core H3K4 methyltransferase complexes to bind to H3 in monomeric form; Extended Data Fig. 6h) or the reconstituted MLL1 complex. While H3Q5ser significantly potentiated interactions of WDR5 and MLL1 with H3, H3Q5his displayed significantly reduced binding (Fig. 2d and Extended Data Fig. 6i). Similarly, H3K4me3Q5his was found to attenuate binding of WDR5 to H3 versus H3K4me3Q5ser or H3K4me3 alone (Fig. 2e).

Given that H3Q5his, in contrast to H3Q5ser, is monoprotic in a dynamic manner at physiological pH, we examined whether its charge may contribute to the inhibition of binding versus H3Q5ser. We used X-ray crystallography to determine the structure of the WDR5-WD40–H3Q5his complex (Extended Data Table 1). While the electron density of H3Q5 was traceable, the density around the histamine moiety was not fully resolved (Fig. 2f,g). However, we were able to predict H3Q5his interactions based on the orientation of the H3Q5 residue. While the H3R2 sidechain was inserted into the binding pocket of WDR5-WD40 (with H3K4's sidechain at the surface of WDR5-WD40), as shown previously[31], H3Q5his displayed surface binding adjacent to the binding pocket. We observed that H3Q5his exists within proximity to the positively charged WDR5-WD40 Lys259 residue, suggesting a potential electrostatic repulsion that is predicted to result in binding inhibition. The flexible orientation of H3Q5his within this complex was found to differ in comparison to that of H3Q5ser, where the binding pocket of H3Q5ser is neutral or even slightly negatively charged[28]. After alignments to compare WDR5-WD40–H3Q5unmod[32] versus WDR5-WD40–H3Q5ser[28] versus

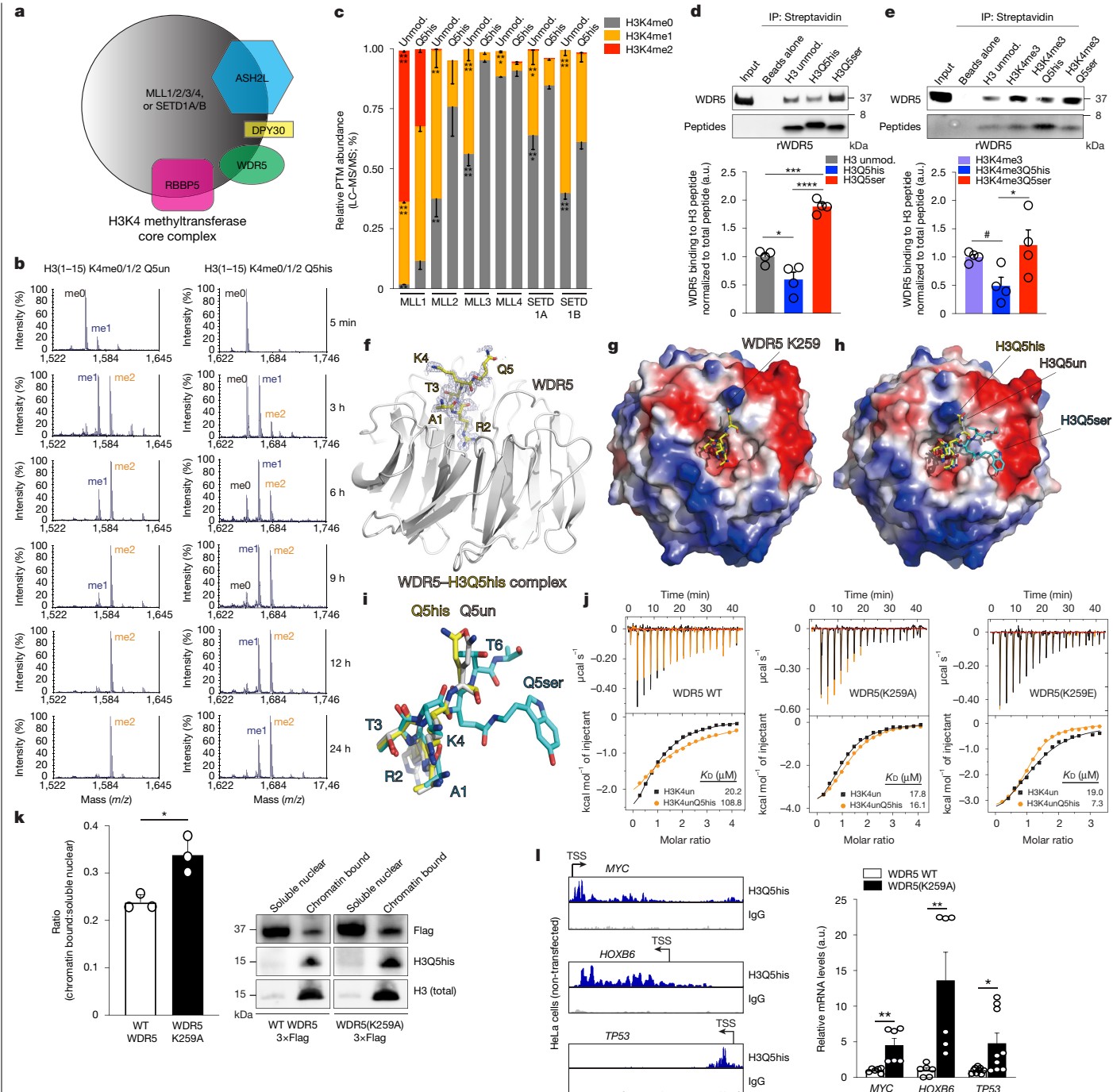

**Fig. 2 | H3Q5his antagonizes H3K4 methyltransferase activities and WDR5 binding. a**, H3K4 methyltransferase complexes. **b**, MALDI-TOF analysis of MLL1-mediated H3K4 methylation on peptides. $n = 1$ per peptide per timepoint. Q5un, unmodified Gln5. **c**, LC–MS/MS analysis of MLL–SETD1-mediated H3K4 methylation on peptides. $n = 3$ per peptide per complex. **$P < 0.01$, ***$P < 0.001$, ****$P < 0.0001$. Statistical details are provided in Extended Data Fig. 6b–g. Unmod, unmodified. **d**, Peptide IPs against WDR5. Streptavidin was used to visualize peptides after IP (used for normalization). $n = 4$ per peptide. Statistical analysis was performed using one-way analysis of variance (ANOVA) ($P < 0.0001$, $F_{2,9} = 57.34$) with Tukey's multiple-comparison test; significant comparisons are noted (H3un versus H3Q5his (*$P = 0.0243$); H3un versus H3Q5ser (***$P = 0.0001$); H3Q5his versus H3Q5ser (****$P < 0.0001$)). **e**, Peptide IPs against WDR5. Streptavidin was used to visualize peptides after IP (used for normalization). $n = 4$ per peptide. Statistical analysis was performed using one-way ANOVA ($P = 0.0492$, $F_{2,9} = 4.287$) with Tukey's multiple-comparison test; significant comparisons are noted (H3K4me3Q5his versus H3K4me3Q5ser (*$P = 0.0456$)); #$P = 0.0177$ indicates a significant difference as determined using an a posteriori

unpaired Student's $t$-test ($t_6 = 3.237$). **f,g**, Electron density map (**f**) and electrostatic potential surface view (**g**) of the WDR5-WD40–H3Q5his complex. The peptide is shown as sticks. **h,i**, Electrostatic potential surface view (**h**) and amino acid orientation (**i**) of alignments between WDR5-WD40–H3Q5un versus WDR5-WD40–H3Q5ser versus WDR5-WD40–H3Q5his. **j**, ITC assessments of WDR5-WD40 (left), WDR5-WD40(K259A) (middle) or WDR5-WD40(K259E) (right) peptide binding. **k**, 3×Flag-tagged WT versus K259A WDR5 chromatin binding in HeLa cells. $n = 3$ biological replicates per construct. Statistical analysis was performed using unpaired Student's $t$-tests ($t_4 = 4.163$); *$P = 0.0141$. **l**, Quantitative PCR (qPCR) analysis (right) of H3Q5his-enriched target genes in HeLa cells (IGV browser tracks of H3Q5his versus IgG enrichment at H3Q5his targets are shown on the left) after transfection with WDR5 WT versus WDR5(K259A). Statistical analysis was performed using unpaired Student's $t$-tests for each gene (*MYC*: **$P = 0.0052$, $t_{10} = 3.559$, $n = 6$ per group; *HOXB6*: **$P = 0.0098$, $t_{10} = 3.182$, $n = 6$ per group; *TP53*: *$P = 0.0186$, $t_{16} = 2.620$, $n = 9$ biological replicates per group). Uncropped blots are shown in Supplementary Fig. 1. Data are mean ± s.e.m. a.u., arbitrary units, all normalized to the respective controls.

WDR5-WD40–H3Q5his complexes, we found that, while the H3A1–H3K4 positions remained consistent across structures, H3Q5his adopts a different binding orientation versus H3Q5ser, indicating that the histamine moiety promotes flexibility of H3Q5, which might allow H3Q5his to interact with WDR5 Lys259 (Fig. 2h,i). To test this, we performed ITC assessments using WDR5-WD40 mutants. We found that, while H3Q5his inhibits WDR5-WD40 binding to H3, WDR5-WD40(K259A) (which eliminates the positive charge of Lys259) rescued the interaction (Fig. 2j and Supplementary Table 1). Reciprocally, the negatively charged WDR5-WD40(K259E) mutant displayed a modest increase in binding to H3Q5his; however, owing to the larger side chain of WDR5(K259E) (versus WDR5-K259A), we cannot fully exclude the possibility that additional favourable intramolecular contact(s) promote binding interactions that are unrelated to the presence of histamine.

To examine the cellular impact of H3Q5his-mediated antagonism of WDR5 binding, we transfected HeLa cells (which endogenously contain H3Q5his when cultured in histamine-containing serum) with 3×Flag-tagged WT WDR5 versus WDR5(K259A) and monitored 3×Flag–WDR5 binding to chromatin (versus enrichment in soluble nuclear fractions) using high-salt extraction[33]. WDR5(K259A) was found to bind more tightly to cellular chromatin containing H3Q5his versus WT WDR5 (Fig. 2k). We next performed CUT&RUN-seq for H3Q5his using our antibody that recognizes H3Q5his in the presence or absence of H3K4me3 (Extended Data Fig. 4f) to identify H3Q5his-enriched target genes (Extended Data Fig. 4j,k). We additionally performed CUT&RUN-seq for H3K4me3Q5his using an antibody that can recognize only the combinatorial modification (Extended Data Fig. 4g,h); our results demonstrated concordant enrichment patterns genome-wide for both the single and combinatorial marks, with more than 85% of peaks for both found within genic regions of the genome. Transcription-level assessments using qPCR were then performed after transfection with WT WDR5 versus WDR5(K259A) to examine the effect of WDR5(K259A) expression on H3Q5his target gene regulation. We confirmed that not only does WDR5(K259A) bind more tightly to chromatin versus WT WDR5, but that such increased enrichment also coincides with elevated expression of H3Q5his-enriched target genes (such as *MYC*, *HOXB6* and *TP53*; Fig. 2l).

## Neural H3Q5 monoaminylations are diurnally rhythmic

To examine the biological significance of H3(K4me3)Q5his and H3(K4me3)Q5ser dynamics in vivo, we next turned our investigations to the posterior hypothalamic tuberomammillary nucleus (TMN). The TMN is the only brain region that contains neurons expressing histidine decarboxylase (HDC), the enzyme that catalyses the formation of histamine from histidine[34]. This region consists largely of histaminergic neurons and is involved in diverse biological functions, ranging from control of arousal to maintenance of sleep–wakes cycles and energy balance[35]. Moreover, afferent projections to the TMN are widespread, including brainstem innervation arising from serotonergic brain structures, which depolarize TMN neurons; notably, other monoaminergic nuclei in brain, such as the locus coeruleus (noradrenergic) and substantia nigra/VTA (dopaminergic), only send limited fibres to TMN[36]. Indeed, the TMN was found to be enriched for H3Q5his versus other non-histaminergic brain nuclei (Extended Data Fig. 4m). Given its prominent role in diurnal behavioural rhythmicity, we next assessed whether the TMN displays rhythmic patterns of gene expression that may require chromatin-based control. We performed RNA-sequencing (RNA-seq) analysis of mouse TMN tissues collected at various timepoints across zeitgeber time (ZT) beginning at time ZT0 (which marks the initiation of their inactive phase) and then every 4 h after that for 24 h, with ZT12 marking the beginning of the mouse's active phase. These data were analysed using JTKcycle[37], which is a nonparametric algorithm designed to detect rhythmic components in genome-wide datasets. This analysis revealed that the TMN displays

rhythmic (Fig. 3a), CLOCK-associated (Fig. 3b) gene expression, with many known circadian genes (such as *Arntl*, *Dbp*, *Per1/2*) identified as being significantly regulated in this manner (Fig. 3c and Supplementary Table 2).

Given that histamine release from the TMN fluctuates across circadian time to control states of arousal[35], we next investigated whether H3Q5his displays alterations in its overall abundance across ZT. We began by performing western blotting for H3Q5his, as well as the combinatorial H3K4me3Q5his mark. We found that both H3Q5his and H3K4me3Q5his displayed significant fluctuations in expression, with their levels highest during the mouse active phase and lowest during the mouse inactive phase (Extended Data Fig. 7a). These observations were not dependent on (1) alterations in TG2 (Extended Data Fig. 7b) or HDC expression across ZT (Extended Data Fig. 8a); or (2) overall levels of monoamines present within the TMN, as measured using enzyme-linked immunosorbent assays (ELISAs; Extended Data Fig. 8b,c). We further confirmed the accuracy of our dissections by comparisons of HDC expression in the TMN versus the suprachiasmatic nucleus (SCN), the latter of which does not express HDC, as well as by western blotting for NeuN to control for cellular distribution within tissue dissections. As ELISA assays cannot provide accurate measurements of intracellular versus extracellular pools of monoamines under study, it remains possible that intranuclear fluctuations of histamine still contribute to the regulation of H3(K4me3)Q5his. Similarly, while HDC levels were not altered across ZT, the gene encoding HDC was found to be significantly rhythmic in the TMN (Fig. 3a and Supplementary Table 2), suggesting that even subtle alterations in *Hdc* expression and, in turn, histamine biosynthesis may contribute to the regulation of the histaminyl marks. Finally, to assess whether other brain regions that receive histaminergic projections also display H3Q5his dynamics, we evaluated H3Q5his expression across ZT in the hypothalamic SCN, which is a critical brain region important for producing circadian rhythms[38]. In contrast to the TMN, SCN H3Q5his levels were not found to significantly fluctuate (Extended Data Fig. 7c).

We next aimed to further dissect the mechanistic roles for these marks in the regulation of circadian gene expression. We performed CUT&RUN-seq for H3K4me3Q5his (which is largely enriched throughout genic loci, similar to that in HeLa cells)—along with H3K4me3Q5ser, H3K4me2, H3K4me3 (using an antibody that recognizes H3K4me3 in both the presence and absence of H3Q5his or H3Q5ser; Extended Data Fig. 4i) and WDR5—in the TMN across ZT. IgG and *Escherichia coli* DNA spike-in controls were included for precise normalizations, and enrichment patterns for the H3 monoaminylation marks were a priori determined to be TG2 dependent, as assessed in *TGM2*[−/−] cells (Fig. 3d, Extended Data Fig. 4l and Extended Data Fig. 8d–i). We observed that H3K4me3Q5his was indeed rhythmic in the TMN (Supplementary Table 3), displaying its highest levels of enrichment during the mouse's active phase (ZT16–ZT20) and its lowest levels of enrichment during the mouse's inactive phase (ZT0–ZT4) (Fig. 3d). A second maxima for the combinatorial mark was observed at ZT8, which denotes a period when the animals are beginning to transition from their inactive to active phase. Considerable overlap was also observed between genes determined to be circadian in the TMN (Fig. 3a) and those displaying rhythmic fluctuations in H3K4me3Q5his (1,041 genes). These overlapping genes were significantly enriched for 'upstream regulator' ontologies related to the transcription factor CLOCK (Fig. 3e), a master mediator of circadian gene expression[39].

We next compared rhythmic patterns of H3K4me3Q5his to H3K4me3Q5ser, which was also shown to be circadian in the TMN (Extended Data Fig. 8g and Supplementary Table 4). We found that both H3K4me3Q5his and H3K4me3Q5ser displayed increased enrichment at the height of the mouse active phase (ZT16), yet only H3K4me3Q5ser was observed to display a marked reduction in its enrichment towards the end of the mouse active phase (ZT20) (Fig. 3f,h and Extended Data Fig. 9a). This loss of H3K4me3Q5ser at ZT20 corresponded to a loss

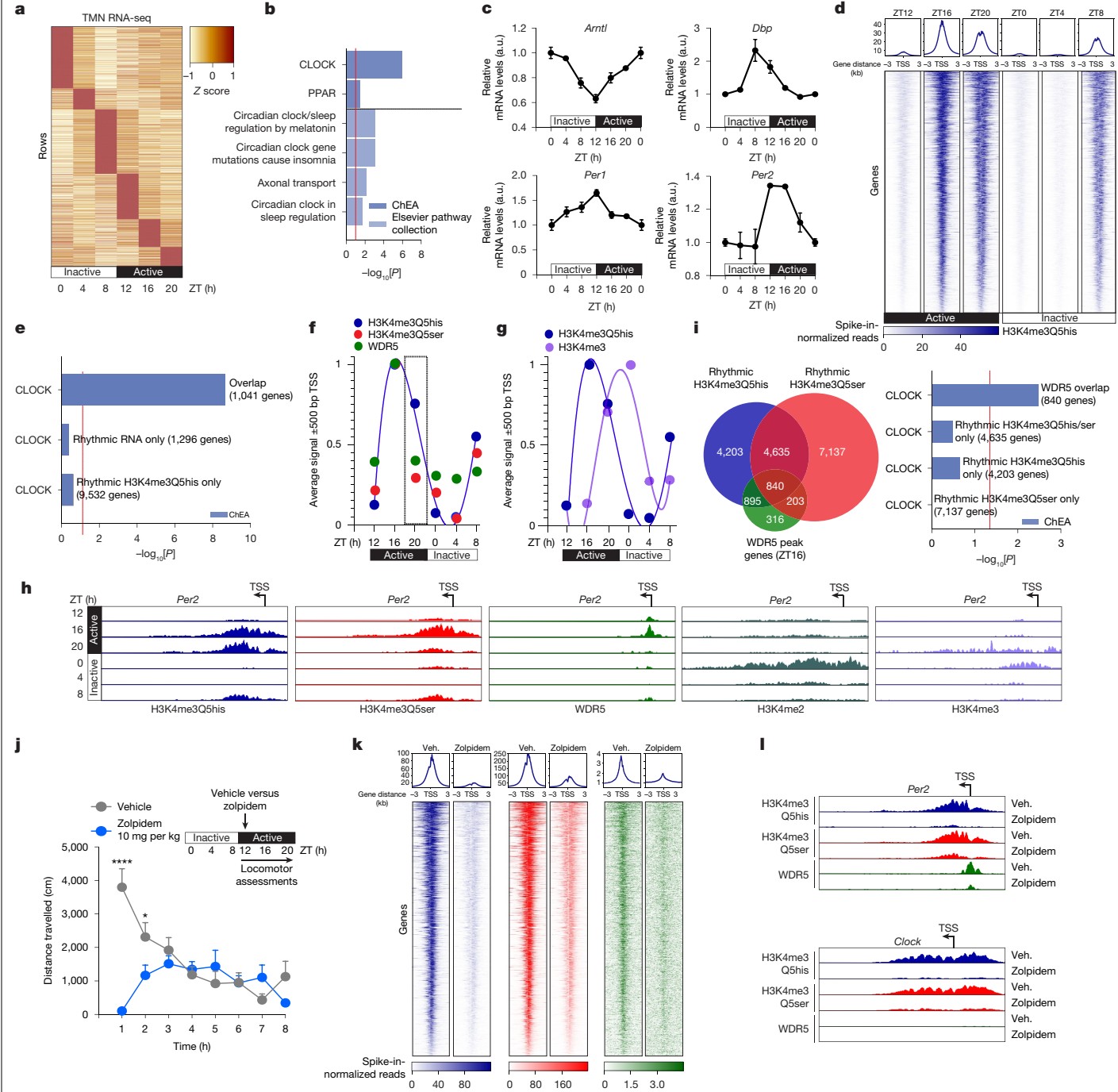

**Fig. 3 | Neural H3Q5 monoaminylations are diurnally rhythmic. a**, RNA-seq analysis of the mouse TMN across ZT (JTKcycle, $P_{adj} < 0.05$). $n = 3$ biological replicates per timepoint. **b**, Ontology analysis ($P < 0.05$) of RNA-seq data from **a**. **c**, Known circadian genes identified in **a**. **d**, CUT&RUN–seq enrichment for H3K4me3Q5his across ZT at genic loci in TMN. $n = 3$ biological replicates per timepoint (JTKcycle, $P_{adj} < 0.05$). **e**, Ontology analysis ($P < 0.05$) comparing enrichment of CLOCK-mediated gene targets for genes displaying rhythmic H3K4me3Q5his only versus circadian genes not displaying rhythmic H3K4me3Q5his and/or gene expression. **f**, The average signal intensity ±500 bp from the transcription start site (TSS) across ZT for H3 monoaminylations and WDR5. **g**, The average signal intensity ±500 bp from the TSS across ZT for H3K4me3Q5his and H3K4me3. **h**, IGV tracks for *Per2* across ZT for H3 monoaminylations versus WDR5 versus H3K4me2/3. **i**, Overlap of rhythmic H3 monoaminylation genes with WDR5-enriched genes at ZT16 (left), and ontology

analysis ($P < 0.05$) comparing enrichment of CLOCK-mediated gene targets for genes displaying rhythmic H3 monoaminylation versus WDR5 enrichment at ZT16 (right). **j**, Locomotor activity in mice treated with zolpidem ($n = 12$ mice) versus vehicle ($n = 13$ mice) during their active phase. Statistical analysis was performed using repeated-measures two-way ANOVA (main effects of interaction, time × zolpidem: $P < 0.0001$, $F_{7,161} = 11.98$) with Tukey's multiple-comparison test; significant post hoc comparisons (vehicle (veh.) versus zolpidem) are noted (1 h: ****$P < 0.0001$; 2 h: *$P = 0.0412$). **k**, CUT&RUN–seq enrichment for H3 monoaminylations and WDR5 at ZT20 after treatments with zolpidem versus vehicle (at ZT12) at genic loci in TMN. $n = 3$ biological replicates per timepoint. **l**, IGV tracks for *Per2* and *Clock* at ZT20 for H3 monoaminylations and WDR5 after treatments with zolpidem versus vehicle (at ZT12) at genic loci in the TMN. Data are mean ± s.e.m.

of enrichment for WDR5, which itself displayed its highest level of chromatin enrichment at ZT16 (Fig. 3f,h and Extended Data Figs. 8f and 9a). WDR5 peaks were primarily enriched at permissive, CLOCK–BMAL1-target genes (for example, *Per1/2/3*, *Dbp* and *Nr1d1/2*; Fig. 3i, Extended Data Fig. 9a–d and Supplementary Table 5), which are induced during the mouse active phase[40]. These were not enriched, however, at other circadian loci (for example, *Clock* and *Arntl*) that are regulated during the mouse inactive phase by downstream targets of CLOCK–BMAL1. We also observed that H3K4 methylation patterns (H3K4me2 and H3K4me3) were out of phase with H3K4me3Q5his, displaying their highest levels of enrichment after entry into the mouse's active phase (ZT0–ZT4) (Fig. 3g,h and Extended Data Figs. 8d,e and 9a).

We next sought to investigate whether perturbing diurnal rhythms results in altered regulation of H3Q5 monoaminylations and WDR5 dynamics in the TMN. We treated mice with the sleep aid zolpidem during the beginning of the mouse's active phase (ZT12)—a perturbation that robustly resulted in an immediate loss of activity (Fig. 3j). We then collected mouse TMN tissues 8 h after treatment for CUT&RUN-seq assessments of H3K4me3Q5his and H3K4me3Q5ser, as well as WDR5. In comparison to the vehicle-treated controls, zolpidem greatly reduced the enrichment of H3K4me3Q5his, H3K4me3Q5ser and WDR5 genome-wide (Fig. 3k), including at CLOCK–BMAL1-target genes (for example, *Per2*, but not *Clock* itself; Fig. 3l), thus phenocopying the molecular regulation of these marks/proteins observed during the mouse's inactive phase.

## H3Q5 monoaminylations contribute to rhythmicity

While the dynamics of H3Q5 monoaminylations across ZT correlated with fluctuations in WDR5 enrichment at CLOCK–BMAL1-target genes (along with related circadian gene expression), it remained unclear whether these marks have causal roles in the regulation of molecular and/or behavioural rhythmicity. We therefore performed adeno-associated viral vector (AAV) transduction in the TMN to introduce a mutant form of histone H3, H3.3(Q5A), which actively incorporates into neuronal chromatin (Fig. 4a). This mutant functions as a dominant negative in brain (where H3.3 is the only H3 variant that can be actively incorporated into neuronal chromatin[41]), reducing the global levels of monoaminylated H3[4,7,10–13]. Two AAV controls were also transduced: an empty vector expressing GFP and a vector expressing WT H3.3, the latter of which does not alter H3Q5his expression versus GFP (Fig. 4b). However, H3.3(Q5A) was found to significantly decrease H3Q5his levels compared with both controls in the TMN (Fig. 4b). Transduction with WT H3.3 or H3.3(Q5A) (versus GFP) was also not observed to alter levels of other H3 modifications (Extended Data Fig. 9e), highlighting the selectivity of our manipulations. While this viral approach is not specific for a given monoaminylation mark, we considered this to be an advantage in our paradigm, as we wished to be able to disrupt all H3 monoaminylation events in the TMN for subsequent downstream assessments. We transduced adult mice intra-TMN with one of the three AAVs and then collected virally infected tissues 3 weeks later across ZT for RNA-seq analysis. Comparing the gene expression profiles between the two viral control groups, we observed negligible differences in expression. As such, we collapsed the data from the two controls before comparisons against H3.3(Q5A). These control data were then analysed using JTKcycle, which revealed significant patterns of circadian gene expression (2,566 genes), including regulation of many well-established circadian genes (such as *Clock*, *Per1* and *Dbp*; Fig. 4c (left) and Supplementary Table 6). However, after transduction with H3.3(Q5A), circadian transcriptional regulation in the TMN was largely disrupted (Fig. 4c (right) and Supplementary Table 7). Genes displaying disrupted circadian gene expression in H3.3(Q5A) mice were found to be significantly enriched for ontologies related to CLOCK-mediated transcription, with further analyses indicating prominent roles in processes associated with circadian entrainment, neurotrophin signalling

and synaptic regulation (Fig. 4d). Given this, we next examined the impact of these perturbations on diurnal behaviour, as assessed on the basis of locomotor activity across ZT. After viral transductions, mice were monitored for locomotor activity beginning 12 h after a shift from light–dark to dark–dark to examine whether disrupting H3Q5 monoaminylations in the TMN alters the circadian entrainment. We found that perturbing H3Q5 monoaminylations in the TMN resulted in shifted diurnal locomotor activity, particularly during transitions from inactive to active states, and vice versa (Fig. 4e).

## Discussion

Here we investigated the dynamics of monoamine installation on histone H3 by TG2, which revealed that these PTMs directly compete. Mechanistically, we determined that this is due to the ability of TG2 to act as writer, eraser and exchanger of monoaminylated moieties on H3. We found that TG2 can use any of the monoamine co-factors examined on either unmodified or monoaminylated glutamines, a process that is dependent on microenvironmental concentrations of provided nucleophiles. Furthermore, when TG2 erases the monoamine adduct in the absence of alternative monoamine donors, it leaves a glutamate at the site of modification, which results in a form of protein-induced mutagenesis that would probably prove deleterious to cellular function. As such, TG2's monoaminylation exchange activity on H3 probably represents a prominent mechanism for maintaining cellular homeostasis during periods of intracellular monoamine fluctuations.

Leveraging this mechanistic insight, we identified a third class of H3 monoaminylation, histaminylation, which is also facilitated by TG2 and occurs on H3Q5. Given previous evidence indicating that histamine levels in the neonatal rodent brain are largely nuclear[42], the functions of which have remained unclear, we were interested in further exploring the mechanistic roles for this new PTM in vivo. We found that H3Q5his expression is enriched at the site of histamine production in brain, and showed that H3(K4me3)Q5his levels and genomic enrichment are dynamic in the TMN as a function of circadian time. These dynamics opposed patterns of H3K4 methylation, which have previously been implicated in circadian gene expression[43]. We observed that these dynamics can be disrupted by pharmacological manipulations of sleep–wake cycles, and provided evidence that intra-TMN viral manipulations that reduce H3Q5 monoaminylation levels substantially disrupt normal patterns of circadian gene expression, as well as locomotor activity during sleep–wake transitions. Finally, based on a series of biochemical and structural assessments, we found that H3(K4me3)Q5his, in comparison to H3(K4me3)Q5ser, attenuates WDR5 complex binding to the H3 tail and inhibits the H3K4 methyltransferase activities of MLL1–4 and SETD1A/B complexes, all of which contain WDR5 as a co-factor. We hypothesize that H3Q5his may have two independent effects on MLL (and possibly SETD1) complex activities: (1) the mark alters the recruitment of MLL complexes to chromatin in a WDR5-dependent manner, possibly owing to its flexibility within the WDR5–H3Q5his complex and unfavourable electrostatics; and (2) H3Q5his may additionally inhibit the activity of the catalytic centre of MLL–SETD1 complexes, although this possibility has yet to be tested.

Rhythmic fluctuations in, and divergent patterns of, H3K4me3Q5his and H3K4me3Q5ser (along with associated changes in WDR5 binding) correlate with rhythmic patterns of gene expression that appear to be dependent on the CLOCK–BMAL1 transcription factor complex. Our observations also complement earlier studies that found that the MLL1 complex binds to both CLOCK and BMAL1, with MLL1 being recruited to CLOCK–BMAL1-target genes in a rhythmic manner[44]. Moreover, previous genome-wide ChIP–seq-based analyses in mouse liver across circadian time[45] provided robust evidence indicating that, while co-activator recruitment of chromatin regulators by CLOCK–BMAL1 precedes transcription, there is a clear lag in enrichment patterns for H3K4 methylations relative to RNA Pol II recruitment. While

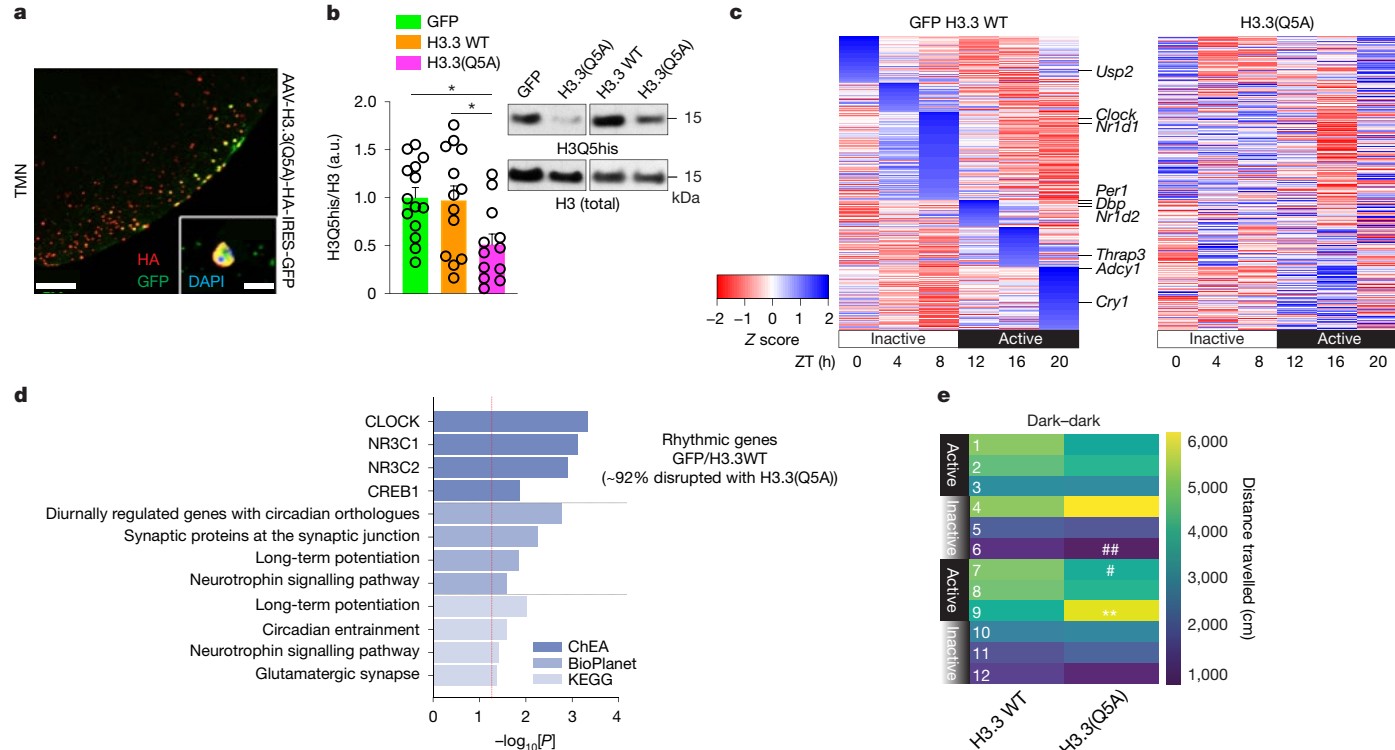

**Fig. 4 | H3Q5 monoaminylations causally contribute to transcriptional and behavioural rhythmicity. a**, Immunohistochemistry and immunofluorescence analysis confirming nuclear expression of H3.3(Q5A)–HA in the TMN of mice expressing AAV-H3.3(Q5A)-HA-IRES-GFP. The experiment was repeated three times. **b**, Western blot validation of H3Q5his downregulation in the TMN after AAV-mediated expression of H3.3(Q5A) ($n = 12$ biological replicates) versus H3.3 WT ($n = 13$ biological replicates) or empty vector controls ($n = 13$ biological replicates). Statistical analysis was performed using one-way ANOVA ($P = 0.0154$, $F_{2,36} = 4.701$) with Tukey's multiple-comparison test; significant comparisons are noted (GFP versus H3.3(Q5A): *$P = 0.0228$; WT H3.3 versus H3.3(Q5A): *$P = 0.0378$). Data are mean ± s.e.m. normalized to total H3 signal. a.u. are normalized to GFP controls. **c**, RNA-seq data from the mouse TMN across ZT for mice that were transduced with GFP ($n = 32$ mice) or WT H3.3 ($n = 39$ mice) (collapsed; JTKcycle, $P_{adj} < 0.05$) versus H3.3(Q5A) ($n = 40$ mice). Notable circadian genes are highlighted. **d**, ChEA, BioPlanet and KEGG ontology analyses ($P < 0.05$) revealed that rhythmic genes disrupted by H3.3(Q5A) are CLOCK targets and enriched for pathways/processes related to circadian entrainment, neurotrophin signalling and synaptic function. **e**, After intra-TMN transduction with H3.3(Q5A) ($n = 11$ mice) versus WT H3.3 ($n = 10$ mice), mice were monitored for locomotor activity beginning 12 h after a shift from light–dark to dark–dark to examine whether disrupting H3Q5his alters normal circadian cycling. Disrupting normal H3Q5his dynamics in the TMN resulted in shifts in diurnal locomotor activity during transitions from inactive to active states and vice versa. The heat map presents locomotor data binned into 4 h intervals for a total of 48 h. Statistical analysis was performed using two-way repeated-measures ANOVA (interaction of time × virus, $P = 0.0040$), with Šidák's multiple-comparison test (**$P = 0.0079$); and a posteriori unpaired Student's $t$-tests ($^{\#}P < 0.05$, $^{\#\#}P < 0.01$). Data are mean ± s.e.m. Uncropped blots are shown in Supplementary Fig. 1.

mechanistic roles for H3K4me2/3 in circadian transcriptional regulation have yet to be fully elucidated, our observations indicate that H3 monoaminylation dynamics have important roles in dictating WDR5 recruitment to CLOCK–BMAL1-target loci to control H3K4 methylation states and circadian gene expression (the model is shown in Extended Data Fig. 9f).

Taken together, these data provide a paradigm that addresses the dynamic regulation of histone monoaminylation events in vivo. However, much still remains unclear regarding mechanistic roles for TG2-mediated H3 monoaminylation exchange in processes associated with adaptive neural plasticity. Among others, there is the possibility of additional roles for H3Q5his and H3Q5ser rhythmicity outside of their regulation of WDR5, as many of these events were found to be unrelated to WDR5 binding. For example, we have previously demonstrated that H3Q5ser functions to maintain adjacent H3K4 methylation levels by inhibiting the enzymatic removal of these marks by KDM5 and LSD1 demethylases[5]. Whether H3Q5his functions in a similar or opposite capacity, or whether these maintenance mechanisms contribute to rhythmic gene expression across circadian time, have yet to be determined. Furthermore, although we know that H3 monoaminylation signal/genomic enrichment is TG2-dependent, both in vitro and in cells, there remains a paucity of tools available for studying

TG2's genomic interactions in brain. As such, future efforts focused on developing such reagents for use in exploring this enzyme's chromatin interactions across circadian time are warranted and promise to provide further mechanistic insights into TG2's molecular control over H3 monoaminylation exchange. Finally, owing to challenges in being able to accurately identify these monoaminylation modifications in vivo using standard analytical approaches in the absence of antibody-based enrichment or bio-orthogonal labelling[4,9], the relative stoichiometries of these H3 modifications in cells/brain remain unclear. Future efforts focused on developing new methodologies to accurately quantify all H3 monoaminylation marks simultaneously within a given cell population/tissue will be paramount in gaining a more holistic understanding of how H3 monoaminylation dynamics contribute to transcriptional regulation in brain health and disease.

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

# Methods

## General methods (equipment, reagents, chemicals, cell lines)

UV spectrometry was performed on the NanoDrop 2000c (Thermo Fisher Scientific) system. Biochemicals and medium were purchased from Thermo Fisher Scientific or Sigma-Aldrich unless otherwise stated. T4 DNA ligase, DNA polymerase and restriction enzymes were obtained from New England BioLabs. PCR amplifications were performed on the Applied Biosystems Veriti Thermal Cycler using either Taq DNA polymerase (Vazyme Biotech) for routine genotype verification or Phanta Max Super-Fidelity DNA Polymerase (Vazyme Biotech) for high-fidelity amplification. Site-specific mutagenesis was performed according to standard procedures of the QuickChange Site-Directed Mutagenesis Kit purchased from Stratagene (GE Healthcare) or Mut Express II (Vazyme Biotech). Primer synthesis and DNA sequencing were performed by Integrated DNA Technologies and Genewiz, respectively. PCR amplifications were performed on a Bio-Rad T100TM Thermal Cycler. Centrifugal filtration units were purchased from Millipore, and MINI dialysis units purchased from Pierce. Size-exclusion chromatography was performed on an AKTA FPLC system from GE Healthcare equipped with a P-920 pump and UPC-900 monitor. Sephacryl S-200 columns were obtained from GE Healthcare. All the western blots were performed using the primary antibodies annotated in Supplementary Table 8 and secondary antibodies annotated in Supplementary Table 9 following protocols recommended by the manufacturer. Blots were imaged on an Odyssey CLx Imaging System (Li-Cor). All uncropped and annotated western blots, dot blots and SDS−PAGE gels are included in Supplementary Fig. 1. Amino acid derivatives and coupling reagents were purchased from AGTC Bioproducts. Dimethylformamide (DMF), dichloromethane and triisopropylsilane were purchased from Thermo Fisher Scientific and used without further purification. Hydroxybenzotriazole and $O$-(benzotriazol-1-yl)-$N$,$N$,$N'$,$N'$-tetramethyluronium hexafluorophosphate (HBTU) were purchased from Thermo Fisher Scientific. Trifluoroacetic acid (TFA) was purchased from Thermo Fisher Scientific. $N$,$N$-diisopropylethylamine (DIPEA) was purchased from Thermo Fisher Scientific. Analytical reversed-phase HPLC (RP-HPLC) was performed on the Agilent 1200 series instrument with the Agilent C18 column (5 µm, 4 × 150 mm), used 0.1% TFA in water (HPLC solvent A) and 90% acetonitrile, 0.1% TFA in water (HPLC solvent B) as the mobile phases. Analytical gradients were 0–70% HPLC buffer B over 45 min at a flow rate of 0.5 ml min⁻¹, unless stated otherwise. Preparative scale purifications were conducted on the Agilent LC system. An Agilent C18 preparative column (15–20 µm, 20 × 250 mm) or a semi-preparative column (12 µm, 10 mm × 250 mm) was employed at a flow rate of 20 ml min⁻¹ or 4 ml min⁻¹, respectively. HPLC electrospray ionization MS (HPLC-ESI-MS) analysis was performed on the Agilent 6120 Quadrupole LC/MS spectrometer (Agilent Technologies). All immunoblotting experiments for which quantifications were not included were performed three times. HeLa (CRM-CCL-2) and HEK293T (CRL-3216) cell lines were obtained from the American Type Culture Collection (ATCC). Human tissue culture cell lines (HeLa, HEK293T) were imaged for appropriate morphology and tested negative for mycoplasma contamination.

## Expression of TG2 in HEK293T cells

The pShooter pCMV-nuc-myc vector expressing NLS-tagged WT human TG2 was used in our previous research[4]. The catalytically dead mutant TG2(C277A) plasmid was constructed by site-directed mutagenesis using 5′-GTCAAGTATGGCCAG**GCC**TGGGTCTTCGCCGCC-3′ and 5′-GGCGGCGAAGACCCA**GGC**CTGGCCATACTTGAC-3′ as primers (the mutation sites are labelled in bold). The gene sequences were confirmed by using the sequencing primer: 5′-GATGACCAGGGTGTGCTGCTG-3′. WT TG2 and the TG2(C277A) mutant were overexpressed in HEK293T cells using Lipofectamine 2000 Transfection Reagent (Thermo Fisher Scientific) according to the manufacturer's protocol. HEK293T cells

(ATCC) were cultured at 37 °C with 5% $CO_2$ in DMEM medium supplemented with 10% FBS (Sigma-Aldrich), 2 mM L-glutamine and 500 U ml⁻¹ penicillin and streptomycin. The cells were stimulated with 2 µM calcium ionophore (Sigma-Aldrich, A23187) for 6 h at 37 °C before lysis in DPBS buffer (Gibco), and then the expression of TG2 was detected by western blot analyses with anti-TGM2 antibody (CST, 3557).

## Knockout and rescue of TG2 expression in HeLa cells

Plasmids psPAX2 and pMD2G (gifts from Q.-E. Wang's laboratory) were used for viral packing and transfection; TGM2-1_pLentiCRISPR v2 (GenScript; the sequence is provided in the Supplementary Note) was used for gene knockout of *TGM2* in HeLa cells. The gRNA sequence of TGM2-1_pLentiCRISPR v2 was CGTCGTGACCAACTACAACT.

The TGM2-1_pLentiCRISPR v2-containing lentivirus was packed in HEK293T cells using Lipofectamine-2000-mediated transfection, where the DNA ratio of TGM2-1_pLentiCRISPR v2, psPAX2 and pMD2G was 4:3:1. Lenti-X concentrators (Clontech, 631231, 631232) were used to collect the packed lentivirus from the clarified supernatants. The viral supernatants were slowly titrated to HeLa cells cultured in DMEM containing 8 µg ml⁻¹ polybrene. Transfected cells were incubated under 37 °C with 5% $CO_2$ for 24 h before being split and cultured in DMEM containing 50–4,000 ng ml⁻¹ puromycin. Selection medium was replaced every 3-4 days to obtain resistant colonies. *TGM2*-knockout cell lines were validated using western blot analyses. WT TG2 and the TG2(C277A) mutant were overexpressed in *TGM2*-knockout HeLa cells using Lipofectamine 2000, as described above, and validated using western blot analyses.

## Salt extraction of histones from cells

The extraction of histones from cells was performed according to the previously described high-salt extraction method[46]. In brief, cell lysis solution was prepared using extraction buffer (10 mM HEPES pH 7.9, 10 mM KCl, 1.5 mM $MgCl_2$, 0.34 M sucrose, 10% glycerol, 0.2% NP40, protease and phosphatase inhibitors to 1 × from stock). After centrifuging, the pellet was extracted using a no-salt buffer (3 mM EDTA, 0.2 mM EGTA). After discarding the supernatant, the final pellet was extracted using high-salt buffer (50 mM Tris pH 8.0, 2.5 M NaCl, 0.05% NP40) in 4 °C cold room for 1 h. After centrifuging, the supernatant containing extracted histones was collected for further analyses.

## Cell fractionation

Cytosolic and nuclear fractions were prepared using NEPER nuclear and cytoplasmic extraction reagents (Thermo Fisher Scientific) according to the manufacturer's protocol. Histones were extracted from the pellet using the high-salt extraction protocol, as described above[46]. The purity of fractionation was evaluated using the following antibodies: anti-actin (cytosol), anti-MEK 1/2 (nucleoplasm; not provided in manuscript) and anti-H3 (chromatin).

## Pulse−chase experiments and inhibitor treatment

HEK293T and HeLa cells were cultured at 37 °C with 5% $CO_2$ in DMEM medium supplemented with 10% FBS (Sigma-Aldrich), 2 mM L-glutamine and 500 U ml⁻¹ penicillin and streptomycin. WT TG2 or TG2(C277A) mutant was overexpressed in HEK293T cells. The cultured cells were incubated with 500 µM monoamines (5-PT or histamine) for 6 h before the medium was changed to monoamine-free DMEM. Cells were cultured for an additional 6 h, 12 h or 18 h, after which they were washed with DPBS and collected, and the cytosolic and histone fractions were prepared as previously described[46,47]. Samples were separated on a single SDS−PAGE, transferred to a PVDF membrane and blotted with the indicated antibodies (or Cy5 dye) for further analyses. The cells were stimulated with 2 µM calcium ionophore (Sigma-Aldrich, A23187) during incubation with monoamine donors, as described above.

For the inhibitor treatment assays, 100 µM of TG2 inhibitors (ERW1041E or ZDON) were added to cell medium 2 h before adding the corresponding monoamine donors. Cells were incubated for additional

6 h, after which they were collected and histones were extracted and analysed, as described above. The samples were separated on a single SDS–PAGE, transferred to a PVDF membrane and blotted with the indicated antibodies. For the inhibitor-treated in vitro biochemical assays, where 1 µM NCPs were used as substrate and 0.1 µM TG2 was used as catalyst, 1 µM ERW1041E was added to the reaction systems for inhibiting the activity of TG2.

## Visualization of H3 serotonylation by CuAAC

HEK293T and HeLa cells were treated with 500 µM 5-PT and stimulated with 2 µM calcium ionophore (Sigma-Aldrich, A23187) during incubation, as described above. The cells were washed by DPBS, collected and then histone factions were extracted through high-salt extraction. Extracted histones were desalinated, lyophilized and then resuspended in DPBS buffer containing 0.4% SDS. A total of 50 µl of freshly dissolved histones was added to a premixed solution containing 3 µl of 10 mM Cy5-azide (Sigma-Aldrich, 777323), 10 µl of a 3:7 mixture of 50 mM $CuSO_4$ and 100 mM THPTA, and then vortexed. Thereafter, 5 µl of 100 mM freshly made TCEP was added to initiate the click reaction followed by incubation (1–2 h) at 30 °C. Then, 10 µl of 0.5 M EDTA was added to quench the reactions. Excess reagents were removed by $MeOH/CHCl_3$ protein precipitation or concentration–dilution using a 0.5 ml centrifugal filter (3K, Millipore). The pellets were washed by 500 µl $MeOH/H_2O$ (9:1) before a second centrifugation[48]. The air-dried protein samples were then analysed by SDS–PAGE, followed by in-gel imaging using the Odyssey CLx Imaging System (wavelength 680 nm).

## Expression of recombinant WT TG2 and C277A mutant

The pHis-hTGM2 plasmid was gifted from B. I. Lee (Addgene, 100719). The $His_8$-tagged TG2 C277A mutation was cloned by site-directed mutagenesis using pHis-hTGM2 as the template and the following primer sequences: 5′-GTCAAGTATGGCCAG**GCC**TGGGTCTTCGCCGCC-3′ and 5′-GGCGGCGAAGACCCA**GGC**CTGGCCATACTTGAC-3′. The gene sequences were confirmed by using the sequencing primer: 5′-GAT GACCAGGGTGTGCTGCTG-3′. The $His_8$-tagged WT and mutant TG2 proteins were expressed in E. coli Rosetta (DE3) cells with an overnight 0.125 M IPTG induction at 16 °C. The bacterial pellet was lysed by sonication and the lysate was cleared by centrifugation at 12,000 rpm for 30 min. The lysate was loaded onto HisTrap HP Column (GE Healthcare) and eluted on the AKTA FPLC, followed by desalting using Zeba Spin Desalting Columns (7 K MWCO, 10 ml) according to the manufacturer's protocol. Purified recombinant proteins were analysed by SDS–PAGE and concentrated using stirred ultrafiltration cells (Millipore) according to the manufacturer's protocol. The concentration of each protein was determined using 280 nm wavelength on a NanoDrop 2000c (Thermo Fisher Scientific).

## Peptide synthesis

Standard Fmoc-based solid-phase peptide synthesis (FmocSPPS) was used for the synthesis of peptides in this study. Generally, the peptides were synthesized on ChemMatrix resins with rink amide to generate C-terminal amides. Peptides were synthesized using manual addition of the reagents (using a stream of dry $N_2$ to agitate the reaction mixture). For amino acid coupling, 5 equiv. Fmoc protected amino acids was pre-activated with 4.9 equiv. HBTU, 5 equiv. hydroxybenzotriazole and 10 equiv. DIPEA in DMF and then reacted with the N-terminally deprotected peptidyl resin. Fmoc deprotection was performed in an excess of 20% (v/v) piperidine in DMF, and the deprotected peptidyl resin was washed thoroughly with DMF to remove trace piperidine. Cleavage from the resin and side-chain deprotection were performed with 95% TFA, 2.5% triisopropylsilane and 2.5% $H_2O$ at room temperature for 1.5 h. The peptides were then precipitated with cold diethyl ether, isolated by centrifugation and dissolved in water with 0.1% TFA followed by RP-HPLC and ESI-MS analyses. Preparative RP-HPLC was used to purify the peptides of interest.

For the synthesis of site-specific monoaminylated H3 peptides ($H3_{1-10}$, $H3_{1-15}$ and $H3_{1-21}$) for biochemical assays in this study, Fmoc-Glu(OAll)-OH was incorporated at position 5 for orthogonal deprotection and further monoaminylation. In brief, the peptides were deprotected by $Pd(PPh_3)_4$ and $PhSiH_3$ on resins[49] and then conjugated with monoamine donors (that is, histamine hydrochloride, serotonin hydrochloride[4] and acetonide-protected dopamine[50]) through PyAOP and DIEA catalysis[51].

For the synthesis of modified H3 peptide antigens ($H3_{1-10}$) in this study (as shown in Extended Data Fig. 4a), (1) Fmoc-Glu(OAll)-OH was incorporated at position 5 and either Fmoc-Lys(Boc)-OH or Fmoc-Lys(Me3)-OH was incorporated at position 4 on 2-Cl trityl resin through iterative FmocSPPS. (2) The deallylation was conducted using $Pd(PPh_3)_4$ and $PhSiH_3$, (3) followed by the coupling of Trt-protected histamine (4) and then acidolytic cleavage from the resin as well as global deprotection. Note that both the validated H3Q5his and H3K4me3Q5his antibodies were licensed to Millipore for sale (the catalogue numbers are provided below).

## Recombinant histone expression and purification

Recombinant human histones H2A, H2B, H3.2 and H4 were expressed in E. coli BL21 (DE3) or E. coli C41 (DE3), extracted by guanidine hydrochloride and purified by flash reverse chromatography, as previously described[46]. The purified histones were analysed by RP-LC–ESI-MS[46].

## Preparation of histone octamers and 601 DNA

Octamers were prepared as previously described[46]. In brief, recombinant histones were dissolved in unfolding buffer (20 mM Tris-HCl, 6 M GdmCl, 0.5 mM DTT, pH 7.5), and combined with the following stoichiometry: 1.1 equiv. H2A, 1.1 equiv. H2B, 1 equiv. H3.2, 1 equiv. H4. The combined histone solution was adjusted to 1 mg ml⁻¹ concentration and transferred to a dialysis cassette with a 7,000 Da molecular cut-off. Octamers were assembled by dialysis at 4 °C against 3 × 1 l of octamer refolding buffer (10 mM Tris-HCl, 2 M NaCl, 0.5 mM EDTA, 1 mM DTT, pH 7.5) and subsequently purified by size-exclusion chromatography on the Superdex S200 10/300 column. The fractions containing octamers were combined, concentrated, diluted with glycerol to a final 50% (v/v) and stored at −20 °C. The 147 bp 601 DNA fragment was prepared by digestion from a plasmid containing 30 copies of the desired sequence (flanked by blunt EcoRV sites on either site) and purified by PEG-6000 precipitation as described before[46].

## Mononucleosome assembly

Mononucleosome assembly was performed according to the previously described salt dilution method with slight modifications[46]. In brief, the purified WT octamers were mixed with 601 DNA (1:1 ratio) in a 2 M salt solution (10 mM Tris pH 7.5, 2 M NaCl, 1 mM EDTA, 1 mM DTT). After incubation at 37 °C for 15 min, the mixture was gradually diluted (9 × 15 min) at 30 °C by dilution buffer (10 mM Tris pH 7.5, 10 mM NaCl, 1 mM EDTA, 1 mM DTT). The assembled mononucleosomes were concentrated and characterized by native gel electrophoresis (5% acrylamide gel, 0.5× TBE, 120 V, 40 min) using ethidium bromide staining.

## In vitro TG2 (de)monoaminylation biochemical assays

TG2 (de)monoaminylation assays were generally performed in the buffer (pH 7.5) containing 50 mM Tris-HCl, 5 mM $CaCl_2$ and 2 mM DTT (freshly added). For H3 peptide ($H3_{1-21}$) (de)monoaminylation and monoamine-replacement, 2 mM peptides were treated with 100 µM TG2 (or cell lysates) at 37 °C in the presence (or absence) of the corresponding monoamines (4 mM) for 2 h and then analysed by LC–MS. For NCP (de)monoaminylation and monoamine replacement, 1 µM NCPs were treated with 0.1 µM TG2 at 37 °C in the presence (or absence) of the corresponding monoamaines (0.5 µM) for 2 h. The (de)monoaminylated NCPs were analysed by SDS–PAGE followed by western blot analysis. H3 was used as the loading control in SDS–PAGE and western blot analyses. Buffer exchange for monoamine-replacement assays was performed

using a 0.5 ml centrifugal filter (3K, Millipore) with a 120-fold (v/v) for the removal of excess monoamine from the old reaction buffer systems.

## Immunoprecipitation and pull-down of TG2–H3 thioester complexes

To capture the TG2–H3 thioester complex, 50 μM of free H3 proteins were treated with 50 μM of WT TG2 or TG2(C277A) mutant in a buffer (pH 7.0) containing 50 mM Tris-HCl, 5 mM CaCl$_2$ and 2 mM DTT (freshly added) at 37 °C for 1 h. His$_8$-tagged TG2 was first pulled-down by BSA-blocked Ni$^{2+}$-NTA agarose beads (Thermo Fisher Scientific). Next, the beads were washed three times with Tris-HCl buffer (pH 7.0), boiled, separated on SDS–PAGE and analysed by western blotting with anti-TGM2 and anti-H3 antibodies to detect the enrichment of H3.

## LC–MS/MS validation of histaminylated H3$_{1–15}$

Samples were analysed by LC–MS/MS (Dionex 3000 coupled to Q-Exactive mass spectrometer, Thermo Fisher Scientific). Peptides were separated by C-18 reversed phase chromatography (inner diameter, 75 μm, particle size, 3 μm, Nikkyo Technologies) using a gradient increasing from 1% B to 25% B in 16 min (A: 0.1% formic acid, B: 80% acetonitrile in 0.1% formic acid). The mass spectrometer was operated in parallel reaction monitoring (PRM) mode[4]–R35 (MS and MS/MS resolution of 70,000 and 35,000, respectively)[52] with an AGC target, $5 \times 10^5$, maximum injection time of 60 ms and an isolation $m/z$ window of 1.3). MS was acquired from $m/z$ 300 to 1,650 while $m/z$ 100 was set as lowest mass for MS/MS. Charge states 2+ to 5+ of the modified peptide: (ARTKQ(histamine)TARKSTGGKA-NH$_2$) were targeted. An energy of 25 NCE was used for the peptide in charge states 3+ and 4+ while NCE of 35 was used for the doubly charge version. Extended Data Fig. 3d represents a high-resolution, high-accuracy tandem mass spectra (sample: +histamine/+TG2) of the doubly charged peptide: ARTKQTARKSTGGKA-NH2 modified by histamine at Gln5 ($m/z$ 551.9930 (1.5 ppm)). The full amino acid sequence was accounted for. Selected fragment ions, including y10/b5 and y11/b4 that identify the histamine modified glutamine are annotated.

## Identification of H3 glutamine 5 histaminylation from cells

HEK293T cells transfected with WT TG2 or TG2(C277A) mutant plasmids were treated with 500 μM histamine, as described above. Histones were then extracted from collected cell pellets, as described previously[53], for further MS analysis. In brief, histones were extracted with chilled 0.2 M sulfuric acid (5:1, sulfuric acid: pellet) and incubated with constant rotation for 4 h at 4 °C, followed by precipitation with 33% trichloroacetic acid overnight at 4 °C. The supernatant was then removed and the tubes were rinsed with ice-cold acetone containing 0.1% hydrochloric acid, centrifuged and rinsed again using 100% ice-cold acetone. After centrifugation, the supernatant was discarded and the pellet was dried using a lyophilizer. The pellet was dissolved in 50 mM ammonium bicarbonate (pH 8.0), and histones were subjected to derivatization using 5 μl of propionic anhydride and 14 μl of ammonium hydroxide (Sigma-Aldrich) to balance the pH at 8.0. The mixture was incubated for 15 min and the procedure was repeated. Histones were then digested with 1 μg of sequencing grade trypsin (Promega) diluted in 50 mM ammonium bicarbonate (1:20, enzyme: sample) overnight at room temperature. Derivatization reaction was repeated to derivatize peptide N termini. The samples were dried by a lyophilizer. Before MS analysis, the samples were desalted using a 96-well plate filter (Orochem) packed with 1 mg of Oasis HLB C-18 resin (Waters). In brief, the samples were resuspended in 100 μl of 0.1% TFA and loaded onto the HLB resin, which was previously equilibrated using 100 μl of the same buffer. After washing with 100 μl of 0.1% TFA, the samples were eluted with a buffer containing 70 μl of 60% acetonitrile and 0.1% TFA and then dried by lyophilizer.

Samples were analysed using nano LC coupled online with MS/MS (nLC–MS/MS). In brief, the samples were resuspended in 10 μl of 0.1% TFA and loaded onto the Dionex RSLC Ultimate 300 (Thermo Fisher Scientific), coupled online with an Orbitrap Fusion Lumos (Thermo Fisher Scientific). Chromatography separation was performed using a two-column system, consisting of a C-18 trap cartridge (300 μm inner diameter, 5 mm length) and a picofrit analytical column (75 μm inner diameter, 25 cm length) packed in-house with reversed-phase Repro-Sil Pur C18-AQ 3 μm resin. Histone peptides were separated using a 30 min gradient from 4 to 30% buffer B (buffer A: 0.1% formic acid; buffer B: 80% acetonitrile + 0.1% formic acid) at a flow rate of 300 ml min$^{-1}$. The mass spectrometer was set to acquire spectra in a data-independent acquisition mode. The full MS scan was set to 300–1,100 $m/z$ in the orbitrap with a resolution of 120,000 (at 200 $m/z$) and an AGC target of $5 \times 10^5$. MS/MS was performed in the orbitrap with sequential isolation windows of 50 $m/z$ with an AGC target of $2 \times 10^5$ and an HCD collision energy of 30.

Targeted MS/MS was performed for the endogenous Q5his peptide ($m/z$ 455.7587) and compared to the MS/MS spectra of a synthetic H3Q5his peptide. Data were manually inspected and the peak intensity was obtained by calculating the area of the extracted ion chromatogram. Histone peptides raw files were imported into EpiProfile 2.0 software[54] to also quantify acetylated, methylated and phosphorylated peptides to ensure the quality of sample preparations and LC–MS/MS analysis.

## ITC

For ITC measurements, synthetic histone peptides and recombinant proteins (CHD1 chromodomain, TAF3 PHD finger, WDR5 WD40 domain, BPTF PHD finger, JMJDA Tudor domain or JARID1A PHD finger; purified, as previously described[5]) were extensively dialysed against ITC buffer: 100 mM NaCl and 20 mM Tris pH 7.5. The titrations were performed using the MicroCal iTC200 system (GE Healthcare) at 25 °C. Each ITC titration consisted of 17 successive injections with 0.4 μl for the first and 2.4 μl for the rest. Peptides were titrated into proteins in all of the experiments. The resultant ITC curves were processed using Origin 7.0 software (OriginLab) according to the 'one set of sites' fitting model. ITC statistics are provided in Supplementary Table 1.

## Recombinant protein cloning and purification of WDR5 WT, WDR5 mutant and MLL1 complex, the latter of which was used in MALDI-TOF experiments

Full-length WDR5 and truncated WDR5 residues 22 to 334 (WDR5(22–334)) were cloned into pET28b vector for protein purification. Full-length WDR5(K259A) and WDR5(K259E) mutants were generated using a site-directed mutagenesis kit (Agilent). All proteins were expressed in the *E. coli* BL21 DE3 (Novagen) and induced overnight by 0.2 mM isopropyl β-D-thiogalactoside at 16 °C in the LB medium. The collected cells were suspended in 500 mM NaCl, 20 mM Tris, pH 7.5. After cell lysis and centrifugation, the supernatant was applied to HisTrap column (GE Healthcare). After washing 5 column volumes with the suspension buffer, the protein was eluted with buffer containing 100 mM NaCl, 20 mM Tris pH 7.5, 500 mM imidazole, and cut with Thrombin enzyme overnight. Proteins were further purified by the HiTrap SP (GE Healthcare) cation-exchange column and a HiLoad 16/60 Superdex 75 (GE Healthcare) gel-filtration column using AKTA Purifier 10 systems (GE Healthcare). All proteins were stored in 100 mM NaCl, 20 mM Tris, pH 7.5 at around 10 mg ml$^{-1}$ in an −80 °C freezer.

Human MLL1 constructs (3745–3969), as well as full-length human WDR5, RBBP5, DPY30 and ASH2L (95–628) proteins, were individually expressed in *E. coli* BL21 cells. All proteins were induced overnight with 0.2 mM isopropyl β-D-thiogalactoside at 16 °C in LB medium. Cell pellets were suspended in lysis buffer (20 mM Tris-HCl, pH 8.0, 500 mM NaCl, 5% glycerol, 1 mM DTT). After cell lysis and centrifugation, the supernatants were purified using HisTrap columns (GE Healthcare) or Glutathione Sepharose 4B beads (GE Healthcare), followed by enzyme digestion to remove tags. All proteins were further purified

on HiTrap SP (GE Healthcare) cation-exchange columns or HiTrap Q (GE Healthcare) anion-exchange columns. MLL1, WDR5 and DPY30 were further purified with the HiLoad 10/300 Superdex 75, while ASH2L and RBBP5 were further purified with the HiLoad 10/300 Superdex 200. The buffer for gel-filtration chromatography contained 50 mM Tris-HCl, pH 7.5, 300 mM NaCl, 1 mM DTT and 10% glycerol. Purified proteins were concentrated to 10–20 mg ml$^{-1}$ and stored at −80 °C. Note that, for MLL1 complex experiments presented in Fig. 2c and Extended Data Fig. 6a,b, the MLL1 complex was purchased from Active Motif (31423).

## MALDI-TOF analysis of MLL1 enzymatic activity

H3K4 methyltransferase assays were conducted by combining 1.2 μM of the MLL(3745–3969)–WDR5–RBBP5–ASH2L–DPY30 complex with 10 μM histone H3 peptide (1–15) and 250 μM methyl-*S*-adenosyl-methionine in 50 mM Tris, pH 8.5, 50 mM KCl, 5 mM dithiothreitol, 5 mM MgCl$_2$ and 5% glycerol at 15 °C. The reactions were quenched by the addition of HPLC solvent A (H$_2$O + 0.1% TFA) and were desalted using C18 ZipTip (Millipore) according to the manufacturer's protocol before being diluted 1:1 with α-cyano-4- hydroxycinnamic acid (CHCA) matrix in 50% ACN plus 20% acetone with 0.1% TFA and spotted on a MALDI-TOF plate for analysis. The samples were analysed using the Bruker UltrafleXtreme MALDI TOF/TOF mass spectrometer and data were analysed using Bruker Compass flexAnalysis v.3.4.

## In vitro enzymatic assays with recombinant MLL1/2/3/4 and SETD1A/B complexes for LC–MS/MS

H3K4 methyltransferase assays were conducted, unless otherwise indicated, by combining 1.2 μM of either MLL1 (Active Motif, 31423), MLL2 (Active Motif, 31498), MLL3 (Active Motif, 31478), MLL4 (Active Motif, 31499), SETD1A (Active Motif, 81341) or SETD1B (Active Motif, 81342) complexes with 10 μM histone H3 peptide (unmodified versus H3Q5his; 1–21) and 100 μM *S*-adenosyl methionine in 50 mM Tris, pH 7.5, 50 mM KCl, 5 mM DTT, 5 mM MgCl$_2$ and 5% glycerol at 25 °C for 3 h. The reactions were quenched by the addition of HPLC solvent A (H$_2$O + 0.1% TFA).

**Histone peptide derivatization and digestion.** Derivatization of samples was performed as previously described[53]. In brief, the samples were dissolved in 25 μl of a solution containing 50 mM ammonium bicarbonate, pH 8.0 and 20% acetonitrile. In the fume hood, the samples were mixed with 2 μl of propionic anhydride and 10 μl of ammonium hydroxide (all Sigma-Aldrich) to balance the pH at 8.0. The mixture was incubated at room temperature for 15 min and the procedure was repeated. The samples were digested with 500 ng of sequencing-grade trypsin (Promega) diluted in 50 mM ammonium bicarbonate overnight at room temperature. The derivatization reaction was repeated to derivatize peptide N termini. The samples were then dried in a vacuum centrifuge.

**LC–MS/MS acquisition and analysis.** Before MS analysis, the samples were desalted using a 96-well plate filter (Orochem) packed with 1 mg of Oasis HLB C-18 resin (Waters). In brief, the samples were resuspended in 100 μl of 0.1% TFA and loaded onto the HLB resin, which was previously equilibrated using 100 μl of the same buffer. After washing with 100 μl of 0.1% TFA, the samples were eluted with a buffer containing 70 μl of 60% acetonitrile and 0.1% TFA and then dried in a vacuum centrifuge.

The samples were loaded onto the Dionex RSLC Ultimate 300 (Thermo Fisher Scientific) system, coupled online with the Orbitrap Fusion Lumos (Thermo Fisher Scientific). Chromatography separation was performed using a two-column system, consisting of a C-18 trap cartridge (300 μm inner diameter, 5 mm length) and a picofrit analytical column (75 μm inner diameter, 25 cm length) packed in-house with reversed-phase Repro-Sil Pur C18-AQ 3 μm resin. The samples were separated using a 45 min gradient from 1 to 30% buffer B (buffer A, 0.1% formic acid; buffer B, 80% acetonitrile + 0.1% formic acid) at a flow rate of 300 nl min$^{-1}$. The mass spectrometer was set to acquire spectra in a data-independent acquisition mode using isolation

windows as previously described[55]. In brief, the full MS scan was set to 300–1,100 *m/z* in the orbitrap with a resolution of 120,000 (at 200 *m/z*) and an AGC target of $5 \times 10^5$. MS/MS was performed in the orbitrap with sequential isolation windows of 50 *m/z* with an AGC target of $2 \times 10^5$ and an HCD collision energy of 30.

Targeted MS/MS was performed for H3Q5his (*m/z* 455.7587), H3K4me1Q5his (*m/z* 462.7665), H3K4me2Q5his (*m/z* 441.7613) and H3K4me3Q5his (*m/z* 448.7692) peptides (targeted MS/MS was similarly performed for H3Q5ser, H3K4me1Q5ser, H3K4me2Q5ser and H3K4me3Q5ser, as presented in Extended Data Fig. 6a). Data were manually inspected, and the peak intensity was obtained by calculating the area of the extracted ion chromatogram. To achieve the relative abundance of PTMs, the sum of all different modified forms of a histone peptide was considered as 100%, and the area of the particular peptide was divided by the total area for that histone peptide in all of its modified forms.

## Histone tail peptide IPs against recombinant WDR5 and MLL1 complex

Biotinylated unmodified H3, H3Q5ser, H3Q5his, H3K4me3, K4me3Q5his or H3k4me3Q5ser peptides (2 μg; 1–21) were resuspended with 25 μl of prewashed immobilized Streptavidin beads (DynaBeads Streptavidin M-280) in 0.01% DPBS/Triton-X 100, with subsequent incubation (rotating) for 1 h at room temperature. For each IP, 1 μg of full-length recombinant WDR5 (purified as described above) or 1 μg of MLL1 complex (Active Motif, 31423) was added to the beads in 1 ml of binding buffer (250 mM KCl, 25mM HEPES pH 7.5, 5 mM MgCl$_2$, 0.1% NP-40, 5% glycerol 1 mM DTT and 4% BSA), and each sample was rotated at 4 °C overnight. IPs were then centrifuged for 1 min at 1,000 rpm to pellet the beads. Beads were subsequently washed six times in binding buffer substituted with 500 mM KCl and 0.2% NP-40 (with no BSA). The beads were then washed once in cold DPBS and proteins eluted by boiling for 8 min in 30 μl of denaturing sample buffer before loading onto a gel.

## Crystallization and X-ray structure determination

Truncated WDR(22–334) was firstly incubated with H3Q5his peptide at a molar ratio 1:2 for 1 h. Crystallization was performed by the sitting-drop vapour diffusion method under 18 °C by mixing equal volumes (1–2 μl) of protein and reservoir solution. The crystal was obtained at the condition 0.1 M sodium citrate tribasic dihydrate (pH 5.5), 22% polyethylene glycol (PEG) 3350 and 0.1% *n*-octyl-β-D-glucoside at 18 °C. The crystals were briefly soaked in the cryo-protectant and were flash-frozen in liquid nitrogen for data collection at 100 K. Complex datasets were collected at beamline BL17U at the Shanghai Synchrotron Radiation Facility. All data were indexed, integrated and merged using the HKL2000 software package[56]. The complex structures were solved by molecular replacement using MOLREP[57]. All structures were refined using PHENIX[58], with iterative manual model building using COOT[59]. Model geometry was analysed with PROCHECK. The electron density of H3Q5 was visible while the histamine modification density was not clear. In the WDR5–H3Q5his structure and MLL3–RBBP5–ASH2L–H3 complex structure (Protein Data Bank (PDB): 5F6K), the model of histamine was built based on the orientation of the H3Q5 residue and restricted within the Ramachandran plot (favoured (95.08%), allowed (4.92%), outliers (9%)). All structural figures were created using PYMOL (http://www.pymol.org/). See Extended Data Table 1.

## High-salt extraction of soluble nuclear and chromatin bound fractions

The generation of nuclear soluble extracts (NE) and chromatin bound (CB) fractions was performed as previously described[33] from cells over-expressing 3X-FLAG tagged WDR5 (Addgene, 59974) or a 3× Flag-tagged WDR5(K259A) mutant (generated using a site-directed mutagenesis kit, NEB E0554S, and confirmed by sequencing). Cells were collected, washed in PBS and resuspended in a low-salt buffer (LSB) containing 20 mM HEPES pH 7.9, 25% glycerol, 1.5 mM MgCl$_2$, 2 mM EDTA, 1 mM

DTT and Halt Protease and Phosphatase Inhibitor Cocktail. The cells were then incubated on ice for 15 min to allow swelling. To lyse the cells, non-ionic detergent NP-40 was added (final concentration of 0.75%), and the mixture was gently passed through a 21-gauge needle ten times. The nuclei were collected by centrifugation at 1,100$g$ for 5 min at 4 °C, and the supernatant was collected as the cytoplasmic extract. The nuclei were washed twice with LSB and resuspended in 500 µl of LSB. The pelleted nuclear volume (PNV) was calculated by subtracting 500 µl LSB from the total volume. Nuclei were then repelleted and resuspended in half PNV of LSB. An equal volume of high-salt buffer (20 mM HEPES pH 7.9, 25% glycerol, 1.5 mM MgCl$_2$, 1.6 M NaCl, 1 mM DTT, Halt Protease and Phosphatase Inhibitor Cocktail) was added dropwise while vortexing at low speed to reach a final NaCl concentration of 400 mM. The samples were incubated at 4 °C with rotation for 1 h before being centrifuged at 21,000$g$ for 10 min at 4 °C. The supernatant was collected as the soluble nuclear extract. The pellet (chromatin fraction) was washed twice with 400 mM NaCl high-salt buffer for 10 min each with shaking, then pelleted and resuspended in 1× SDS–PAGE loading dye (final concentrations: 50 mM Tris-HCl pH 6.8, 3% SDS, 10% glycerol, 5% β-mercaptoethanol, 0.002% bromophenol blue). The samples were boiled at 95 °C for 5 min and cooled on ice three times before the chromatin was sheared by sonication.

## Enzymatic assays for antibody validations

For assessments of TG2-mediated transamidation of histamine to histone H3, 0.25 µg guinea pig TG2 (Zedira, T006), 5 mM histamine (Sigma-Aldrich) and 10 µg of recombinant H3.2 were combined with enzymatic buffer containing 250 mM tris-acetate (pH 7.5), 8.75 mM CaCl$_2$ and 1× protease inhibitor cocktail, followed by incubation for 3 h at room temperature. After incubation, enzymatic reactions were boiled with Laemmli buffer and then run on 4–12% NuPage BisTris gels (Invitrogen) and blotted, as described previously. Enzymatic assays were also performed using serotonin and dopamine to confirm the specificity of our in-house anti-H3Q5his antibody. Enzymatic assays were also performed using reconstituted unmodified (81070) versus K4me3 (31584) mononucleosomes from Active Motif.

## Animals

Male and female mice (C57BL/6J; aged 8–10 weeks) were purchased from The Jackson Laboratory. Animals were group housed (2–5 per cage) under a 12 h–12 h light–dark cycle (lights on from 07:00 to 19:00) at constant temperature (23 °C) with ad libitum access to food and water. All animal protocols were approved by the IACUC at the Icahn School of Medicine at Mount Sinai (ISMMS). No wild animals or field collected samples were used in this study. Adequate sample sizes were generally determined based on intersample variability. Throughout the Article, we determined the significance of results based on a general confidence interval of 95%. We do not include specific justifications of sample size within the methods (such as power analyses), as sample sizes were based on extensive laboratory experience with these end points. The sample sizes chosen are consistent with those used by others in the field to achieve statistically significant results comparing stressed versus control animals. Where appropriate, animals were randomly assigned to groups (segregated by viral treatments, or ZT). Tissue samples were not pooled from multiple animals in these studies for western blotting and RNA-seq experiments (that is, each $n$ represents a discrete datapoint). For CUT&RUN, each replicate per antibody was pooled from punches from three animals (per $n$) for initial processing, with 3 independent biological replicates conducted ($n = 3$) per antibody. For all viral experiments (RNA-seq, western blotting and behaviour), investigators were blinded to conditions such viral treatment before analysis. No data were excluded from these studies.

## Immunoblotting analysis of the brain

Brain tissues were extracted from euthanized mice and immediately frozen whole. Brains were later sectioned using razor blades and a brain block to 1 mm thickness, with tissue punches (1–2 mm) collected for corresponding brain regions. To purify nuclear fractions, punches were homogenized in buffer A containing 10 mM HEPES (pH 7.9), 10 mM KCl, 1.5 mM MgCl$_2$, 0.34 M sucrose, 10% glycerol, 1 mM EDTA and 1× protease inhibitor cocktail. After homogenization, 0.1% Triton X-100 was added to each homogenate, incubated and rotated at 4 °C for 30 min and then centrifuged for 5 min at 1,300$g$ at 4 °C. Supernatants containing cytosolic fractions were discarded, and the nuclear pellets were resuspended in buffer A to remove any remaining cytosolic contamination, followed by centrifugation for 5 min at 1,300$g$ at 4 °C. After centrifugation, the supernatants were discarded and the pellets were resuspended and sonicated in sample buffer containing 0.3 M sucrose, 5 mM HEPES, 1% SDS and 1× protease inhibitor cocktail. Protein concentrations were measured using the DC protein assay kit (Bio-Rad), and 1–20 µg of protein was loaded onto 4–12% NuPage BisTris gels (Invitrogen) for electrophoresis. Proteins were then transferred to PVDF membranes and blocked for 30 min in 5% milk in PBS + 0.1% Tween-20 (PBST), followed by incubation with primary antibodies overnight at 4 °C. For competition assays, antibodies were pre-incubated with indicated peptides at a 5:1 ratio for 1 h at room temperature before being incubated with the membrane. The following antibodies were used: rabbit anti-H3Q5his (1:200, Millipore, ABE2578), rabbit anti-H3K4me3Q5his (1:500, Millipore, ABE2605), rabbit anti-H3K4me3 (1:1000, Abcam, ab8580) and rabbit anti-H3 (1:50,000, Abcam, ab1791). The next day, the membranes were washed three times in PBST (10 min) and incubated for 1 h with horseradish-peroxidase-conjugated anti-rabbit secondary antibody (Bio-Rad 170-6515; 1:10,000; 1:50,000 for anti-H3 antibody) in 5% milk/PBST at room temperature. After three final washes with PBST, bands were detected using enhanced chemiluminescence (ECL; Millipore). Densitometry was used to quantify protein bands using ImageJ Software (NIH), and proteins were normalized to total H3 or H4. For peptide dot blots, peptides (unmodified versus H3Q5his versus H3Q5ser versus H3Q5dop; 1–10) were dotted as progressive protein concentrations (0.25, 0.5, 1 µg) on a nitrocellulose membrane. Membranes were left to dry at room temperature for 1 h and then blocked in 5% milk/PBST for 1 h. Membranes were treated similar to that described above.

## ELISAs

Serotonin and histamine ELISAs were performed using kits from Abcam (ab133053 and ab213975). Mouse brain was collected across the ZT, sliced and punched bilaterally for TMN. TMN punches were resuspended in 100 µl of hypotonic lysis buffer and allowed to swell. Swollen tissue was then denounced homogenized and centrifuged at 20,000 rcf for 10 min to pellet insoluble cellular debris. Once pelleted, the supernatant was collected and used directly as substrate in the ELISA. Both histamine and serotonin ELISAs were performed according to the manufacturer's instructions. UV-Vis absorbance for all plates were imaged using a spectramax id5 multi-mode microplate reader. Histamine ELISAs were read at 450 nm absorbance while serotonin ELISAs were read at 405 nm. Calculations were performed according to the manufacturer's instructions.

## AAV constructs and viral transduction

AAV H3.3 constructs (empty versus WT versus H3.3(Q5A)-Flag-HA) were generated and validated, as previously described[12]. All three vectors contain an IRES-driven GFP fluorescent tag to allow visualization of the injection site during tissue dissection. Animals were anaesthetized with isoflurane (1–3%) and positioned in a stereotaxic frame (Kopf instruments) and 0.5 µl of viral construct was infused bilaterally into TMN using the following coordinates; anterior–posterior (AP) −2.0, medial–lateral (ML) + 0.6, dorsal–ventral (DV) −5.2. After surgery, mice received meloxicam (1 mg per kg) subcutaneously and topical antibiotic treatments for 3 days. All tissue collections or behavioural testing commenced 21 days after surgery to allow for maximal expression of the viral constructs.

## Immunohistochemistry

Mice were anaesthetized with ketamine–xylazine (100 and 12 mg per kg) intraperitoneally (i.p.), and then perfused transcardially with cold phosphate-buffered saline (PBS 1×) followed by 4% paraformaldehyde in 1× PBS. Next, brains were post-fixed in 4% paraformaldehyde overnight at 4 °C and then transferred into 30% sucrose/PBS 1× for 2 days. The brains were then cut into serial 40 μm coronal slices. Free-floating TMN slices were washed three times in Tris-buffered saline (TBS 1×), incubated for 30 min in 0.2% Triton X-100/1× TBS, to permeabilize tissue, and then incubated for 1 h at room temperature in blocking buffer (0.3% Triton X-100, 3% donkey serum, 1× TBS). Brain slices were then incubated overnight at room temperature with mouse anti-GFP (1:200; Abcam, ab65856) and HA-488 (1:200; Life Technologies, Alexa Fluor SC-805). The next day, brain slices were washed three times in 1× TBS and then incubated for 2 h at room temperature with a fluorescent-tagged Alexa Fluor 568 anti-mouse secondary antibody (1:500; Life Technologies A11004). Brain sections were then washed three times in 1× TBS, incubated with DAPI (1:10,000, Thermo Fisher Scientific, 62248) for 5 min at room temperature, mounted onto Superfrost Plus slides (Thermo Fisher Scientific) and then coverslipped with Prolong Gold (Invitrogen). Immunofluorescence was visualized using a confocal microscope (Zeiss, LSM 780).

## RNA-seq analysis

**RNA extractions, library preparation and sequencing.** Brain tissues were collected every 4 h across the 24 h zeitgeber in non-virally transduced C57BL/6J mice (aged 8–10 weeks), or 21 days after viral transduction. All brain tissues were immediately frozen after collection. For non-virally transduced brains, tissues were sectioned at 1 mm thick and TMN tissues were collected by tissue punch (1 mm). For virally infused brains, tissues were sectioned at 150 μm on the cryostat, and GFP was illuminated using the NIGHTSEA BlueStar flashlight to microdissect virally infected tissues. Tissues were resuspended in 800 μl of Trizol and homogenized using a small dounce homogenizer (30 strokes loose, 30 strokes tight) at room temperature. Chloroform was added and the aqueous phase isolated. 70% ethanol was added 1:1, and then passed over an RNAeasy minelute column. The Qiagen RNAeasy MicroKit protocol was followed, including all optional steps and DNase treatment. RNA was eluted in 15 μl and quantified using the NanoDrop spectrophotometer. The RNA quality was assessed using a Tapestation RNA screentape (Agilent). Then, 100 μg of total RNA was used as input for library preparation using the Illumina Stranded mRNA Prep, Ligation kit. A total of 14 cycles of PCR amplification was performed, libraries were pooled at an equimolar concentration and sequenced on the Illumina HiSeq 2500 or NovaSeq X+ sequencer by the NYU Genome Technology Center.

**RNA-seq data analysis.** Raw sequencing reads were demultiplexed using bcl2fastq2 (Illumina, v.2.20). The samples were aligned to the GRCm38 mouse genome using STAR (v.2.7.11b) alignReads in mode BAM SortedByCoordinate[60]. Gene counts were generated using htseq-count (HTSeq v.2.0.5) with the following parameters: --format=bam --minaqual=10 --type=exon --idattr=gene_name --stranded=yes --mode=union using the Ensembl v93 annotation[61]. Gene counts were normalized using DEseq2[62] (v.1.44.0) before analysis using JTKcycle[37] (v.3.1) to identify cycling genes from the dataset using the parameters jtkdist (varying depending on replicates), periods(2:6) and jtk. init(periods,4). Genes with $P_{adj} < 0.05$ were deemed to be significant and $Z$ scores were computed in R using tidyverse (v.2.0.0). heatmap.2 in R was used to visualize the cycle genes (gplots v.3.1.3.1) across the zeitgeber. Cyclic genes were further assessed using ChEA in Enrichr (https://maayanlab.cloud/Enrichr/, (v.3.2)), which infers transcription factor regulation from integration of previous genome-wide chromatin immunoprecipitation (ChIP) analyses. Further ontology was also conducted using Enrichr. Odds ratios were calculated using the GeneOverlap

R package (v.1.26.0). DEseq2 was run to perform pairwise differential expression analyses between H3.3 and GFP viral treated samples (to ensure limited to no significant differences between control groups; number of differentially expressed genes between GFP versus H3.3 WT at: ZT0 = 1, ZT4 = 0, ZT8 = 0, ZT12 = 40, ZT16 = 1, ZT20 = 1) before combining them together to run JTKcycle, as previously described. Differentially expressed genes were defined at FDR < 0.01. Overlap of JTK cycle genes with peak lists from CUT&RUN-seq (see below) was performed in R using dplyr (v.1.1.4). Individual circadian rhythm controlling genes identified in Enrichr were highlighted on heat maps manually.

## RT–qPCR

Cell pellets were resuspended in 200 μl Trizol and homogenized using a clean pestle. A total of 600 μl of Trizol was added to the homogenate and allowed to rest at room temperature for 4–5 min. 160 μl chloroform was then added to each tube and mixed vigorously for 15 s, followed by a 3 min rest period. The samples were then centrifuged at 12,000$g$ for 15 min at 4 °C. The top layer was decanted and transferred to a separate Eppendorf tube. 1 volume of 70% ethanol was added to the lysate and mixed thoroughly. The lysate was then transferred to a RNAeasy mini column and centrifuged at 12,000$g$ for 1 min. A master mix of DNase (79256) and RDD buffer were added to the RNAeasy spin column and allowed to incubate at room temperature for 15 min. Then, 350 μl of RW1 was added to the column and centrifuged at 12,000$g$ for 1 min at room temperature. The RNAeasy columns were then transferred to a new 2 ml collection tube and 500 μl of RPE buffer was added to the column and centrifuged at 12,000$g$ for 1 min. Next, 500 μl of 80% ethanol was added to the tube and centrifuged at 12,000$g$ for 1 min. An additional 5 min spin at maximum speed was performed to remove residual ethanol. RNA was eluted in 13 μl of double-distilled $H_2O$.

To convert RNA into cDNA, 5 μg of total RNA was mixed with 1 μl of 50 μM oligo dT, 1 μl of 10 mM dNTPs and allowed to prime at 65 °C for 5 min, followed by a 1 min incubation of ice. Then, 4 μl of 5× SSIV buffer (18090010), 1 μl of 100 mM DTT, 1 μl of RNAseOUT recombinant RNase inhibitor and 1 μl of Superscript IV reverse transcriptase were added to the sample, and allowed to incubate at 52 °C for 10 min, followed by inactivation by heating at 80 °C for 10 min. Subsequent cDNA was diluted 1:10 and 1 μl per well for RT–qPCR. Data were analysed using the $\Delta\Delta C_t$ method using the 18S RNA gene for normalization. A list of primers used in this study is provided in Supplementary Table 10.

## CUT&RUN–seq

**Cells.** HeLa cells were grown in DMEM with glucose, 10% fetal bovine serum (Sigma-Aldrich) and 1× penicillin–streptomycin (Gibco). Cells were cultured at 37 °C at 5% $CO_2$. Cells were collected on ice using a cell scraper in the cold room, and washed in PBS. Cells from each 10 cm plate were resuspended in 1 ml of nuclear extract (NE) buffer (20 mM HEPES-KOH, pH 7.9, 10 mM KCl, 0.5 mM spermidine, 0.1% Triton X-100, 20% glycerol and freshly added protease inhibitors (Halt Protease Inhibitor Cocktail, EDTA-free, Thermo Fisher Scientific) and passed through a 21 gauge needle 20 times to lyse the cells. Nuclei were pelleted at 1,100$g$ for 5 min at 4 °C and the supernatant was discarded. Nuclei were washed again in 1 ml NE buffer and counted. In total, 30,000 nuclei were used per biological replicate.

**TMN.** Brain tissues were collected every 4 h across the 24 h zeitgeber. Tissues were collected after vehicle/zolpidem treatment at ZT = 20. Brain tissues were extracted from euthanized mice and immediately frozen whole in methylbutane (Thermo Fisher Scientific, 277258). Brains were later sectioned using razor blades and a brain block to 1 mm thickness, with 1 mm tissue punches of the TMN bilaterally. For CUT&RUN, each biological replicate per antibody was pooled from three punches animals (per $n$) for initial processing, with three independent biological replicates conducted ($n$ = 3) per antibody. Each pool of tissue punches was resuspended in 500 μl of NE buffer, and homogenized using 30

strokes of a plastic pestle (Sigma-Aldrich) in a 1.7 ml tube (Eppendorf). Nuclei were pelted at 1,100g for 5 min at 4 °C in a swinging-bucket rotor and supernatant discarded. Nuclei were washed again in 500 µl NE buffer and counted. In total, 30,000 nuclei were used per biological replicate.

**CUT&RUN.** BioMag Plus Concanavalin A beads (Polysciences) were prepared (15 µl bead slurry per reaction) by washing three times with binding buffer (20 mM HEPES-KOH, pH 7.9, 10 mM KCl, 1 mM CaCl$_2$, 1 mM MnCl$_2$), and resuspending in the original volume[63–65]. 15 µl was aliquoted into a 1.7 ml DNA low-bind tube (Eppendorf) and 300 µl binding buffer was added to each tube. Nuclei from cells or tissues were added (30,000 nuclei in 300 µl of NE buffer) to each tube, and rotated end over end at room temperature for 10 min. The bead-bound nuclei were washed with 1 ml wash buffer (WB: 20 mM HEPES, pH 7.5, 150 mM NaCl, 0.1% Triton X-100, 0.1% Tween-20, 0.5 mM spermidine, 0.1% BSA, freshly added protease inhibitors) three times. All washes were done so to minimize pipetting and beads mixed by inversion and light flicking of the tube. Beads were resuspended in 100 µl of antibody buffer (1 ml wash buffer with 2 mM EDTA) and mixed by flicking. Then, 2 µl of antibody (1:50) was added to each tube and mixed by flicking (H3K4me2 (Active Motif, 39141); WDR5 (CST, D9E1I), H3K4me3 (Epicypher, 13-0041), H3Q5his (Millipore, ABE2578), H3K4me3Q5his (Millipore, ABE2570), H3K4me3Q5ser (Millipore, ABE2580)). Bead-bound nuclei were incubated on a mixer (tubes on their side at ~20 degree upward angle) with primary antibodies overnight at 4 °C. Nuclei were washed twice the next day with 1 ml of cold WB. Nuclei were resuspended in 50 µl of cold wash buffer by flicking, and 2.5 µl of pAG-MNase (Epicypher, 15-1016) was added and mixed by flicking and incubated for 1 h at 4 °C on the same mixer. Nuclei were then washed four times with 1 ml ice-cold wash buffer, followed by one wash in 1 ml low-salt rinse buffer (20 mM HEPES, pH 7.5, 0.5 mM spermidine, 0.1% Tween-20 and 0.1% Triton X-100). Nuclei were resuspended in ice-cold calcium incubation buffer (3.5 mM HEPES, pH 7.5, 10 mM CaCl$_2$, 0.1% Tween-20, 0.1% Triton X-100) and immediately placed into an ice-cold metal block in a 4 °C deli fridge to maintain the temperature. The samples were incubated for 30 min, and then 100 µl of 2× stop buffer (340 mM NaCl, 20 mM EDTA, 5 mM EGTA, 0.1% Tween-20, 0.1% Triton X-100, 25 µg ml$^{-1}$ RNase A (Thermo Fisher Scientific) and 0.05 ng per 100 µl of E. coli spike-in DNA (Epicypher, 18-1401)) was added, and the beads were mixed by flicking. Nuclei were incubated at 37 °C for 15 min with no shaking to allow for release of chromatin and digestion of RNA. Beads were placed onto a magnet, and the supernatant (200 µl) was collected. DNA was isolated using the Zymo ChIP DNA Clean & Concentrator kit (D5205) and eluted in 30 µl and frozen at −20 °C for library preparation.

**CUT&RUN-seq library preparation and sequencing.** Library preparation was performed using the NEBnext Ultra II DNA library kit (E7645L) with multiplexed adapters with minor modifications. CUT&RUN DNA underwent end repair and adapter ligation according to the manufacturer's protocol (1:15 adapter dilution was used). DNA was amplified using 16 PCR cycles with 10 s of extension time per cycle. Libraries were quantified using the Qubit fluorometer (Thermo Fisher Scientific) DNA high sensitivity kit, and the library size distribution was checked using the Tapestation DNA High Sensitivity ScreenTape (Agilent). Libraries were pooled at an equimolar concentration and sequenced on the Illumina NovaSeq 6000 sequencer by the NYU Genome Technology Center.

**CUT&RUN qPCR.** Ssoadvanced universal SYBR green master mix (1725270) was used according to the manufacturer's instructions. A master mix containing 2× ssoadvanced mastermix (10 µl per 20 µl reaction) and the desired primers (1 µl of 10 µM forward and reverse primer premixed per 20 µl reaction) and ultra-pure water were premixed to prepare a master mix. Once the master mix was dispensed into a 384-well plate (HSP3801), 1 µl of 1 ng µl$^{-1}$ CUT&RUN DNA library was dispensed into the desired wells. A QuantStudio5 real-time qPCR instrument was used according to the manufacturer's instructions. Data were analysed to examine enrichment over IgG controls. A list of primers used in this study is provided in Supplementary Data 7.

**CUT&RUN-seq data analysis.** Raw sequencing files were demultiplexed using bcl2fastq2 (Illumina, v.2.2.20). Between 20 and 100 million total reads were achieved for each replicate (average, 42.6 million). The samples were aligned to the hg19 or mm10 genome using bowtie2 (v.2.5.0)[66], with the following parameters: --local --very-sensitive-local --phred33 -I 10 -X 700 --dovetail --no-unal --no-mixed --no-discordant[63]. Low-quality reads were filtered out using Samtools (v.1.9) with a cut-off MAPQ score of 30, and only unique reads were retained for further processing[67]. Unique read files for each replicate/timepoint/antibody were merged and used for peak calling using MACS2 (v.3.0.0a6) with the callpeak function and the options -f BAMPE -q 0.05 --broad --broad-cutoff .05, using the corresponding IgG sample as the -c. For visualization, each sample was normalized by scaling the samples based on the E. coli spike-in DNA. Each sample was aligned to the E. coli genome (MG1655), and the uniquely aligned reads were counted. The number of E. coli reads for each replicate between timepoints was compared, with the sample with the lowest number of E. coli reads set at a scaling factor of 1×. The other samples were scaled down by a scaling factor that was computed by dividing the lowest number of E. coli reads by the sample number of E. coli reads. This was done separately for each antibody, as an internal normalization between timepoints across zeitgeber time[68]. The same was done comparing vehicle- and zolpidem-treated animals. Genome coverage tracks (bigwig files) were produced using the deepTools (v.3.5.1) bamCoverage function with the options --binSize 10 --smoothLength 30 --normalizeUsing None --scaleFactor # (derived from E. coli spike in) and using an ENCODE hg19 or mm10 blacklist file (https://doi.org/10.1038/s41598-019-45839-z, v2 for both) to discard regions with consistently non-specific signal[69]. Peak annotation and motif analysis of MACS2[70] called peaks were performed using HOMER (v.4.11)[71]. Heat maps were made using Deeptools (v.3.5.5) computeMatrix and plotHeatmap in reference-point mode, centred over TSSs or peak centres, using binSize 10 and --sortUsing mean, sorted in descending order. TSSs were downloaded from the UCSC (mm10/hg19) table browser using the canonically annotated transcript for each gene. Overlap of various peaks and TSSs was achieved using bedtools intersect (v.2.31)[72]. For generation of average plot profiles (±500 bp of TSS) and input for running JTK cycle on H3K4me3Q5his and H3K4me3Q5ser marks, Deeptools computeMatrix was run centred over TSSs with the parameters -a 500 -b 500 -binSize 100. The resulting matrix and coordinates files were merged in R using dplyr (v.1.1.4), and the average signal over the 1 kb window for each timepoint was computed for the plot profiles. For JTK cycle[37], the average signal over the same 1 kb window was computed for all individual biological replicates, and JTK cycle (v.3.1) was run with the options jtkdist(6,3), periods(2:6) and jtk.init(periods,4). Genes with $P_{adj} < 0.05$ were further analysed, with Z-scores being computed in R using tidyverse (v.2.0.0) and plotted as a heat map using the function heatmap.2 (gplots v.3.1.3.1).

## Sleep manipulation

Mice were individually habituated to locomotor activity monitoring cages and chambers (as described below in the 'Circadian locomotor activity' section) for 24 h, and then received either an injection of vehicle or zolpidem (10 mg per kg, i.p.) at 19:00 (the beginning of active phase, ZT12). Animals were placed back into the locomotor activity monitoring cages immediately and activity was measured for 8 h. Animals were immediately euthanized and the brains were collected and frozen for subsequent CUT&RUN-seq experiments.

## Circadian locomotor activity

To monitor locomotor activity across sleep–wake cycles, mice were individually placed into clean, transparent home cages with minimal bedding and access to food and water ad libitum. Home cages were placed into larger activity chambers with infrared beams to detect movement across 24 h (clear plexiglass 40 × 40 × 30 cm, Omnitech Electronics), starting at 19:00 (lights off). Activity was monitored through beam breaks, which was collected by Fusion Software (v.5.0) (Omnitech

Electronics) software and calculated into 4 h time bins. Animals were monitored 12 h after switching them from their normal light–dark cycle (lights on at 07:00, lights off at 19:00) to dark–dark for 48 h to assess their locomotor behaviour.

## Statistics and reproducibility

All in vitro and in cellulo western blotting and MS analyses were repeated independently at least three times with similar results. For western blot comparisons in brain examining rhythmic patterns of expression for H3Q5his, H3K4me3Q5his, HDC and NeuN (all normalized to total H3), nonlinear regression 'comparison of fits' analyses were performed between third-order polynomial, cubic trends (alternative hypothesis; that is, rhythmic) versus first-order polynomial, straight line trends (null hypothesis; non-rhythmic). Circadian blots were also assessed using one-way ANOVA with Tukey's multiple-comparison tests. For behavioural locomotor testing involving two viral treatments and multiple timepoints, repeated-measures two-way ANOVA was performed with subsequent Šidák's post hoc analyses for multiple comparisons, as well as a posteriori Student's $t$-tests (as indicated in the text). For biochemical quantifications of MLL–SETD1 activity (LC–MS/MS), two-way ANOVA was used with Šidák's multiple-comparison tests. For peptide IP/western blotting experiments with recombinant WDR5 (or MLL1), data were assessed using one-way ANOVA with Tukey's multiple-comparison tests. All animals used were included as separate $n$s (samples were not pooled). Significance was determined at $P < 0.05$. All data are represented as mean ± s.e.m. Statistical analyses were performed in GraphPad Prism 9.

## Inclusion and ethics statement

All collaborators associated with this work have fulfilled the criteria for authorship required by Nature Portfolio journals. To obtain authorship, their participation in this study was deemed to be essential for the design and implementation of the work presented. Roles and responsibilities were agreed upon among collaborators ahead or during the research.

## Reporting summary

Further information on research design is available in the Nature Portfolio Reporting Summary linked to this article.

## Data availability

The RNA-seq and CUT&RUN–seq data generated in this study have been deposited in the National Center for Biotechnology Information Gene Expression Omnibus (GEO) database under accession number GSE270434. All MS proteomics data have been deposited at the ProteomeXchange Consortium via the PRIDE partner repository (PXD053429 and PXD053788). The atomic coordinates and structure factors have been deposited at the PDB (8HMX). The data supporting findings of this study are available within the Article and its Supplementary Information. Related data are available from the corresponding author on reasonable request. No restrictions on data availability apply. Source data are provided with this paper.

## Code availability

Related code is available from the corresponding author on reasonable request.

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

**Acknowledgements** We thank the members of the I.M. and Y.D. laboratories for reading the manuscript. This work was partially supported by grants from the National Institutes of Health: R01 MH116900 (I.M.), R35 GM138386 (Y.D.), P30 CA008748 (Y.D.), P50 CA192937 (Y.D.), R35 GM150676 (Q.Z.), T32 DA007135 (R.M.B.), S10 OD030286 (S. Sidoli), P30 CA013330 (S. Sidoli), R37 GM086868 (T.W.M.), as well as funds from OSUCCC (Q.Z.), AFAR (Sagol Network GerOmics award; S. Sidoli), Deerfield (Xseed award; S. Sidoli), Relay Therapeutics (S. Sidoli), Merck (S. Sidoli) and the Einstein-Mount Sinai Diabetes Research Center (S. Sidoli). The Y.D. laboratory is also supported by the Josie Robertson Foundation, the Pershing Square Sohn Cancer Research Alliance, the Parker Institute for Cancer Immunotherapy, the STARR Cancer Alliance award and the Anna Fuller Trust. Moreover, the Y.D. laboratory is supported by W. H. Goodwin, A. Goodwin and the Commonwealth Foundation for Cancer Research and the Center for Experimental Therapeutics at MSKCC. This research was also supported by grants from the National Natural Science Foundation of China (T2488301, 92153302 to H.L.) and the National Key R&D Program of China (2021YFA1300103, 2020YFA0803303 to H.L.). The I.M. laboratory is also supported by funds from the Howard Hughes Medical Institute.

**Author contributions** I.M., Y.D. and H.L. conceived of the project with input from Q.Z., R.M.B., S.Z. and T.W.M.; Q.Z., B.H.W., D.A.V., S.Z., R.M.B., S. Sidoli, H.L., Y.D. and I.M. designed the experiments and interpreted the data. Q.Z., B.H.W., D.A.V., S.Z., R.M.B., R.E.T., S. Stransky, A.M.C., S.D., J.C.C., G.D.S., M.C., N.Z., J.W., S.L.F., L.K., H.W., B.Z., L.V., A.U., L.D., H.M., S. Sidoli, T.W.M., H.L., Y.D. and I.M. collected and analysed the data. B.H.W., A.R. and L.S. performed the sequencing-based bioinformatics with input from S.L.F.; Q.Z., B.H.W., D.A.V., S.Z., R.M.B., H.L., Y.D. and I.M. wrote the manuscript with input from all of the other authors.

**Competing interests** The authors declare no competing interests.

**Additional information**
**Correspondence and requests for materials** should be addressed to Haitao Li, Yael David or Ian Maze.

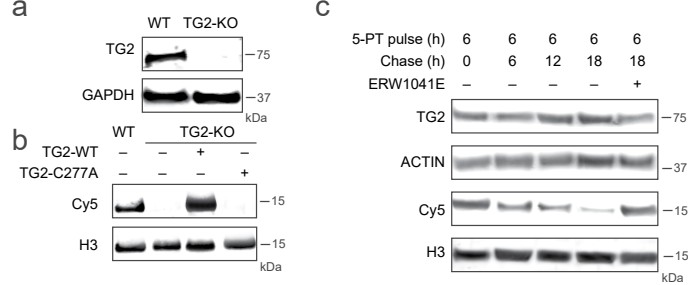

**Extended Data Fig. 1 | TG2 is a *bona fide writer* of histone H3 monoaminylations. a**, Western blotting validation of TG2 KO in HeLa cells following CRISPR-mediated deletion of TG2. GAPDH served as a loading control. **b**, 5-PT experiment in WT *vs.* TG2 KO HeLa −/+ 'rescue' with TG2-WT *vs.* TG2-C277A. **c**, HeLa cells treated with 5-PT and cultured for 6 h before media was changed to monoamine-free DMEM with or without TG2 inhibitor, ERW1041E. Cells were cultured for an additional 6, 12 or 18 h before being harvested. Histones were extracted, labelled by Cy5 and blotted with indicated antibodies. All WB experiments repeated 3X. Supplementary Fig. 1 = uncropped blots.

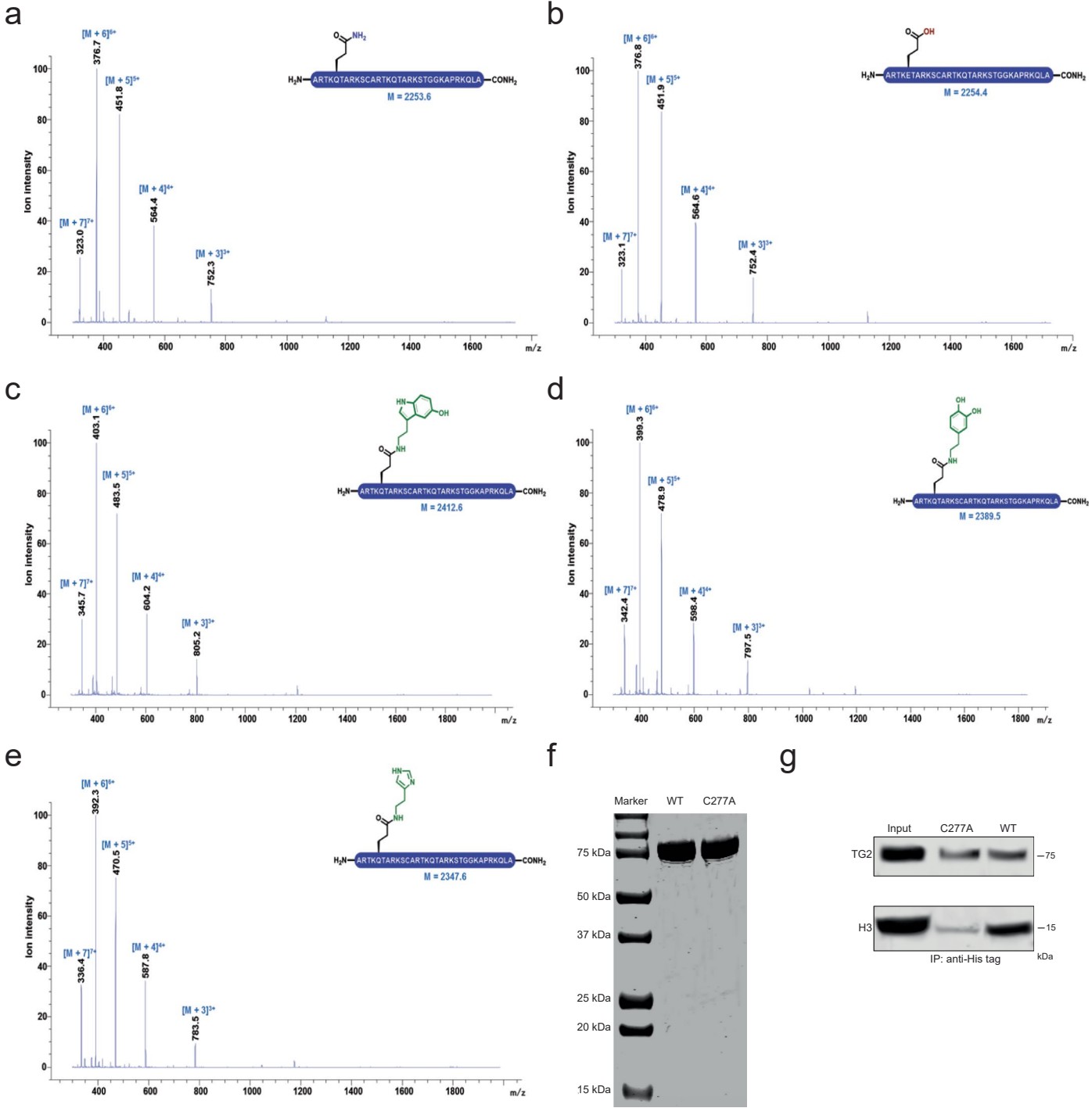

**Extended Data Fig. 2 | Structures/LC-MS analysis of monoaminylated peptides, TG2 purification and identification of a TG2-H3 thioester complex.** **a**–**e**, The synthetic H3 N-terminal peptides (containing 21 amino acid residues) used as standards and substrates include: **(a)** H3Q5, **(b)** H3Q5E, **(c)** H3Q5ser, **(d)** H3Q5dop and **(e)** H3Q5his. Side-chain protecting groups are omitted for clarity. **f**, Validation of recombinant TG2 purification via western blotting. Lanes from left to right = protein marker, WT TG2 and the TG2-C277A mutant, respectively. **g**, Identification of a TG2-H3 thioester complex from an in vitro biochemical reaction. All WB experiments repeated 3X. Supplementary Fig. 1 = uncropped blots.

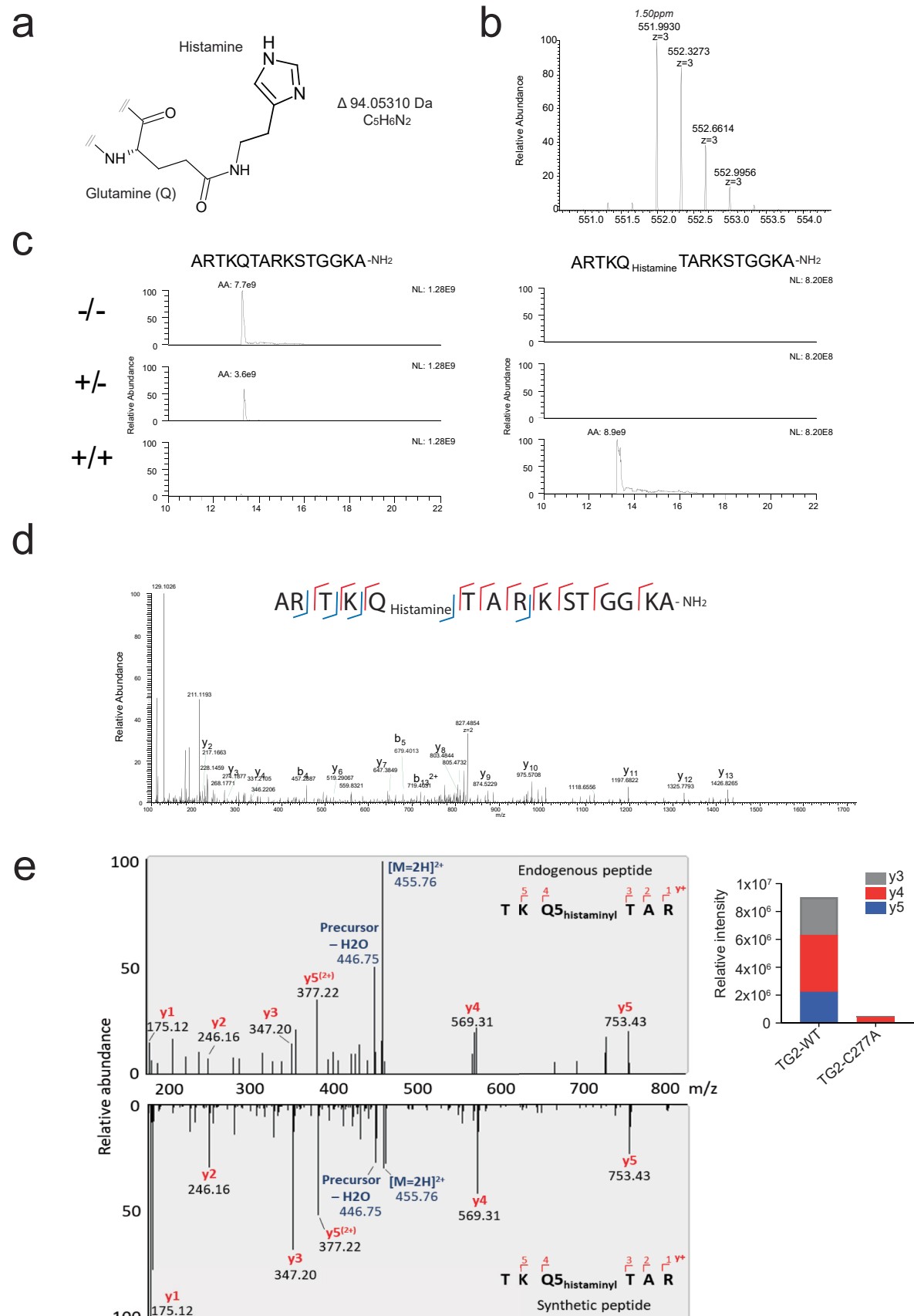

**Extended Data Fig. 3 |** See next page for caption.

**Extended Data Fig. 3 | LC-MS/MS validation of H3Q5his on peptides and in cells. a**, Proposed structure of histaminylated glutamine. **b**, High resolution/high mass accuracy mass spectrum of the triply charged histaminylated H3 tail peptide (ARTKQTARKSTGGKA-NH2; +histamine/+TG2 condition). The difference between measured and expected mass of the amidated peptide was 1.5 ppm. **c**, Extracted m/z (5 ppm) ion traces of the 2+, 3+ and 4+ amidated H3 tail peptide with (right panels) and without (left panels) histaminylated glutamine. Top, middle and bottom panels show signals measured under −/−, −/+ and +/+ (histamine/TG2) conditions, respectively. Integrated areas under curve are shown next to the peaks. Based on extracted signals, the reaction is close to complete. **d**, Tandem mass spectrum (35,000 resolution) of the doubly charged glutamine 5 histaminylated H3 tail peptide (+histamine/+TG2 condition). Selected fragment ions (y and b) are labelled. Lowest mass was m/z 100. Vertical lines in red and blue within the peptide sequence are used to show matched peptide fragment ions. **e**, LC-MS/MS identification of endogenous H3Q5his in HEK293T cells transfected with TG2-WT following histamine treatments. The cellular MS/MS spectra were aligned to that of a synthetic H3Q5his peptide, and y+ ions are annotated. Right: relative intensities of the most abundant y fragment ions from the H3Q5his peptide in 293T cells expressing TG2-WT *vs*. TG2-C277A. All experiments repeated 2X.

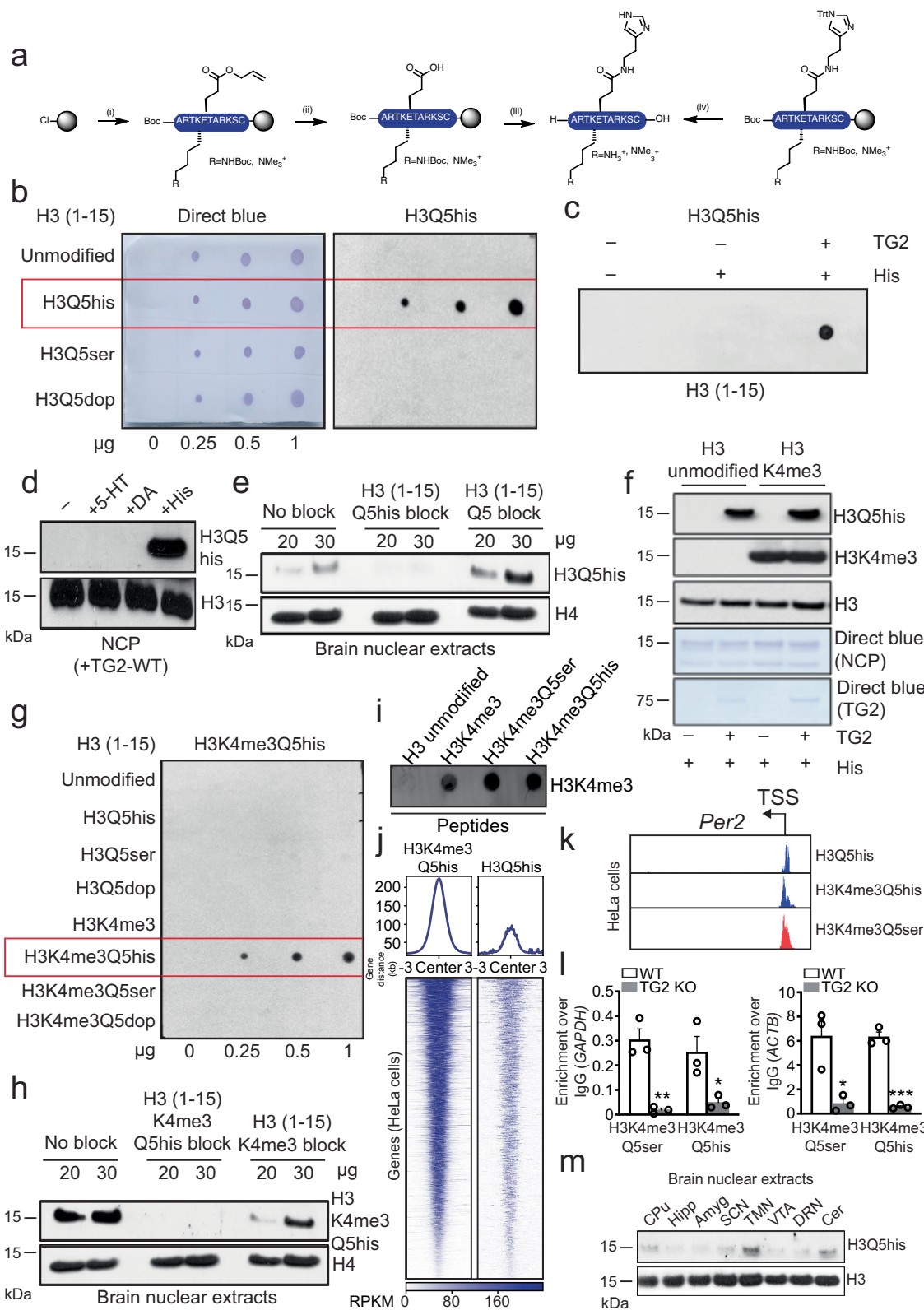

**Extended Data Fig. 4** | See next page for caption.

**Extended Data Fig. 4 | Generation and validation of H3Q5his antibodies.**
**a**, Synthesis of peptide antigens on 2-Cl trityl resin by (i) iterative Fmoc solid-phase peptide synthesis incorporating Fmoc–Glu(OAll)-OH at position 5 and either Fmoc–Lys(Boc)-OH or Fmoc–Lys(Me$_3$)-OH at position 4, (ii) followed by Pd(0) deallylation, (iii) coupling of Trt-protected histamine (iv) acidolytic cleavage from the resin and global deprotection. Side-chain protecting groups are omitted for clarity. **b**, Peptide dot blot titrations testing the H3Q5his antibody's specificity against unmodified $vs$. Q5his $vs$. Q5ser $vs$. Q5dop peptides. Direct blue staining was used to control for peptide loading. **c**, Peptide dot blots testing the H3Q5his antibody's reactivity −/+ histamine, −/+ TG2. **d**, Western blot analysis testing the H3Q5his antibody's reactivity/specificity on NCPs following TG2-mediated transamidation of histamine $vs$. serotonin $vs$. dopamine. Total H3 served as a loading control. **e**, Peptide competition (i.e., no block $vs$. unmodified H3 block $vs$. histaminyl blocks) western blotting analysis of lysates from mouse brain indicating the specificity of our H3Q5his antibody. **f**, Western blotting following TG2-mediated histaminylation on unmodified $vs$. H3K4me3 mononucleosomes revealed that TG2 can transamidate histamine to H3 in the context of adjacent H3K4me3. **g**, Peptide dot blot titrations testing the H3K4me3Q5his antibody's specificity against unmodified $vs$. Q5his $vs$. Q5ser $vs$. Q5dop $vs$. K4me3 $vs$. K4me3Q5his $vs$. K4me3Q5ser $vs$. K4me3Q5dop peptides. **h**, Peptide competition (i.e., no block $vs$. H3K4me3 block $vs$. H3K4me3Q5his blocks) western blotting analysis of lysates from mouse brain indicating the specificity of our H3K4me3Q5his antibody. **i**, H3 unmodified $vs$. H3K4me3 $vs$. H3K4me3Q5ser $vs$. H3K4me3Q5his peptide dot blot, followed by western blotting for H3K4me3, demonstrating that the H3K4me3 antibody used in CUT&RUN-seq experiment in brain recognizes H3K4me3 in the context of Q5ser and Q5his. **j**, Heatmap of H3K4me3Q5his and H3Q5his peak enrichment at genic loci in HeLa cells anchored on H3Q5his peak-enriched genes. **k**, Representative IGV browser tracks of overlapping H3Q5his $vs$. H3K4me3Q5his $vs$. H3K4me3Q5ser enrichment at the $Per2$ locus in HeLa cells. **l**, CUT&RUN-qPCRs ($n$ = 3 biological replicates) for H3K4me3Q5ser and H3K4me3Q5his at H3Q5his-enriched genic loci ($GAPDH$ and $ACTB$) in WT vs. TG2 KO HeLa cells, demonstrating loss of H3 monoaminylation signal in TG2 KO cells ($GAPDH$: *$p$ = 0.0362, $t_4$ = 3.101/ **$p$ = 0.0033, $t_4$ = 6.297; $ACTB$: *$p$ = 0.0179, $t_4$ = 3.875/***$p$ = 0.0002, $t_4$ = 13.78). **m**, Western blotting for H3Q5his across multiple brain regions in mouse reveals relative enrichment for the mark in TMN $vs$. other non-histaminergic monoaminergic and non-monoaminergic brain structures. Bar graphs presented as mean +/− SEM. Supplementary Fig. 1 = uncropped blots. Source data provided as a **Source Data file**.

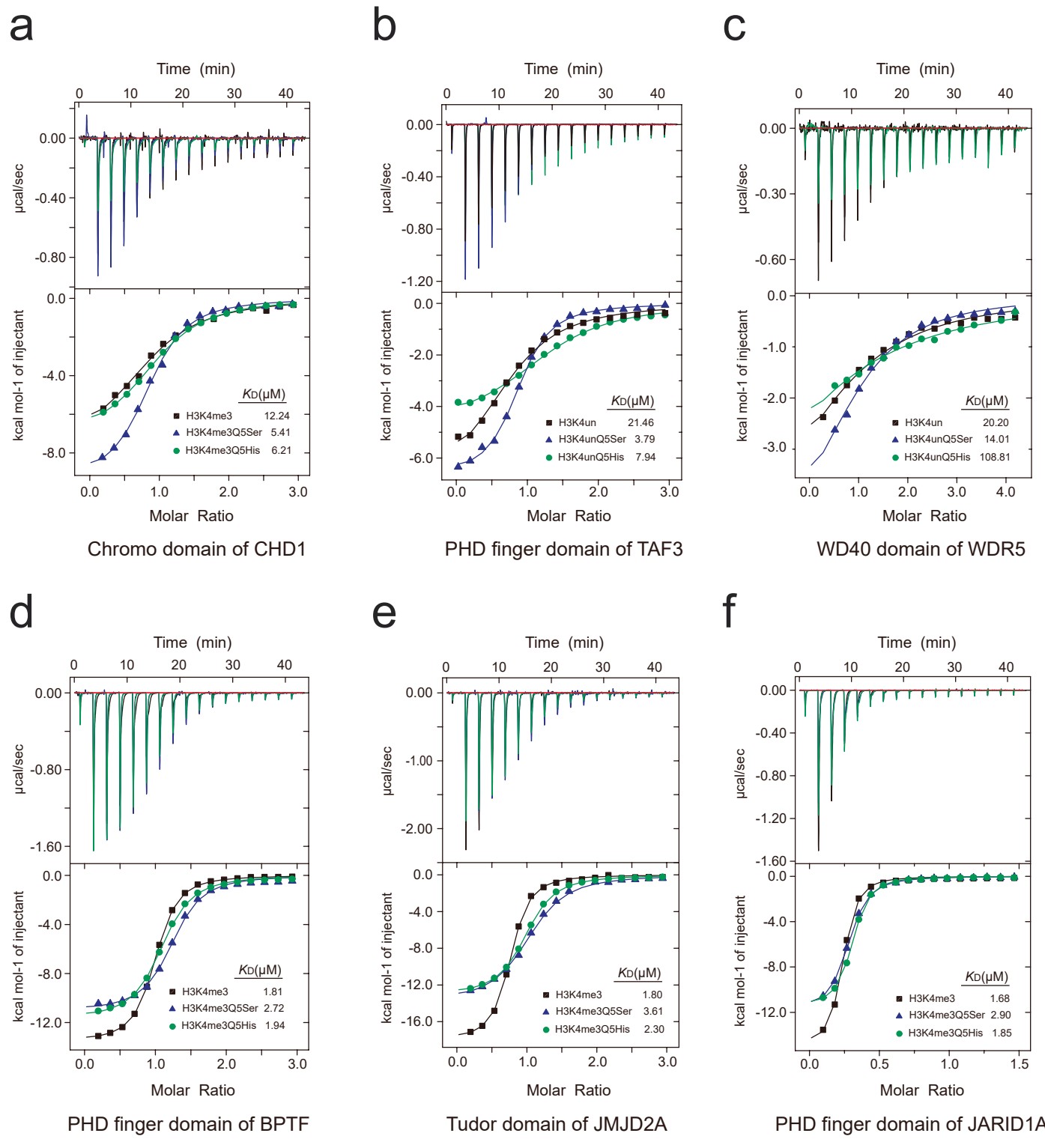

**Extended Data Fig. 5 | H3Q5his selectively antagonizes WDR5 binding to H3 vs. other H3 tail readers. a–f**, ITC assessments of **(a)** CHD1$_{Chromodomain}$, **(b)** TAF3$_{PHD}$, **(c)** WDR5$_{WD40}$, **(d)** BPTF$_{PHD}$, **(e)** JMJDA$_{Tudor}$ and **(f)** JARID1A binding to H3K4(K4me3)Q5unmod vs. H3K4(K4me3)Q5his vs. H3K4(K4me3)Q5ser peptides.

Note that titrations of Q5un/Q5ser/Q5his peptides were performed in the same batch, and the Q5un/Q5ser data have been published in earlier work[1]. The Q5un/Q5ser data are included here as controls for the Q5his data. Supplementary Table 1 = ITC statistics. All experiments repeated 2X.

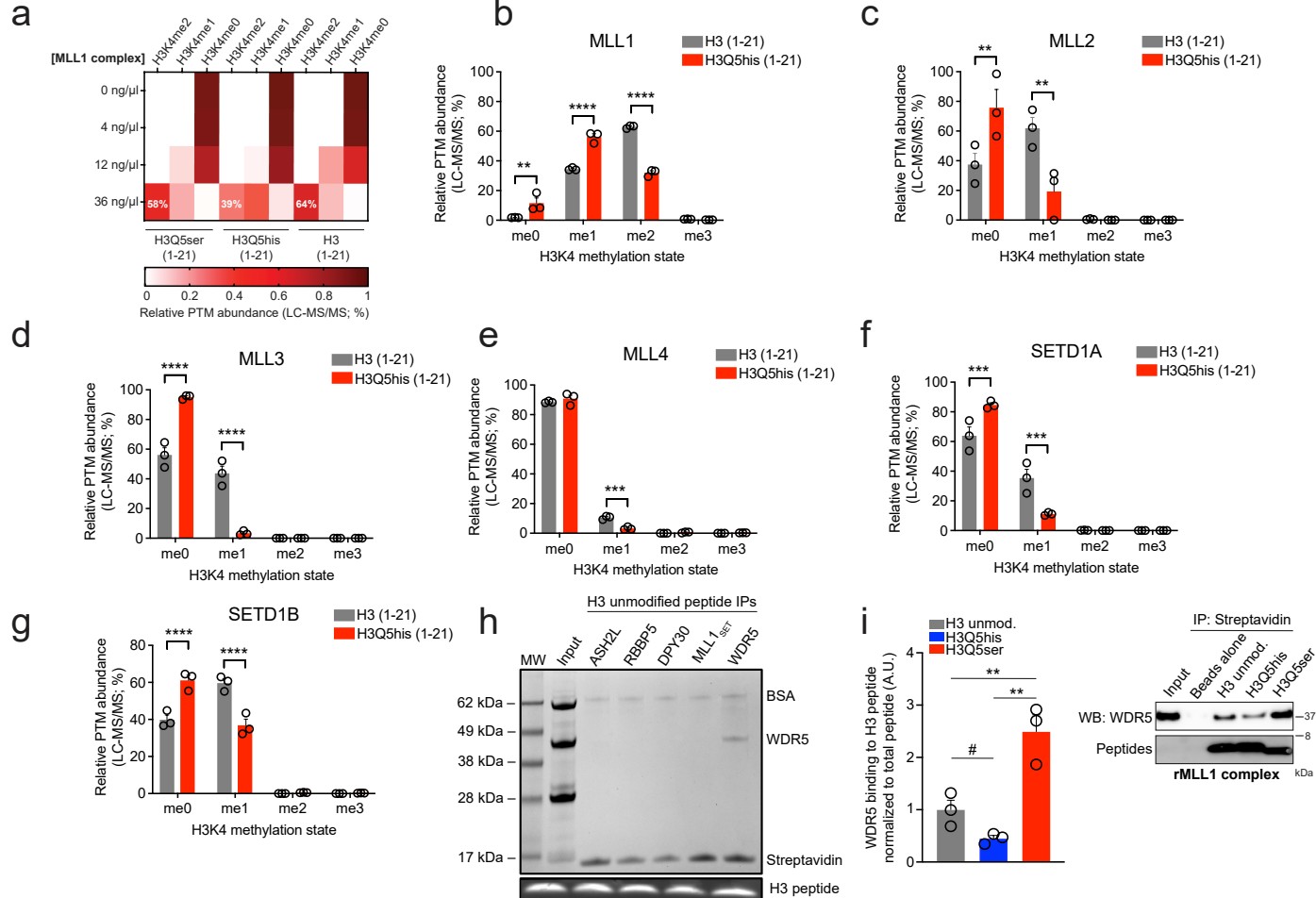

**Extended Data Fig. 6 | MLL and SETD1 enzymatic assay quantifications.**
**a**, LC-MS/MS quantification of H3K4 methylation states (H3K4me0 *vs.* H3K4me1 *vs.* H3K4me2; H3K4me3 signal was negligible and was thus omitted) on H3 (1-21) unmodified *vs.* H3Q5ser *vs.* H3Q5his peptides, titrating the concentration of MLL1 complex in the system. **b**–**g**, Enzymatic quantifications related to Fig. 2c: LC-MS/MS quantification of H3K4 methylation states (H3K4me0 *vs.* H3K4me1 *vs.* H3K4me2 *vs.* H3K4me3) on H3 (1-21) unmodified *vs.* H3Q5his peptides for **(b)** MLL1, **(c)** MLL2, **(d)** MLL3, **(e)** MLL4, **(f)** SETD1A and **(g)** SETD1B complexes ($n = 3$ per peptide/complex). Two-way ANOVA (main effects of interaction: MLL1 – $p < 0.0001$, $F_{3,16} = 105.5$; MLL2 – $p = 0.0002$, $F_{3,16} = 12.29$; MLL3 – $p < 0.0001$, $F_{3,16} = 86.82$; MLL4 – $p = 0.0003$, $F_{3,16} = 11.26$; SETD1A – $p < 0.0001$, $F_{3,16} = 18.86$; SETD1B – $p < 0.0001$, $F_{3,16} = 41.49$), Sidak's MC tests; significant *post hoc* comparisons are noted [MLL1: **$p = 0.0017$ (H3K4me0), ****$p < 0.0001$ (H3K4me1/2); MLL2: **$p = 0.0036$ (H3K4me0), **$p = 0.0014$ (H3K4me1); MLL3: ****$p < 0.0001$ (H3K4me0 and H3K4me1); MLL4: ***$p = 0.0001$ (H3K4me1);

SETD1A: ***$p = 0.0006$ (H3K4me0), ***$p = 0.0001$ (H3K4me1); SETD1B: ****$p < 0.0001$ (H3K4me0/1)]. **h**, H3 (1-21) unmodified peptide IPs against recombinant core members of the MLL1 complex (ASH2L, RBBP5, DPY30, MLL1$_{SET}$ and, WDR5) demonstrating that only WDR5 interacts with the unmodified H3 tail in monomeric form. **i**, H3 (1-21) unmodified *vs.* H3Q5his *vs.* H3Q5ser peptide IPs (Streptavidin) against the recombinant MLL1 complex, followed by western blotting for WDR5 ($n = 3$ per peptide). Streptavidin was used to visualize the peptides post-IP, which were used for normalization. One-way ANOVA ($p = 0.0014$, $F_{2,6} = 24.03$), Tukey's MC test: significant comparisons are noted (H3 unmodified *vs.* H3Q5ser, $p = 0.0065$; H3Q5his *vs.* H3Q5ser, $p = 0.0013$). #$p = 0.0462$ indicates a significant difference via an *a posteriori* unpaired Student's t-test ($t_4 = 2.854$). Data presented as mean +/– SEM. A.U., arbitrary units, normalized to respective controls (e.g., H3 unmodified peptide). Supplementary Fig. 1 = uncropped blots. Source data provided as a **Source Data file**.

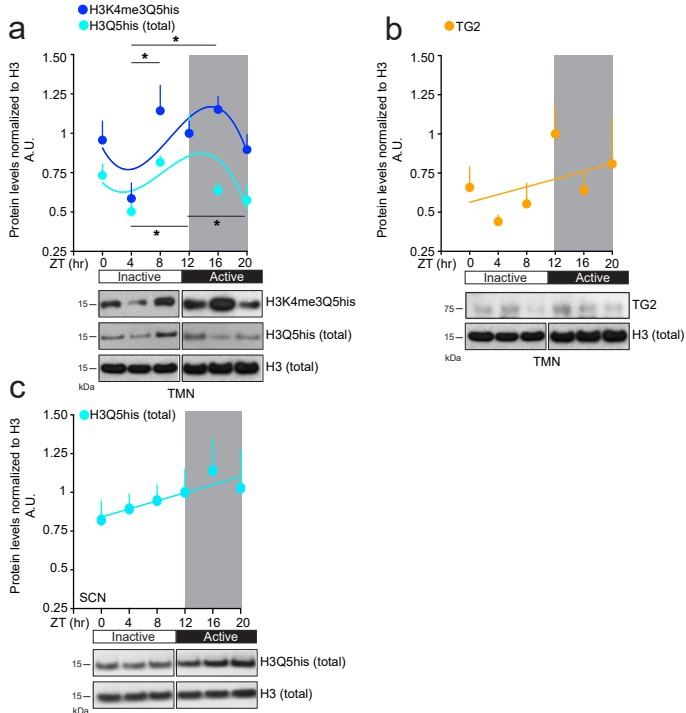

**Extended Data Fig. 7 | H3Q5his and H3K4me3Q5his display diurnal rhythmicity in TMN, but not SCN. a,c**, H3Q5his and H3K4me3Q5his in **(a)** TMN, but not **(c)** SCN (H3Q5his only), display a rhythmic pattern of expression across ZT in mice [TMN – H3K4me3Q5his: ZT0/ZT4/ZT8, $n$ = 9 biological replicates per time point; ZT12/ZT16/ZT20, $n$ = 10 biological replicates per time point; TMN – H3Q5his: ZT0/ZT4/ZT12/ZT20, $n$ = 9 biological replicates per time point; ZT8/ZT16, $n$ = 7 biological replicates per time point; SCN – H3Q5his: ZT0/ZT4/ZT8/ZT12/ZT16/ZT20, $n$ = 9 biological replicates per time point). Comparison of fits analysis between third order polynomial, cubic/ rhythmic trend (alternative hypothesis) *vs.* first order polynomial, straight line/linear trend (null hypothesis): TMN H3K4me3Q5his – $p$ = 0.0428, null hypothesis rejected; TMN H3Q5his – $p$ = 0.0135, null hypothesis rejected; SCN H3Q5his – $p$ = 0.8500, null hypothesis not rejected. Data additionally analysed via one-way ANOVA (TMN H3K4me3Q5his: $p$ = 0.0181, $F_{5,51}$ = 3.029; TMN H3Q5his: $p$ = 0.0066, $F_{5,40}$ = 3.796; SCN H3Q5his: $p$ = 0.8077, $F_{5,42}$ = 0.454) with Tukey's MC tests. Statistical differences between time points are noted [TMN H3K4me3Q5his: *$p$ = 0.0317 (ZT4 *vs.* ZT8), *$p$ = 0.0133 (ZT4 *vs.* ZT16); TMN H3Q5his: *$p$ = 0.0108 (ZT4 *vs.* ZT12), *$p$ = 0.0381 (ZT12 *vs.* ZT20)]. Data presented as mean +/– SEM. **b**, TG2 in TMN does not display rhythmic fluctuations in expression across ZT (ZT0/8/12/16, $n$ = 8 biological replicates per time point; ZT4/20, $n$ = 7 biological replicates per time point). Comparison of fits analysis between third order polynomial, cubic/rhythmic trend (alternative hypothesis) *vs.* first order polynomial, straight line/linear trend (null hypothesis): – $p$ = 0.5718, null hypothesis not rejected. Data additionally analysed via one-way ANOVA ($p$ = 0.2246). Supplementary Fig. 1 = uncropped blots. Source data provided as a **Source Data file**.

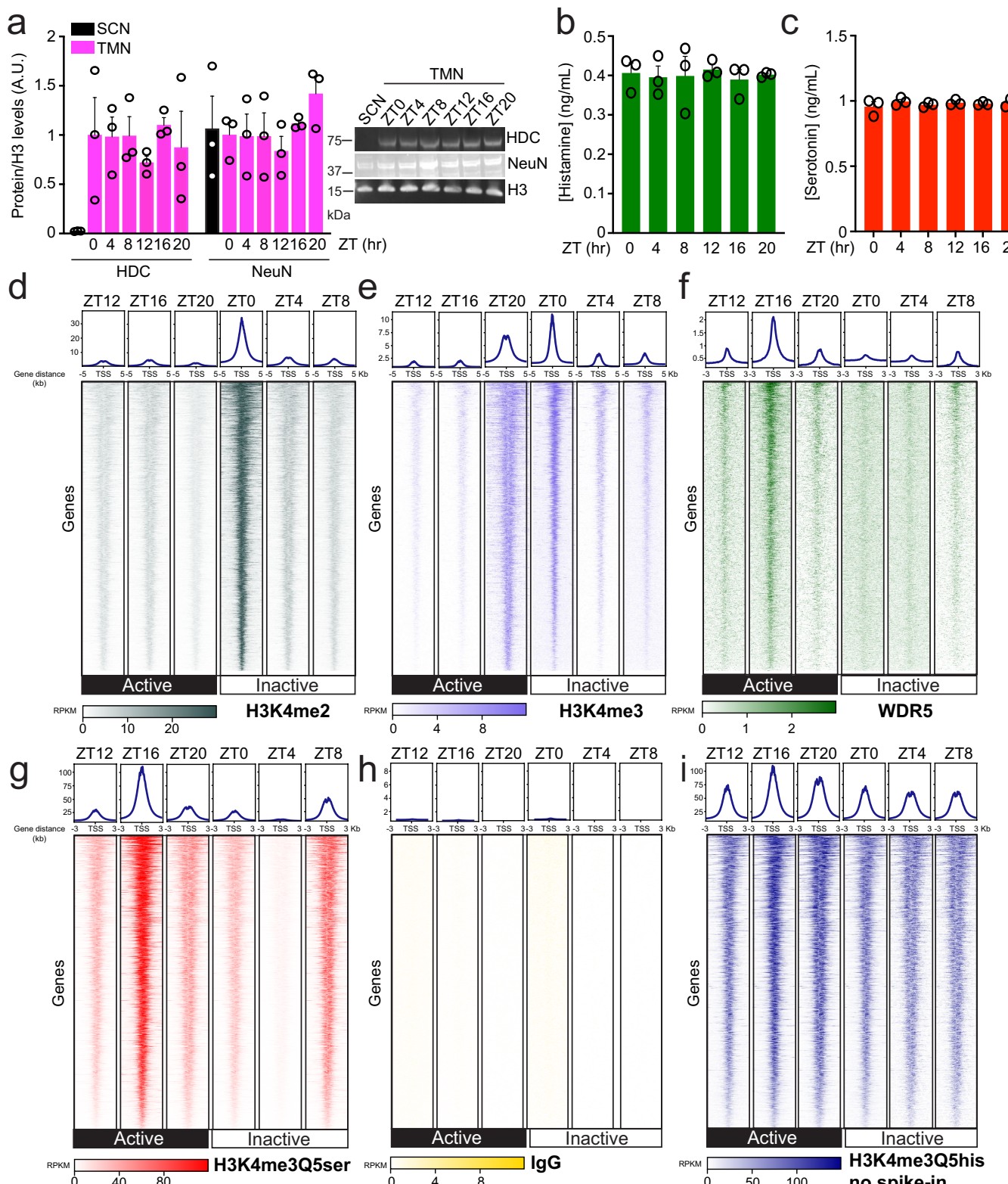

**Extended Data Fig. 8 | Dynamic epigenomic regulation of H3 monoaminylations *vs.* H3K4 methylation across ZT. a**, Neither Hdc nor NeuN levels display rhythmic regulation across ZT in TMN (*n* = 3 biological replicates/time point). Data analysed via one-way ANOVA (Hdc: *p* = 0.9132, $F_{5,12}$ = 0.2835; NeuN: *p* = 0.3223, $F_{5,12}$ = 1.313). SCN was used as a negative control to monitor the accuracy of dissections for TMN tissues. **b,c**, Neither **(b)** histamine nor **(c)** serotonin levels (total concentrations, not exclusively intracellular) in TMN display rhythmic regulation across the ZT in TMN (*n* = 3 biological replicates/time point). Data analysed via one-way ANOVA (histamine

ELISA: *p* = 0.9901, $F_{5,12}$ = 0.1007; serotonin ELISA: *p* = 0.7543, $F_{5,12}$ = 0.5236). **d–i**, Heatmaps displaying enrichment for **(d)** H3K4me2, **(e)** H3K4me3, **(f)** WDR5 and **(g)** H3K4me3Q5ser across ZT at genic loci in TMN. Data were normalized to both **(h)** IgG (the signal of which was not appreciable in TMN) and E. coli spike-in DNA; **(i)** note that without appropriate normalization to E. coli spike-in DNA, rhythmic patterns of H3K4me3Q5his are largely not observed. See Supplementary Tables 3 and 4 for rhythmic analyses of H3K4me3Q5his and H3K4me3Q5ser. Bar graphs presented as mean +/– SEM. Supplementary Fig. 1 = uncropped blots. Source data provided as a **Source Data file**.

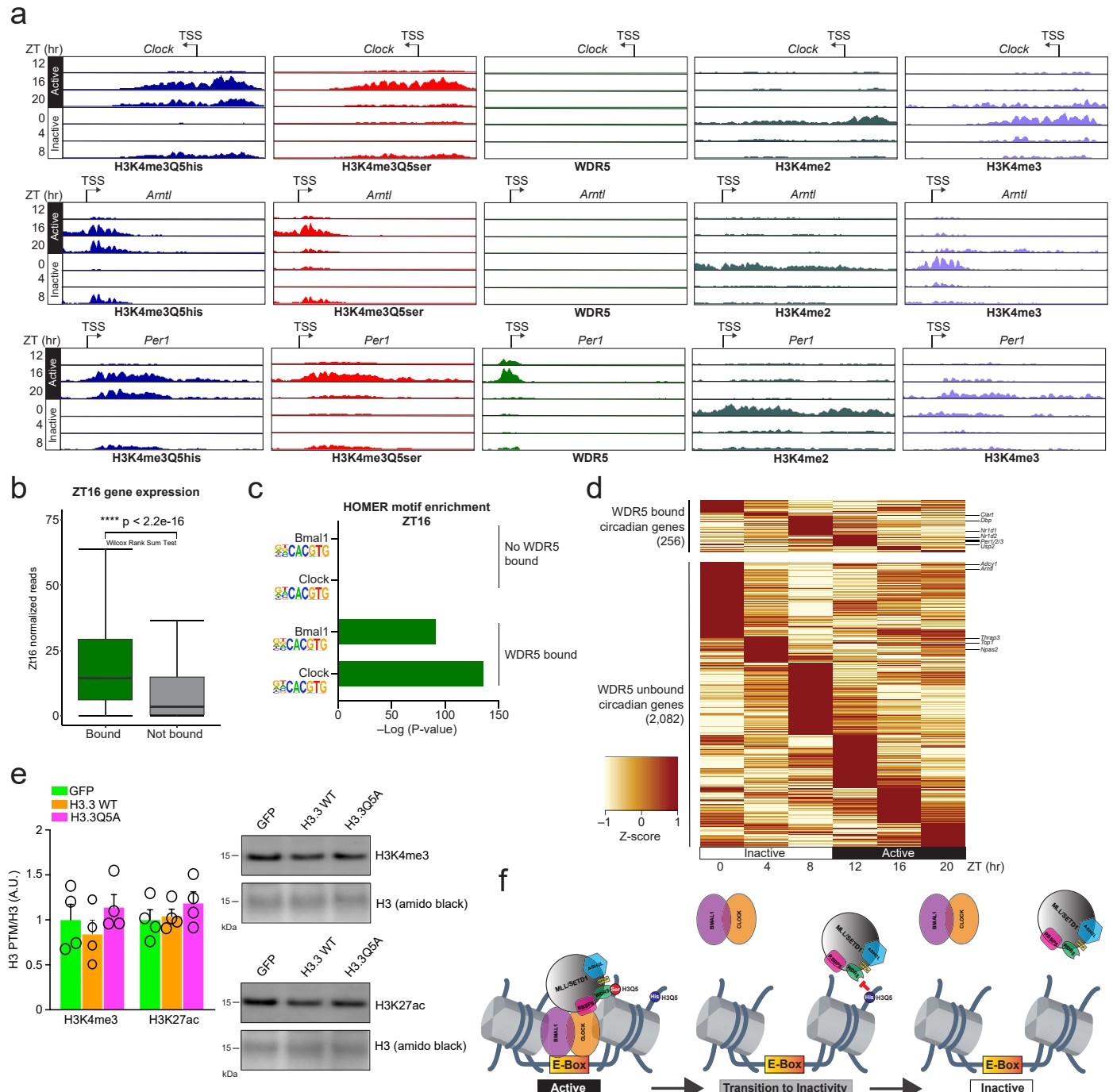

**Extended Data Fig. 9 | WDR5 enriches at Clock/Bmal1 motifs and correlates with rhythmic regulation of Clock/Bmal1 target genes. a**, Representative IGV browser tracks of H3K4me3Q5his, H3K4me3Q5ser, WDR5, H3K4me2 and H3K4me3 at Clock/Bmal1 targets (e.g., *Per1*) *vs.* non-Clock/Bmal1 target loci (e.g., *Clock* and *Arntl*), demonstrating selective enrichment/rhythmicity of WDR5 at Clock/Bmal1 target genes. **b**, Assessment of gene expression (normalized reads) for genes bound by WDR5 *vs.* those that are not bound by WDR5 at ZT16 (the height of WDR5's binding to chromatin across the ZT; Supplementary Table 5), indicating that WDR5 bound genes are more highly expressed *vs.* genes not bound by WDR5 (Wilcox Rank Sum Test, *p* < 2.2e-16). **c**, HOMER motif enrichment analysis of WDR5 bound *vs.* unbound loci, indicating that WDR5 significantly enriches that Clock/Bmal1 motifs (Benjamini-Hochberg, p < 0.05). **d**, Heatmap of gene expression (related to Fig. 3a and Supplementary Table 2) comparing WDR5 bound rhythmic genes *vs.* expression of genes not bound by WDR5 across the ZT, indicating that WDR5 bound genes are largely direct targets of Clock/Bmal1 (e.g., *Per1/2*, *Dbp*, *Nr1d1/2*, etc.), whereas circadian genes that are not bound by WDR5 are not direct targets of Clock/Bmal1

(e.g., *Arntl*, *Npas2*, etc.). **e**, Western blotting analyses of H3K4me3 and H3K27ac (normalized to H3/amido black staining) in TMN tissues virally transduced with AAV-GFP *vs.* H3.3 WT *vs.* H3.3Q5A. One-way ANOVAs were performed with no significant effects observed (*n* = 4 biological replicates/viral treatment; H3K4me3 – *p* = 0.4306, $F_{2,9}$ = 0.9266; H3K27ac – *p* = 0.4646, $F_{2,9}$ = 0.8357). **f**, Model: our genomics and biochemical data indicate that: (1) during periods of activity in TMN, WDR5 – likely in complex with MLL/SETD1 – is recruited to Clock/BMAL1 (E-Box) target genes (e.g., *Per1/2*, *Dbp*, *Nr1d1/2*, etc.), which are induced in their expression as a result of Clock/Bmal1 binding, and its binding is further stabilized to H3Q5ser; (2) during transitional periods towards inactivity, WDR5/MLL (SETD1) becomes destabilized at Clock/BMAL1 target genes owing, in part, to loss of H3Q5ser and stabilization of H3Q5his, the latter of which is antagonistic to WDR5 binding and H3K4 HMT activities; and (3) during periods of inactivity, H3Q5his is further reduced in its enrichment at Clock/Bmal1 targets, thereby allowing for spreading of H3K4 methylation, and the eventual re-recruitment of WDR5/MLL (SETD1) during transitions back into phases of activity. Supplementary Fig. 1 = uncropped blots. Source data provided as a **Source Data file**.

**Extended Data Table 1 | Data collection and refinement statistics for the WDR5$_{WD40}$-H3Q5his complex**

| | WDR5$_{WD40}$-H3Q5is |
|---|---|
| **Data collection** | |
| Space group | C2 |
| Cell dimensions | |
| $a$, $b$, $c$ (Å) | 113.2, 46.9, 65.3 |
| $\alpha$, $\beta$, $\gamma$ (°) | 90, 112.6, 90 |
| Resolution (Å) | 50-1.7 (1.73-1/70) * |
| $R_{sym}$ or $R_{merge}$ | 0.138 (0.531) |
| $I / \sigma I$ | 18.9 (4.84) |
| Completeness (%) | 99.2 (99.3) |
| Redundancy | 3.5 (3.8) |
| | |
| **Refinement** | |
| Resolution (Å) | 32.05-1/70 |
| No. reflections | 34866 |
| $R_{work}$ / $R_{free}$ | 0.168/0.193 |
| No. atoms | |
| Protein | 2368 |
| Peptide | 41 |
| Water | 393 |
| $B$-factors | |
| Protein | 12.7 |
| Peptide | 25.1 |
| Water | 26.2 |
| R.m.s. deviations | |
| Bond lengths (Å) | 0.006 |
| Bond angles (°) | 0.871 |

*Values in parentheses are for the highest-resolution shell.

# Reporting Summary

## Statistics

For all statistical analyses, confirm that the following items are present in the figure legend, table legend, main text, or Methods section.

| n/a | Confirmed | |
|---|---|---|
| ☐ | ☒ | The exact sample size (*n*) for each experimental group/condition, given as a discrete number and unit of measurement |
| ☐ | ☒ | A statement on whether measurements were taken from distinct samples or whether the same sample was measured repeatedly |
| ☐ | ☒ | The statistical test(s) used AND whether they are one- or two-sided *Only common tests should be described solely by name; describe more complex techniques in the Methods section.* |
| ☒ | ☐ | A description of all covariates tested |
| ☐ | ☒ | A description of any assumptions or corrections, such as tests of normality and adjustment for multiple comparisons |
| ☐ | ☒ | A full description of the statistical parameters including central tendency (e.g. means) or other basic estimates (e.g. regression coefficient) AND variation (e.g. standard deviation) or associated estimates of uncertainty (e.g. confidence intervals) |
| ☐ | ☒ | For null hypothesis testing, the test statistic (e.g. *F*, *t*, *r*) with confidence intervals, effect sizes, degrees of freedom and *P* value noted *Give P values as exact values whenever suitable.* |
| ☒ | ☐ | For Bayesian analysis, information on the choice of priors and Markov chain Monte Carlo settings |
| ☒ | ☐ | For hierarchical and complex designs, identification of the appropriate level for tests and full reporting of outcomes |
| ☒ | ☐ | Estimates of effect sizes (e.g. Cohen's *d*, Pearson's *r*), indicating how they were calculated |

*Our web collection on statistics for biologists contains articles on many of the points above.*

## Software and code

Policy information about availability of computer code

| Data collection | No custom algorithms were used. |
|---|---|
| Data analysis | RNA-seq data analysis<br>Raw sequencing reads were demultiplexed using bcl2fastq2 (Illumina, v2.20). Samples were aligned to the GRCm38 (UCSC build GCA_000001635.2) mouse genome using STAR (v2.7.11b) alignReads in mode BAM SortedByCoordinate. Gene counts were generated using htseq-count (HTSeq v2.0.5) with the following parameters: --format=bam --minaqual=10 --type=exon --idattr=gene_name --stranded=yes --mode=union using the Ensembl v93 annotation. Gene counts were normalized using DEseq2 (v1.44.0) prior to analysis using JTKcycle (v3.1) to identify cycling genes from the dataset using the parameters jtkdist (varying depending on replicates), periods(2:6) and jtk.init(periods,4). Genes with an ADJ.P < 0.05 were deemed significant and Z-scores were computed in R using tidyverse (v2.0.0). heatmap.2 in R was used to visualize the cycle genes (gplots version 3.1.3.1) across the zeitgeber. Cyclic genes were further assessed using ChEA in Enrichr (https://maayanlab.cloud/Enrichr/, v3.2), which infers transcription factor regulation from integration of previous genome-wide chromatin immunoprecipitation (ChIP) analyses. Further ontology was also conducted using Enrichr (v3.2). Odds ratios were calculated using the GeneOverlap R package (Version 1.26.0). DEseq2 (v1.44.0) was run to perform pairwise differential expression analyses between H3.3 and GFP viral treated samples (to ensure limited to no significant differences between control groups; number of DEGs between GFP vs. H3.3 WT at: ZT0 = 1, ZT4 = 0, ZT8 = 0, ZT12 = 40, ZT16 = 1, ZT20 = 1) before combining them together to run JTKcycle, as previously described. Differentially expressed (DE) genes were defined at FDR < .01. Overlap of JTK cycle genes with peak lists from CUT&RUN-seq (see below) was performed in R using dplyr (v1.1.4). Individual circadian rhythm controlling genes identified in Enrichr (v3.2) were highlighted on heatmaps manually.<br><br>CUT&RUN-seq data analysis<br>Raw sequencing files were demultiplexed using bcl2fastq2 (Illumina, v2.20). Between 20-100 million total reads were achieved for each |

replicate (average 42.6 million). Samples were aligned to the hg19 or mm10 genome using bowtie2 (2.5.0), with the following parameters: --local --very-sensitive-local --phred33 -I 10 -X 700 --dovetail --no-unal --no-mixed --no-discordant. Low quality reads were filtered out using Samtools (v1.9) with a cutoff MAPQ score of 30, and only unique reads were kept for further processing. Unique read files for each replicate/timepoint/antibody were merged and used for peak calling using MACS2 (v3.0.0a6) with the callpeaks function and the options -f BAMPE -q 0.05 --broad --broad-cutoff .05, using the corresponding IgG sample as the -c. For visualization, each sample was normalized by scaling the samples based off the E. coli spike-in DNA. Each sample was aligned to the E. coli genome (MG1655), and the uniquely aligned reads counted. The number of E. coli reads for each replicate between timepoints was compared, with the sample with the lowest number of E. coli reads set at a scaling factor of 1X. The other samples were scaled down by a scaling factor that was computed by dividing the lowest number of E. coli reads by the sample number of E. coli reads. This was done separately for each antibody, as an internal normalization between timepoints across zeitgeber time. The same was done comparing vehicle and zolpidem treated animals. Genome coverage tracks (bigwig files) were produced using deepTools (3.5.1) bamCoverage function with the options --binSize 10 --smoothLength 30 --normalizeUsing None --scaleFactor #(derived from E. coli spike in) and using an ENCODE hg19 or mm10 blacklist file (https://doi.org/10.1038/s41598-019-45839-z, hg19.v2/mm10v2 ) to discard regions with consistently non-specific signal. Peak annotation and motif analysis of MACS2 called peaks were performed using HOMER (v4.11). Heatmaps were made using Deeptools (v3.5.5) computeMatrix and plotHeatmaps in reference-point mode, centered over TSSs or peak centers, using binSize 10 and --sortUsing mean, sorted in descending order. TSSs were downloaded from the UCSC (mm10;gencode VM23/hg19;gencode V45lift37) table browser using the canonically annotated transcript for each gene. Overlap of various peaks and TSSs was achieved using bedtools intersect (v2.31). For generation of average plot profiles (+/-500 bp of TSS) and input for running JTK cycle on H3K4me3Q5his and H3K4me3Q5ser marks, Deeptools computeMatrix was run centered over TSSs with the parameters -a 500 -b 500 -binSize 100. The resulting matrix and coordinates files were merged in R using dplyr (v1.1.4), and the average signal over the 1kb window for each timepoint was computed for the plot profiles. For JTK cycle, the average signal over the same 1kb window was computed for all individual biological replicates, and JTK cycle (v3.1) was run with the options jtkdist(6,3), periods(2:6), and jtk.init(periods,4). Genes with a ADJ.P < 0.05 were further analyzed, with Z-scores being computed in R using tidyverse (v2.0.0) and plotted as a heatmap using the function heatmap.2 (gplots version 3.1.3.1).

For manuscripts utilizing custom algorithms or software that are central to the research but not yet described in published literature, software must be made available to editors and reviewers. We strongly encourage code deposition in a community repository (e.g. GitHub). See the Nature Portfolio guidelines for submitting code & software for further information.

## Data

Policy information about availability of data

All manuscripts must include a data availability statement. This statement should provide the following information, where applicable:

- Accession codes, unique identifiers, or web links for publicly available datasets
- A description of any restrictions on data availability
- For clinical datasets or third party data, please ensure that the statement adheres to our policy

The RNA-seq data and CUT&RUN-seq data generated in this study have been deposited in the National Center for Biotechnology Information Gene Expression Omnibus (GEO) database under accession number GSE270434. Raw files for the H3Q5his mass spectrometry proteomics data in 293T cells, as well as the mass spectrometry proteomics data have been deposited to the ProteomeXchange Consortium via the PRIDE partner repository with the dataset identifiers PXD053429 and PXD053788. The atomic coordinates and structure factors have been deposited in the Protein Data Bank (PDB) under PDB ID code 8HMX. We declare that the data supporting findings for this study are available within the article and Supplementary Information. Related data are available from the corresponding author upon reasonable request. No restrictions on data availability apply.

## Human research participants

Policy information about studies involving human research participants and Sex and Gender in Research.

| | |
|---|---|
| Reporting on sex and gender | N/A |
| Population characteristics | N/A |
| Recruitment | N/A |
| Ethics oversight | N/A |

Note that full information on the approval of the study protocol must also be provided in the manuscript.

# Field-specific reporting

Please select the one below that is the best fit for your research. If you are not sure, read the appropriate sections before making your selection.

☒ Life sciences  ☐ Behavioural & social sciences  ☐ Ecological, evolutionary & environmental sciences

For a reference copy of the document with all sections, see nature.com/documents/nr-reporting-summary-flat.pdf

# Life sciences study design

All studies must disclose on these points even when the disclosure is negative.

| | |
|---|---|
| Sample size | Adequate sample sizes are generally determined based upon inter-sample variability. Throughout the manuscript, we determined the significance of results based upon a general confidence interval of 95%. We do not include specific justifications of sample size within the methods (e.g., power analyses), as sample sizes were based on extensive laboratory experience with these endpoints. The sample sizes chosen are consistent with those used by others in the field to achieve statistically significant results comparing stressed vs. control animals. |
| Data exclusions | No data exclusions. |
| Replication | All biological endpoints were reliably reproduced using numerous biological (>3 for all experiments in which statistics were employed) and technical replicates for each experiment. All novel reagents used in this study were extensively validated, as demonstrated in the manuscript submission. |
| Randomization | Where appropriate, animals were randomly assigned to groups (segregated by viral treatments, or ZT). Tissue samples were not pooled from multiple animals in these studies for western-blotting and RNA-seq experiments (i.e., each n represents a discrete data point). For CUT&RUN, each replicate per antibody was pooled from punches from 3 animals (per n) for initial processing, with 3 independent biological replicates conducted (n=3) per antibody. |
| Blinding | For all viral experiments (RNA-seq, western blotting and behavior), investigators were blinded to conditions such viral treatment etc. prior to analysis. |

# Reporting for specific materials, systems and methods

We require information from authors about some types of materials, experimental systems and methods used in many studies. Here, indicate whether each material, system or method listed is relevant to your study. If you are not sure if a list item applies to your research, read the appropriate section before selecting a response.

## Materials & experimental systems

| n/a | Involved in the study |
|---|---|
| ☐ | ☒ Antibodies |
| ☐ | ☒ Eukaryotic cell lines |
| ☒ | ☐ Palaeontology and archaeology |
| ☐ | ☒ Animals and other organisms |
| ☒ | ☐ Clinical data |
| ☒ | ☐ Dual use research of concern |

## Methods

| n/a | Involved in the study |
|---|---|
| ☐ | ☒ ChIP-seq |
| ☒ | ☐ Flow cytometry |
| ☒ | ☐ MRI-based neuroimaging |

## Antibodies

| | |
|---|---|
| Antibodies used | Primary antibodies used in this study: <br> Rabbit Anti-TGM2 1:500 (brain) Abcam <br> CUB 7402 <br> Rabbit Anti-GAPDH 1:1000 (Western) Thermo-Fisher <br> PA1-16777 <br> 1: 1000 (in vitro and in cellulo) CST <br> (3557S) <br> Chicken Anti-H3 1: 1000 Abcam <br> (ab134198) <br> Mouse Anti-H3 1: 1000 Abcam <br> (ab10799) <br> Mouse Anti-Actin 1: 1000 CST <br> (3700S) <br> Rabbit Anti-H3Q5ser 1: 1000 Millipore <br> (ABE1791) <br> Rabbit Anti-H3Q5dop 1: 1000 Millipore <br> (ABE2588) <br> Rabbit Anti-H3Q5his 1: 200 (brain) <br> 1:1000 (in vitro and in cellulo) Millipore <br> (ABE2578) <br> Rabbit Anti-H3K4me3Q5ser 1:50 (CUT&RUN), 1:500 (Western) Millipore <br> (ABE2605) <br> Rabbit Anti-H3K4me3Q5his 1:50 (CUT&RUN), 1:500 (Western) Millipore <br> (ABE2570) <br> Rabbit Anti-H3K4me3 1:1000 Abcam <br> (ab8580) |

Rabbit Anti-H3K4me3 1:50 (CUT&RUN), 1:500 (Western) Epicypher
(13-0041)
Rabbit Anti-H3K4me2 1:1000 Abcam
(ab7766)
Rabbit Anti-H3K4me2 1:50 (CUT&RUN) Active Motif
(39141)
Rabbit Anti-H3K27ac 1:1000 Active Motif
(AB_2614979)
Rabbit Anti-H3 1:50000 (brain) Abcam
(ab1791)
Rabbit Anti-WDR5 1:50 (CUT&RUN) CST
(13105)
Rabbit Anti-WDR5 1:1000 Abcam
(ab307664)
Mouse Anti-FLAG M2 1:1000 Millipore Sigma
(F1804)
Chicken Anti-6XHIS 1:1000 Invitrogen
(PA19531)
Rabbit Anti-HDC 1:1000 ARP
(03–16045)
Rabbit Anti-NeuN 1:1000 Millipore Sigma
(MAB377)

Donkey Anti-Chicken IRDye 800CW 1: 15000 (Li-Cor 926-32218)
Goat anti-Chicken IgY (H+L) Secondary Antibody, Alexa Fluor™ 488, 1:500 Invitrogen (for H3 peptide quantification) (Thermo Fisher A-11039)
Goat Anti-Mouse IRDye 680RD 1: 15000 (Li-Cor 926-68070)
Goat Anti-Mouse IRDye 800CW 1: 15000 (Li-Cor 926-32210)
Goat Anti-Rabbit IRDye 800CW 1: 15000 (Li-Cor 926-32211)
Goat Anti-Rabbit IRDye 680RD 1: 15000 (Li-Cor 926-68071)
Goat Anti-Rabbit Horseradish Peroxidase 1:10000
1:50000 (for anti-H3 antibody) (BioRad 1706515)
Sheep Anti-Mouse Horseradish Peroxidase 1:5000 (Cytiva RPN4201)
Donkey Anti-Rabbit Horseradish Peroxidase 1:5000 ( Cytiva NA934)

| Validation | All antibodies used in this study (all of which have been commercially validated (see manufacturers website) or described in previous publications) were validated in cells/tissues via immunoblotting, IPs or ICC/IHC/IF prior to experimentation. Validation of all primary antibodies was additionally provided according to the manufacturer`s website/datasheet data. All information regarding validation of histone monoaminylation primary antibodies is included in the manuscript and in prior publications (cited in the manuscript). |
|---|---|

## Eukaryotic cell lines

Policy information about cell lines and Sex and Gender in Research

| Cell line source(s) | HeLa (CRM-CCL-2) and HEK293T (CRL-3216) cell lines were obtained from the American Type Culture Collection (ATCC). |
|---|---|
| Authentication | Human tissue culture cell lines (HeLa, HEK293T) were imaged for appropriate morphology. |
| Mycoplasma contamination | We can confirm that all cell lines tested negative for mycoplasma contamination. |
| Commonly misidentified lines (See ICLAC register) | No commonly misidentified cell lines were used. |

## Animals and other research organisms

Policy information about studies involving animals; ARRIVE guidelines recommended for reporting animal research, and Sex and Gender in Research

| Laboratory animals | Mice (C57BL/6J) were purchased from The Jackson Laboratory. Animals were group housed (2-5 per cage) on a 12-hour light/dark cycle (lights on from 7:00 A.M. to 7:00 P.M.) at constant temperature (23ºC) and with controlled humidity (50%) and ad libitum access to food and water. All mice used for the experiments in this manuscript were aged 8-14 weeks old. All animal protocols were approved by the IACUC at the Icahn School of Medicine at Mount Sinai (ISMMS). |
|---|---|
| Wild animals | No wild animals were used in this study. |
| Reporting on sex | Both male and female mice were used in this study. |
| Field-collected samples | No field collected samples were used in this study. |
| Ethics oversight | All animal protocols were approved by the IACUC at both the Icahn School of Medicine at Mount Sinai (ISMMS). |

Note that full information on the approval of the study protocol must also be provided in the manuscript.

# ChIP-seq

## Data deposition

☒ Confirm that both raw and final processed data have been deposited in a public database such as [GEO](GEO).

☒ Confirm that you have deposited or provided access to graph files (e.g. BED files) for the called peaks.

**Data access links**
*May remain private before publication.*

The CUT&RUN-seq data generated in this study have been deposited in the National Center for Biotechnology Information Gene Expression Omnibus (GEO) database under accession number GSE270434. This includes the raw fastq files for each experiment, as well as bigwigs made from merged BAM files (merging the 3 biological replicates). These bigwigs were scaled according to an E. coli spike-in DNA for each CUT&RUN reaction, with scaling happening across time-points or treatments for the same antibody. See methods for details. MACS2 peak files (.broadPeak and .narrowPeak) are available in the GEO as well for certain time points/antibodies.

**Files in database submission**

processed data file  processed data file  raw file raw file raw file raw file
0_K2.scaled.bs10.bw K4me2_ZT0_IgG.BROAD.01_peaks.broadPeak 0_K2_1_S11_L001_R1_001.fastq.gz
0_K2_1_S11_L001_R2_001.fastq.gz 0_K2_1_S11_L002_R1_001.fastq.gz 0_K2_1_S11_L002_R2_001.fastq.gz
0_K2.scaled.bs10.bw K4me2_ZT0_IgG.BROAD.01_peaks.broadPeak 0_K2_2_S12_L001_R1_001.fastq.gz
0_K2_2_S12_L001_R2_001.fastq.gz 0_K2_2_S12_L002_R1_001.fastq.gz 0_K2_2_S12_L002_R2_001.fastq.gz
0_K2.scaled.bs10.bw K4me2_ZT0_IgG.BROAD.01_peaks.broadPeak 0_K2_3_S13_L001_R1_001.fastq.gz
0_K2_3_S13_L001_R2_001.fastq.gz 0_K2_3_S13_L002_R1_001.fastq.gz 0_K2_3_S13_L002_R2_001.fastq.gz
4_K2.scaled.bs10.bw  4_K2_1_S17_L001_R1_001.fastq.gz 4_K2_1_S17_L001_R2_001.fastq.gz
4_K2_1_S17_L002_R1_001.fastq.gz 4_K2_1_S17_L002_R2_001.fastq.gz
4_K2.scaled.bs10.bw  4_K2_2_S18_L001_R1_001.fastq.gz 4_K2_2_S18_L001_R2_001.fastq.gz
4_K2_2_S18_L002_R1_001.fastq.gz 4_K2_2_S18_L002_R2_001.fastq.gz
4_K2.scaled.bs10.bw  4_K2_3_S19_L001_R1_001.fastq.gz 4_K2_3_S19_L001_R2_001.fastq.gz
4_K2_3_S19_L002_R1_001.fastq.gz 4_K2_3_S19_L002_R2_001.fastq.gz
8_K2.scaled.bs10.bw  8_K2_1_S23_L001_R1_001.fastq.gz 8_K2_1_S23_L001_R2_001.fastq.gz
8_K2_1_S23_L002_R1_001.fastq.gz 8_K2_1_S23_L002_R2_001.fastq.gz
8_K2.scaled.bs10.bw  8_K2_2_S24_L001_R1_001.fastq.gz 8_K2_2_S24_L001_R2_001.fastq.gz
8_K2_2_S24_L002_R1_001.fastq.gz 8_K2_2_S24_L002_R2_001.fastq.gz
8_K2.scaled.bs10.bw  8_K2_3_S25_L001_R1_001.fastq.gz 8_K2_3_S25_L001_R2_001.fastq.gz
8_K2_3_S25_L002_R1_001.fastq.gz 8_K2_3_S25_L002_R2_001.fastq.gz
12_K2.scaled.bs10.bw  12_K2_1_S29_L001_R1_001.fastq.gz 12_K2_1_S29_L001_R2_001.fastq.gz
12_K2_1_S29_L002_R1_001.fastq.gz 12_K2_1_S29_L002_R2_001.fastq.gz
12_K2.scaled.bs10.bw  12_K2_2_S30_L001_R1_001.fastq.gz 12_K2_2_S30_L001_R2_001.fastq.gz
12_K2_2_S30_L002_R1_001.fastq.gz 12_K2_2_S30_L002_R2_001.fastq.gz
12_K2.scaled.bs10.bw  12_K2_3_S31_L001_R1_001.fastq.gz 12_K2_3_S31_L001_R2_001.fastq.gz
12_K2_3_S31_L002_R1_001.fastq.gz 12_K2_3_S31_L002_R2_001.fastq.gz
16_K2.scaled.bs10.bw  16_K2_1_S35_L001_R1_001.fastq.gz 16_K2_1_S35_L001_R2_001.fastq.gz
16_K2_1_S35_L002_R1_001.fastq.gz 16_K2_1_S35_L002_R2_001.fastq.gz
16_K2.scaled.bs10.bw  16_K2_2_S36_L001_R1_001.fastq.gz 16_K2_2_S36_L001_R2_001.fastq.gz
16_K2_2_S36_L002_R1_001.fastq.gz 16_K2_2_S36_L002_R2_001.fastq.gz
16_K2.scaled.bs10.bw  16_K2_3_S37_L001_R1_001.fastq.gz 16_K2_3_S37_L001_R2_001.fastq.gz
16_K2_3_S37_L002_R1_001.fastq.gz 16_K2_3_S37_L002_R2_001.fastq.gz
20_K2.scaled.bs10.bw  20_K2_1_S41_L001_R1_001.fastq.gz 20_K2_1_S41_L001_R2_001.fastq.gz
20_K2_1_S41_L002_R1_001.fastq.gz 20_K2_1_S41_L002_R2_001.fastq.gz
20_K2.scaled.bs10.bw  20_K2_2_S42_L001_R1_001.fastq.gz 20_K2_2_S42_L001_R2_001.fastq.gz
20_K2_2_S42_L002_R1_001.fastq.gz 20_K2_2_S42_L002_R2_001.fastq.gz
20_K2.scaled.bs10.bw  20_K2_3_S43_L001_R1_001.fastq.gz 20_K2_3_S43_L001_R2_001.fastq.gz
20_K2_3_S43_L002_R1_001.fastq.gz 20_K2_3_S43_L002_R2_001.fastq.gz
0_K3.scaled.bs10.bw K4me3_ZT0_IgG.BROAD.05_peaks.broadPeak 0_K3_1_S14_L001_R1_001.fastq.gz
0_K3_1_S14_L001_R2_001.fastq.gz 0_K3_1_S14_L002_R1_001.fastq.gz 0_K3_1_S14_L002_R2_001.fastq.gz
0_K3.scaled.bs10.bw K4me3_ZT0_IgG.BROAD.05_peaks.broadPeak 0_K3_2_S15_L001_R1_001.fastq.gz
0_K3_2_S15_L001_R2_001.fastq.gz 0_K3_2_S15_L002_R1_001.fastq.gz 0_K3_2_S15_L002_R2_001.fastq.gz
0_K3.scaled.bs10.bw K4me3_ZT0_IgG.BROAD.05_peaks.broadPeak 0_K3_3_S16_L001_R1_001.fastq.gz
0_K3_3_S16_L001_R2_001.fastq.gz 0_K3_3_S16_L002_R1_001.fastq.gz 0_K3_3_S16_L002_R2_001.fastq.gz
4_K3.scaled.bs10.bw  4_K3_1_S20_L001_R1_001.fastq.gz 4_K3_1_S20_L001_R2_001.fastq.gz
4_K3_1_S20_L002_R1_001.fastq.gz 4_K3_1_S20_L002_R2_001.fastq.gz
4_K3.scaled.bs10.bw  4_K3_2_S21_L001_R1_001.fastq.gz 4_K3_2_S21_L001_R2_001.fastq.gz
4_K3_2_S21_L002_R1_001.fastq.gz 4_K3_2_S21_L002_R2_001.fastq.gz
4_K3.scaled.bs10.bw  4_K3_3_S22_L001_R1_001.fastq.gz 4_K3_3_S22_L001_R2_001.fastq.gz
4_K3_3_S22_L002_R1_001.fastq.gz 4_K3_3_S22_L002_R2_001.fastq.gz
8_K3.scaled.bs10.bw  8_K3_1_S26_L001_R1_001.fastq.gz 8_K3_1_S26_L001_R2_001.fastq.gz
8_K3_1_S26_L002_R1_001.fastq.gz 8_K3_1_S26_L002_R2_001.fastq.gz
8_K3.scaled.bs10.bw  8_K3_2_S27_L001_R1_001.fastq.gz 8_K3_2_S27_L001_R2_001.fastq.gz
8_K3_2_S27_L002_R1_001.fastq.gz 8_K3_2_S27_L002_R2_001.fastq.gz
8_K3.scaled.bs10.bw  8_K3_3_S28_L001_R1_001.fastq.gz 8_K3_3_S28_L001_R2_001.fastq.gz
8_K3_3_S28_L002_R1_001.fastq.gz 8_K3_3_S28_L002_R2_001.fastq.gz
12_K3.scaled.bs10.bw  12_K3_1_S32_L001_R1_001.fastq.gz 12_K3_1_S32_L001_R2_001.fastq.gz
12_K3_1_S32_L002_R1_001.fastq.gz 12_K3_1_S32_L002_R2_001.fastq.gz
12_K3.scaled.bs10.bw  12_K3_2_S33_L001_R1_001.fastq.gz 12_K3_2_S33_L001_R2_001.fastq.gz
12_K3_2_S33_L002_R1_001.fastq.gz 12_K3_2_S33_L002_R2_001.fastq.gz

12_K3.scaled.bs10.bw 12_K3_3_S34_L001_R1_001.fastq.gz 12_K3_3_S34_L001_R2_001.fastq.gz
12_K3_3_S34_L002_R1_001.fastq.gz 12_K3_3_S34_L002_R2_001.fastq.gz
16_K3.scaled.bs10.bw 16_K3_1_S38_L001_R1_001.fastq.gz 16_K3_1_S38_L001_R2_001.fastq.gz
16_K3_1_S38_L002_R1_001.fastq.gz 16_K3_1_S38_L002_R2_001.fastq.gz
16_K3.scaled.bs10.bw 16_K3_2_S39_L001_R1_001.fastq.gz 16_K3_2_S39_L001_R2_001.fastq.gz
16_K3_2_S39_L002_R1_001.fastq.gz 16_K3_2_S39_L002_R2_001.fastq.gz
16_K3.scaled.bs10.bw 16_K3_3_S40_L001_R1_001.fastq.gz 16_K3_3_S40_L001_R2_001.fastq.gz
16_K3_3_S40_L002_R1_001.fastq.gz 16_K3_3_S40_L002_R2_001.fastq.gz
20_K3.scaled.bs10.bw 20_K3_1_S44_L001_R1_001.fastq.gz 20_K3_1_S44_L001_R2_001.fastq.gz
20_K3_1_S44_L002_R1_001.fastq.gz 20_K3_1_S44_L002_R2_001.fastq.gz
20_K3.scaled.bs10.bw 20_K3_2_S45_L001_R1_001.fastq.gz 20_K3_2_S45_L001_R2_001.fastq.gz
20_K3_2_S45_L002_R1_001.fastq.gz 20_K3_2_S45_L002_R2_001.fastq.gz
20_K3.scaled.bs10.bw 20_K3_3_S46_L001_R1_001.fastq.gz 20_K3_3_S46_L001_R2_001.fastq.gz
20_K3_3_S46_L002_R1_001.fastq.gz 20_K3_3_S46_L002_R2_001.fastq.gz
0_DH_merged.scaled.bs10.bw 0_DH_1_S29_L001_R1_001.fastq.gz 0_DH_1_S29_L001_R2_001.fastq.gz
0_DH_1_S29_L002_R1_001.fastq.gz 0_DH_1_S29_L002_R2_001.fastq.gz
0_DH_merged.scaled.bs10.bw 0_DH_2_S30_L001_R1_001.fastq.gz 0_DH_2_S30_L001_R2_001.fastq.gz
0_DH_2_S30_L002_R1_001.fastq.gz 0_DH_2_S30_L002_R2_001.fastq.gz
0_DH_merged.scaled.bs10.bw 0_DH_3_S31_L001_R1_001.fastq.gz 0_DH_3_S31_L001_R2_001.fastq.gz
0_DH_3_S31_L002_R1_001.fastq.gz 0_DH_3_S31_L002_R2_001.fastq.gz
4_DH.scaled.bs10.bw 4_DH_1_S1_L001_R1_001.fastq.gz 4_DH_1_S1_L001_R2_001.fastq.gz
4_DH_1_S1_L002_R1_001.fastq.gz 4_DH_1_S1_L002_R2_001.fastq.gz
4_DH.scaled.bs10.bw 4_DH_2_S2_L001_R1_001.fastq.gz 4_DH_2_S2_L001_R2_001.fastq.gz
4_DH_2_S2_L002_R1_001.fastq.gz 4_DH_2_S2_L002_R2_001.fastq.gz
4_DH.scaled.bs10.bw 4_DH_3_S3_L001_R1_001.fastq.gz 4_DH_3_S3_L001_R2_001.fastq.gz
4_DH_3_S3_L002_R1_001.fastq.gz 4_DH_3_S3_L002_R2_001.fastq.gz
8_DH_merged.scaled.bs10.bw 8_DH_1_S49_L001_R1_001.fastq.gz 8_DH_1_S49_L001_R2_001.fastq.gz
8_DH_1_S49_L002_R1_001.fastq.gz 8_DH_1_S49_L002_R2_001.fastq.gz
8_DH_merged.scaled.bs10.bw 8_DH_2_S50_L001_R1_001.fastq.gz 8_DH_2_S50_L001_R2_001.fastq.gz
8_DH_2_S50_L002_R1_001.fastq.gz 8_DH_2_S50_L002_R2_001.fastq.gz
8_DH_merged.scaled.bs10.bw 8_DH_3_S51_L001_R1_001.fastq.gz 8_DH_3_S51_L001_R2_001.fastq.gz
8_DH_3_S51_L002_R1_001.fastq.gz 8_DH_3_S51_L002_R2_001.fastq.gz
12_DH_merged.scaled.bs10.bw 12_DH_1_S59_L001_R1_001.fastq.gz 12_DH_1_S59_L001_R2_001.fastq.gz
12_DH_1_S59_L002_R1_001.fastq.gz 12_DH_1_S59_L002_R2_001.fastq.gz
12_DH_merged.scaled.bs10.bw 12_DH_2_S60_L001_R1_001.fastq.gz 12_DH_2_S60_L001_R2_001.fastq.gz
12_DH_2_S60_L002_R1_001.fastq.gz 12_DH_2_S60_L002_R2_001.fastq.gz
12_DH_merged.scaled.bs10.bw 12_DH_3_S61_L001_R1_001.fastq.gz 12_DH_3_S61_L001_R2_001.fastq.gz
12_DH_3_S61_L002_R1_001.fastq.gz 12_DH_3_S61_L002_R2_001.fastq.gz
16_DH_merged.scaled.bs10.bw DualHis_ZT16_IgG.BROAD.05_peaks.broadPeak 16_DH_1_S69_L001_R1_001.fastq.gz
16_DH_1_S69_L001_R2_001.fastq.gz 16_DH_1_S69_L002_R1_001.fastq.gz 16_DH_1_S69_L002_R2_001.fastq.gz
16_DH_merged.scaled.bs10.bw DualHis_ZT16_IgG.BROAD.05_peaks.broadPeak 16_DH_2_S70_L001_R1_001.fastq.gz
16_DH_2_S70_L001_R2_001.fastq.gz 16_DH_2_S70_L002_R1_001.fastq.gz 16_DH_2_S70_L002_R2_001.fastq.gz
16_DH_merged.scaled.bs10.bw DualHis_ZT16_IgG.BROAD.05_peaks.broadPeak 16_DH_3_S71_L001_R1_001.fastq.gz
16_DH_3_S71_L001_R2_001.fastq.gz 16_DH_3_S71_L002_R1_001.fastq.gz 16_DH_3_S71_L002_R2_001.fastq.gz
20_DH_merged.scaled.bs10.bw 20_DH_1_S79_L001_R1_001.fastq.gz 20_DH_1_S79_L001_R2_001.fastq.gz
20_DH_1_S79_L002_R1_001.fastq.gz 20_DH_1_S79_L002_R2_001.fastq.gz
20_DH_merged.scaled.bs10.bw 20_DH_2_S80_L001_R1_001.fastq.gz 20_DH_2_S80_L001_R2_001.fastq.gz
20_DH_2_S80_L002_R1_001.fastq.gz 20_DH_2_S80_L002_R2_001.fastq.gz
20_DH_merged.scaled.bs10.bw 20_DH_3_S81_L001_R1_001.fastq.gz 20_DH_3_S81_L001_R2_001.fastq.gz
20_DH_3_S81_L002_R1_001.fastq.gz 20_DH_3_S81_L002_R2_001.fastq.gz
0_DS_merged.scaled.bs10.bw 0_DS_1_S32_L001_R1_001.fastq.gz 0_DS_1_S32_L001_R2_001.fastq.gz
0_DS_1_S32_L002_R1_001.fastq.gz 0_DS_1_S32_L002_R2_001.fastq.gz
0_DS_merged.scaled.bs10.bw 0_DS_2_S33_L001_R1_001.fastq.gz 0_DS_2_S33_L001_R2_001.fastq.gz
0_DS_2_S33_L002_R1_001.fastq.gz 0_DS_2_S33_L002_R2_001.fastq.gz
0_DS_merged.scaled.bs10.bw 0_DS_3_S34_L001_R1_001.fastq.gz 0_DS_3_S34_L001_R2_001.fastq.gz
0_DS_3_S34_L002_R1_001.fastq.gz 0_DS_3_S34_L002_R2_001.fastq.gz
4_DS.scaled.bs10.bw 4_DS_1_S4_L001_R1_001.fastq.gz 4_DS_1_S4_L001_R2_001.fastq.gz
4_DS_1_S4_L002_R1_001.fastq.gz 4_DS_1_S4_L002_R2_001.fastq.gz
4_DS.scaled.bs10.bw 4_DS_2_S5_L001_R1_001.fastq.gz 4_DS_2_S5_L001_R2_001.fastq.gz
4_DS_2_S5_L002_R1_001.fastq.gz 4_DS_2_S5_L002_R2_001.fastq.gz
4_DS.scaled.bs10.bw 4_DS_3_S6_L001_R1_001.fastq.gz 4_DS_3_S6_L001_R2_001.fastq.gz
4_DS_3_S6_L002_R1_001.fastq.gz 4_DS_3_S6_L002_R2_001.fastq.gz
8_DS_merged.scaled.bs10.bw 8_DS_1_S52_L001_R1_001.fastq.gz 8_DS_1_S52_L001_R2_001.fastq.gz
8_DS_1_S52_L002_R1_001.fastq.gz 8_DS_1_S52_L002_R2_001.fastq.gz
8_DS_merged.scaled.bs10.bw 8_DS_2_S53_L001_R1_001.fastq.gz 8_DS_2_S53_L001_R2_001.fastq.gz
8_DS_2_S53_L002_R1_001.fastq.gz 8_DS_2_S53_L002_R2_001.fastq.gz
8_DS_merged.scaled.bs10.bw 8_DS_3_S54_L001_R1_001.fastq.gz 8_DS_3_S54_L001_R2_001.fastq.gz
8_DS_3_S54_L002_R1_001.fastq.gz 8_DS_3_S54_L002_R2_001.fastq.gz
12_DS_merged.scaled.bs10.bw 12_DS_1_S62_L001_R1_001.fastq.gz 12_DS_1_S62_L001_R2_001.fastq.gz
12_DS_1_S62_L002_R1_001.fastq.gz 12_DS_1_S62_L002_R2_001.fastq.gz
12_DS_merged.scaled.bs10.bw 12_DS_2_S63_L001_R1_001.fastq.gz 12_DS_2_S63_L001_R2_001.fastq.gz
12_DS_2_S63_L002_R1_001.fastq.gz 12_DS_2_S63_L002_R2_001.fastq.gz
12_DS_merged.scaled.bs10.bw 12_DS_3_S64_L001_R1_001.fastq.gz 12_DS_3_S64_L001_R2_001.fastq.gz
12_DS_3_S64_L002_R1_001.fastq.gz 12_DS_3_S64_L002_R2_001.fastq.gz
16_DS_merged.scaled.bs10.bw 16_DS_1_S72_L001_R1_001.fastq.gz 16_DS_1_S72_L001_R2_001.fastq.gz
16_DS_1_S72_L002_R1_001.fastq.gz 16_DS_1_S72_L002_R2_001.fastq.gz

```
16_DS_merged.scaled.bs10.bw  16_DS_2_S73_L001_R1_001.fastq.gz 16_DS_2_S73_L001_R2_001.fastq.gz
16_DS_2_S73_L002_R1_001.fastq.gz 16_DS_2_S73_L002_R2_001.fastq.gz
16_DS_merged.scaled.bs10.bw  16_DS_3_S74_L001_R1_001.fastq.gz 16_DS_3_S74_L001_R2_001.fastq.gz
16_DS_3_S74_L002_R1_001.fastq.gz 16_DS_3_S74_L002_R2_001.fastq.gz
20_DS_merged.scaled.bs10.bw  20_DS_1_S82_L001_R1_001.fastq.gz 20_DS_1_S82_L001_R2_001.fastq.gz
20_DS_1_S82_L002_R1_001.fastq.gz 20_DS_1_S82_L002_R2_001.fastq.gz
20_DS_merged.scaled.bs10.bw  20_DS_2_S83_L001_R1_001.fastq.gz 20_DS_2_S83_L001_R2_001.fastq.gz
20_DS_2_S83_L002_R1_001.fastq.gz 20_DS_2_S83_L002_R2_001.fastq.gz
20_DS_merged.scaled.bs10.bw  20_DS_3_S84_L001_R1_001.fastq.gz 20_DS_3_S84_L001_R2_001.fastq.gz
20_DS_3_S84_L002_R1_001.fastq.gz 20_DS_3_S84_L002_R2_001.fastq.gz
0_WD_merged.scaled.bs10.bw  0_WD_1_S42_L001_R1_001.fastq.gz 0_WD_1_S42_L001_R2_001.fastq.gz
0_WD_1_S42_L002_R1_001.fastq.gz 0_WD_1_S42_L002_R2_001.fastq.gz
0_WD_merged.scaled.bs10.bw  0_WD_2_S43_L001_R1_001.fastq.gz 0_WD_2_S43_L001_R2_001.fastq.gz
0_WD_2_S43_L002_R1_001.fastq.gz 0_WD_2_S43_L002_R2_001.fastq.gz
0_WD_merged.scaled.bs10.bw  0_WD_3_S37_L001_R1_001.fastq.gz 0_WD_3_S37_L001_R2_001.fastq.gz
0_WD_3_S37_L002_R1_001.fastq.gz 0_WD_3_S37_L002_R2_001.fastq.gz
4_WD.scaled.bs10.bw  4_WD_1_S7_L001_R1_001.fastq.gz 4_WD_1_S7_L001_R2_001.fastq.gz
4_WD_1_S7_L002_R1_001.fastq.gz 4_WD_1_S7_L002_R2_001.fastq.gz
4_WD.scaled.bs10.bw  4_WD_2_S8_L001_R1_001.fastq.gz 4_WD_2_S8_L001_R2_001.fastq.gz
4_WD_2_S8_L002_R1_001.fastq.gz 4_WD_2_S8_L002_R2_001.fastq.gz
4_WD.scaled.bs10.bw  4_WD_3_S9_L001_R1_001.fastq.gz 4_WD_3_S9_L001_R2_001.fastq.gz
4_WD_3_S9_L002_R1_001.fastq.gz 4_WD_3_S9_L002_R2_001.fastq.gz
8_WD_merged.scaled.bs10.bw  8_WD_1_S55_L001_R1_001.fastq.gz 8_WD_1_S55_L001_R2_001.fastq.gz
8_WD_1_S55_L002_R1_001.fastq.gz 8_WD_1_S55_L002_R2_001.fastq.gz
8_WD_merged.scaled.bs10.bw  8_WD_2_S56_L001_R1_001.fastq.gz 8_WD_2_S56_L001_R2_001.fastq.gz
8_WD_2_S56_L002_R1_001.fastq.gz 8_WD_2_S56_L002_R2_001.fastq.gz
8_WD_merged.scaled.bs10.bw  8_WD_3_S57_L001_R1_001.fastq.gz 8_WD_3_S57_L001_R2_001.fastq.gz
8_WD_3_S57_L002_R1_001.fastq.gz 8_WD_3_S57_L002_R2_001.fastq.gz
12_WD_merged.scaled.bs10.bw  12_WD_1_S65_L001_R1_001.fastq.gz 12_WD_1_S65_L001_R2_001.fastq.gz
12_WD_1_S65_L002_R1_001.fastq.gz 12_WD_1_S65_L002_R2_001.fastq.gz
12_WD_merged.scaled.bs10.bw  12_WD_2_S66_L001_R1_001.fastq.gz 12_WD_2_S66_L001_R2_001.fastq.gz
12_WD_2_S66_L002_R1_001.fastq.gz 12_WD_2_S66_L002_R2_001.fastq.gz
12_WD_merged.scaled.bs10.bw  12_WD_3_S67_L001_R1_001.fastq.gz 12_WD_3_S67_L001_R2_001.fastq.gz
12_WD_3_S67_L002_R1_001.fastq.gz 12_WD_3_S67_L002_R2_001.fastq.gz
16_WD_merged.scaled.bs10.bw WDR5_16_IgG.05_peaks.narrowPeak 16_WD_1_S75_L001_R1_001.fastq.gz
16_WD_1_S75_L001_R2_001.fastq.gz 16_WD_1_S75_L002_R1_001.fastq.gz 16_WD_1_S75_L002_R2_001.fastq.gz
16_WD_merged.scaled.bs10.bw WDR5_16_IgG.05_peaks.narrowPeak 16_WD_2_S76_L001_R1_001.fastq.gz
16_WD_2_S76_L001_R2_001.fastq.gz 16_WD_2_S76_L002_R1_001.fastq.gz 16_WD_2_S76_L002_R2_001.fastq.gz
16_WD_merged.scaled.bs10.bw WDR5_16_IgG.05_peaks.narrowPeak 16_WD_3_S77_L001_R1_001.fastq.gz
16_WD_3_S77_L001_R2_001.fastq.gz 16_WD_3_S77_L002_R1_001.fastq.gz 16_WD_3_S77_L002_R2_001.fastq.gz
20_WD_merged.scaled.bs10.bw  20_WD_1_S85_L001_R1_001.fastq.gz 20_WD_1_S85_L001_R2_001.fastq.gz
20_WD_1_S85_L002_R1_001.fastq.gz 20_WD_1_S85_L002_R2_001.fastq.gz
20_WD_merged.scaled.bs10.bw  20_WD_2_S86_L001_R1_001.fastq.gz 20_WD_2_S86_L001_R2_001.fastq.gz
20_WD_2_S86_L002_R1_001.fastq.gz 20_WD_2_S86_L002_R2_001.fastq.gz
20_WD_merged.scaled.bs10.bw  20_WD_3_S87_L001_R1_001.fastq.gz 20_WD_3_S87_L001_R2_001.fastq.gz
20_WD_3_S87_L002_R1_001.fastq.gz 20_WD_3_S87_L002_R2_001.fastq.gz
0_IgG_.scaled.bs10.bw  0_IgG_S38_L001_R1_001.fastq.gz 0_IgG_S38_L001_R2_001.fastq.gz
0_IgG_S38_L002_R1_001.fastq.gz 0_IgG_S38_L002_R2_001.fastq.gz
4_IgG.scaled.bs10.bw  4_IgG_S10_L001_R1_001.fastq.gz 4_IgG_S10_L001_R2_001.fastq.gz
4_IgG_S10_L002_R1_001.fastq.gz 4_IgG_S10_L002_R2_001.fastq.gz
8_IgG_.scaled.bs10.bw  8_IgG_S58_L001_R1_001.fastq.gz 8_IgG_S58_L001_R2_001.fastq.gz
8_IgG_S58_L002_R1_001.fastq.gz 8_IgG_S58_L002_R2_001.fastq.gz
12_IgG_.scaled.bs10.bw  12_IgG_S68_L001_R1_001.fastq.gz 12_IgG_S68_L001_R2_001.fastq.gz
12_IgG_S68_L002_R1_001.fastq.gz 12_IgG_S68_L002_R2_001.fastq.gz
16_IgG_.scaled.bs10.bw  16_IgG_S78_L001_R1_001.fastq.gz 16_IgG_S78_L001_R2_001.fastq.gz
16_IgG_S78_L002_R1_001.fastq.gz 16_IgG_S78_L002_R2_001.fastq.gz
20_IgG_.scaled.bs10.bw  20_IgG_S88_L001_R1_001.fastq.gz 20_IgG_S88_L001_R2_001.fastq.gz
20_IgG_S88_L002_R1_001.fastq.gz 20_IgG_S88_L002_R2_001.fastq.gz
V_WD.scaled.bs10.bw  V_WD_1_S53_L001_R1_001.fastq.gz V_WD_1_S53_L001_R2_001.fastq.gz
V_WD_1_S53_L002_R1_001.fastq.gz V_WD_1_S53_L002_R2_001.fastq.gz
V_WD.scaled.bs10.bw  V_WD_2_S54_L001_R1_001.fastq.gz V_WD_2_S54_L001_R2_001.fastq.gz
V_WD_2_S54_L002_R1_001.fastq.gz V_WD_2_S54_L002_R2_001.fastq.gz
V_WD.scaled.bs10.bw  V_WD_3_S55_L001_R1_001.fastq.gz V_WD_3_S55_L001_R2_001.fastq.gz
V_WD_3_S55_L002_R1_001.fastq.gz V_WD_3_S55_L002_R2_001.fastq.gz
Z_WD.scaled.bs10.bw  Z_WD_1_S63_L001_R1_001.fastq.gz Z_WD_1_S63_L001_R2_001.fastq.gz
Z_WD_1_S63_L002_R1_001.fastq.gz Z_WD_1_S63_L002_R2_001.fastq.gz
Z_WD.scaled.bs10.bw  Z_WD_2_S64_L001_R1_001.fastq.gz Z_WD_2_S64_L001_R2_001.fastq.gz
Z_WD_2_S64_L002_R1_001.fastq.gz Z_WD_2_S64_L002_R2_001.fastq.gz
Z_WD.scaled.bs10.bw  Z_WD_3_S65_L001_R1_001.fastq.gz Z_WD_3_S65_L001_R2_001.fastq.gz
Z_WD_3_S65_L002_R1_001.fastq.gz Z_WD_3_S65_L002_R2_001.fastq.gz
V_DH.scaled.bs10.bw  V_DH_1_S47_L001_R1_001.fastq.gz V_DH_1_S47_L001_R2_001.fastq.gz
V_DH_1_S47_L002_R1_001.fastq.gz V_DH_1_S47_L002_R2_001.fastq.gz
V_DH.scaled.bs10.bw  V_DH_2_S48_L001_R1_001.fastq.gz V_DH_2_S48_L001_R2_001.fastq.gz
V_DH_2_S48_L002_R1_001.fastq.gz V_DH_2_S48_L002_R2_001.fastq.gz
V_DH.scaled.bs10.bw  V_DH_3_S49_L001_R1_001.fastq.gz V_DH_3_S49_L001_R2_001.fastq.gz
V_DH_3_S49_L002_R1_001.fastq.gz V_DH_3_S49_L002_R2_001.fastq.gz
```

```
Z_DH.scaled.bs10.bw Z_DH_1_S57_L001_R1_001.fastq.gz Z_DH_1_S57_L001_R2_001.fastq.gz
Z_DH_1_S57_L002_R1_001.fastq.gz Z_DH_1_S57_L002_R2_001.fastq.gz
Z_DH.scaled.bs10.bw Z_DH_2_S58_L001_R1_001.fastq.gz Z_DH_2_S58_L001_R2_001.fastq.gz
Z_DH_2_S58_L002_R1_001.fastq.gz Z_DH_2_S58_L002_R2_001.fastq.gz
Z_DH.scaled.bs10.bw Z_DH_3_S59_L001_R1_001.fastq.gz Z_DH_3_S59_L001_R2_001.fastq.gz
Z_DH_3_S59_L002_R1_001.fastq.gz Z_DH_3_S59_L002_R2_001.fastq.gz
V_DS.scaled.bs10.bw V_DS_1_S50_L001_R1_001.fastq.gz V_DS_1_S50_L001_R2_001.fastq.gz
V_DS_1_S50_L002_R1_001.fastq.gz V_DS_1_S50_L002_R2_001.fastq.gz
V_DS.scaled.bs10.bw V_DS_2_S51_L001_R1_001.fastq.gz V_DS_2_S51_L001_R2_001.fastq.gz
V_DS_2_S51_L002_R1_001.fastq.gz V_DS_2_S51_L002_R2_001.fastq.gz
V_DS.scaled.bs10.bw V_DS_3_S52_L001_R1_001.fastq.gz V_DS_3_S52_L001_R2_001.fastq.gz
V_DS_3_S52_L002_R1_001.fastq.gz V_DS_3_S52_L002_R2_001.fastq.gz
Z_DS.scaled.bs10.bw Z_DS_1_S60_L001_R1_001.fastq.gz Z_DS_1_S60_L001_R2_001.fastq.gz
Z_DS_1_S60_L002_R1_001.fastq.gz Z_DS_1_S60_L002_R2_001.fastq.gz
Z_DS.scaled.bs10.bw Z_DS_2_S61_L001_R1_001.fastq.gz Z_DS_2_S61_L001_R2_001.fastq.gz
Z_DS_2_S61_L002_R1_001.fastq.gz Z_DS_2_S61_L002_R2_001.fastq.gz
Z_DS.scaled.bs10.bw Z_DS_3_S62_L001_R1_001.fastq.gz Z_DS_3_S62_L001_R2_001.fastq.gz
Z_DS_3_S62_L002_R1_001.fastq.gz Z_DS_3_S62_L002_R2_001.fastq.gz
V_IgG.scaled.bs10.bw V_IgG_S56_L001_R1_001.fastq.gz V_IgG_S56_L001_R2_001.fastq.gz
V_IgG_S56_L002_R1_001.fastq.gz V_IgG_S56_L002_R2_001.fastq.gz
Z_IgG.scaled.bs10.bw Z_IgG_S66_L001_R1_001.fastq.gz Z_IgG_S66_L001_R2_001.fastq.gz
Z_IgG_S66_L002_R1_001.fastq.gz Z_IgG_S66_L002_R2_001.fastq.gz
HELA_DH_merged.bs10.bw DualHisHela_.BROAD.05_peaks.broadPeak H_DH_1_S1_L001_R1_001.fastq.gz
H_DH_1_S1_L001_R2_001.fastq.gz H_DH_1_S1_L002_R1_001.fastq.gz H_DH_1_S1_L002_R2_001.fastq.gz
HELA_DH_merged.bs10.bw DualHisHela_.BROAD.05_peaks.broadPeak H_DH_2_S2_L001_R1_001.fastq.gz
H_DH_2_S2_L001_R2_001.fastq.gz H_DH_2_S2_L002_R1_001.fastq.gz H_DH_2_S2_L002_R2_001.fastq.gz
HELA_DH_merged.bs10.bw DualHisHela_.BROAD.05_peaks.broadPeak H_DH_3_S3_L001_R1_001.fastq.gz
H_DH_3_S3_L001_R2_001.fastq.gz H_DH_3_S3_L002_R1_001.fastq.gz H_DH_3_S3_L002_R2_001.fastq.gz
HELA_Q5_merged.bs10.bw singleHisHela_.BROAD.05_peaks.broadPeak H_Q5_1_S7_L001_R1_001.fastq.gz
H_Q5_1_S7_L001_R2_001.fastq.gz H_Q5_1_S7_L002_R1_001.fastq.gz H_Q5_1_S7_L002_R2_001.fastq.gz
HELA_Q5_merged.bs10.bw singleHisHela_.BROAD.05_peaks.broadPeak H_Q5_2_S8_L001_R1_001.fastq.gz
H_Q5_2_S8_L001_R2_001.fastq.gz H_Q5_2_S8_L002_R1_001.fastq.gz H_Q5_2_S8_L002_R2_001.fastq.gz
HELA_Q5_merged.bs10.bw singleHisHela_.BROAD.05_peaks.broadPeak H_Q5_3_S9_L001_R1_001.fastq.gz
H_Q5_3_S9_L001_R2_001.fastq.gz H_Q5_3_S9_L002_R1_001.fastq.gz H_Q5_3_S9_L002_R2_001.fastq.gz
H_IgG1_merged.sorted.RPKM.bs10.sl30.bw H_IgG1_S16_L001_R1_001.fastq.gz H_IgG1_S16_L001_R2_001.fastq.gz
H_IgG1_S16_L002_R1_001.fastq.gz H_IgG1_S16_L002_R2_001.fastq.gz
```

| Genome browser session (e.g. UCSC) | Hela: https://ramaka02.dmz.hpc.mssm.edu/UCSC-TrackHub-Ben-Weekly-hg19/<br>TMN: https://ramaka02.dmz.hpc.mssm.edu/UCSC-TrackHub-Ben-Weekly-mm10/ |

## Methodology

| Replicates | There were 3 independent biological replicates for each antibody and timepoint/treatment, each coming from a biologically independent pool of TMN or plate of HeLa cells. These biological replicates were assessed for similarity visually in IGV and by comparing alignment rate, peak calling (between biological replicates), and heatmaps (using MACS2/Deeptools). All replicates had high concordance. |

| Sequencing depth | |

```
CUT&RUN Sample Total Reads Mapped Reads Uniquely Mapped Reads
0_DH_1_S29 27,310,000.00  19,138,070.00  15,231,930.00
0_DH_2_S30 33,857,086.00  22,542,838.00  17,311,850.00
0_DH_3_S31 37,720,716.00  24,222,108.00  19,511,182.00
0_DS_1_S32 36,834,540.00  21,906,184.00  19,646,366.00
0_DS_2_S33 24,263,036.00  18,676,828.00  17,707,738.00
0_DS_3_S34 27,866,782.00  19,837,988.00  17,410,712.00
0_IgG_S38 24,838,152.00  12,933,064.00  9,756,070.00
0_WD_1_S42 145,472,050.00  83,380,212.00  60,985,598.00
0_WD_2_S36 170,384,052.00  98,318,428.00  71,122,742.00
0_WD_3_S37 35,270,882.00  19,165,556.00  13,923,846.00
8_DH_1_S49 52,269,610.00  29,710,832.00  24,397,130.00
8_DH_2_S50 36,998,278.00  30,050,604.00  24,494,390.00
8_DH_3_S51 40,419,368.00  27,696,056.00  22,443,284.00
8_DS_1_S52 50,859,580.00  29,984,662.00  28,075,652.00
8_DS_2_S53 38,031,308.00  29,442,818.00  27,775,760.00
8_DS_3_S54 35,845,198.00  21,444,614.00  20,094,144.00
8_IgG_S58 43,005,152.00  1,299,376.00  895,190.00
8_WD_1_S55 41,689,350.00  3,633,800.00  2,762,854.00
8_WD_2_S56 38,382,544.00  2,285,056.00  1,725,408.00
8_WD_3_S57 50,556,268.00  4,423,702.00  3,387,026.00
12_DH_1_S59 42,259,492.00  28,133,104.00  21,629,474.00
12_DH_2_S60 30,616,660.00  15,163,518.00  12,479,764.00
12_DH_3_S61 40,945,708.00  23,859,002.00  18,039,254.00
12_DS_1_S62 52,338,258.00  33,210,612.00  26,393,732.00
12_DS_2_S63 34,494,070.00  24,143,684.00  22,433,166.00
12_DS_3_S64 38,957,388.00  24,213,422.00  22,296,134.00
12_IgG_S68 40,479,908.00  22,168,700.00  16,662,982.00
```

```
12_WD_1_S65 39,362,640.00  14,189,274.00  10,815,136.00
12_WD_2_S66 43,578,482.00  7,697,774.00  5,919,686.00
12_WD_3_S67 27,574,516.00  4,293,534.00  3,495,976.00
16_DH_1_S69 33,757,586.00  29,361,130.00  23,752,622.00
16_DH_2_S70 33,344,202.00  29,017,722.00  23,642,692.00
16_DH_3_S71 30,410,508.00  26,450,556.00  21,805,204.00
16_DS_1_S72 36,357,196.00  30,456,338.00  28,877,942.00
16_DS_2_S73 29,347,110.00  26,841,350.00  25,421,054.00
16_DS_3_S74 30,114,608.00  23,679,988.00  22,187,514.00
16_IgG_S78 25,302,004.00  5,882,086.00  4,731,038.00
16_WD_1_S75 31,995,666.00  13,152,232.00  11,088,520.00
16_WD_2_S76 30,265,902.00  11,832,936.00  10,303,224.00
16_WD_3_S77 28,633,122.00  7,582,756.00  6,467,248.00
20_DH_1_S79 21,578,164.00  18,400,694.00  14,638,490.00
20_DH_2_S80 34,264,656.00  28,928,486.00  24,103,634.00
20_DH_3_S81 29,568,680.00  27,069,202.00  22,049,074.00
20_DS_1_S82 37,368,162.00  33,076,776.00  31,023,900.00
20_DS_2_S83 26,879,130.00  23,250,150.00  21,905,280.00
20_DS_3_S84 28,888,342.00  23,754,990.00  22,413,478.00
20_IgG_S88 34,850,010.00  5,608,916.00  4,374,330.00
20_WD_1_S85 22,031,330.00  4,625,340.00  3,953,072.00
20_WD_2_S86 33,663,820.00  3,749,416.00  3,002,986.00
20_WD_3_S87 28,524,766.00  6,099,688.00  4,759,672.00
0_K2_1_S11 38,592,784.00  30,304,078.00  28,372,604.00
0_K2_2_S12 50,108,656.00  37,949,908.00  35,239,812.00
0_K2_3_S13 60,436,514.00  42,626,340.00  39,164,730.00
0_K3_1_S14 46,896,318.00  21,782,728.00  17,700,696.00
0_K3_2_S15 46,494,206.00  22,304,170.00  18,191,010.00
0_K3_3_S16 44,252,650.00  20,464,314.00  16,619,598.00
4_DH_1_S1 49,322,612.00  5,530,438.00  4,598,906.00
4_DH_2_S2 51,945,794.00  7,297,390.00  5,914,242.00
4_DH_3_S3 52,749,384.00  3,444,340.00  2,739,378.00
4_DS_1_S4 52,183,116.00  7,318,656.00  6,616,086.00
4_DS_2_S5 43,712,404.00  4,906,552.00  4,527,294.00
4_DS_3_S6 48,815,458.00  4,180,178.00  3,839,134.00
4_IgG_S10 30,956,338.00  776,590.00  594,992.00
4_K2_1_S17 45,992,640.00  15,045,056.00  14,129,568.00
4_K2_2_S18 27,370,070.00  9,026,356.00  8,379,914.00
4_K2_3_S19 52,109,530.00  21,389,512.00  20,195,410.00
4_K3_1_S20 29,391,002.00  2,025,650.00  1,697,494.00
4_K3_2_S21 43,535,218.00  3,826,188.00  3,223,030.00
4_K3_3_S22 48,075,330.00  3,967,366.00  3,283,742.00
4_WD_1_S7 80,362,542.00  25,203,878.00  18,439,628.00
4_WD_2_S8 98,528,138.00  25,490,912.00  19,413,628.00
4_WD_3_S9 53,818,474.00  2,500,754.00  2,011,166.00
8_K2_1_S23 39,711,548.00  21,397,248.00  18,582,456.00
8_K2_2_S24 40,101,454.00  26,623,774.00  24,153,352.00
8_K2_3_S25 32,906,498.00  20,369,976.00  18,568,684.00
8_K3_1_S26 38,113,642.00  9,817,790.00  7,920,204.00
8_K3_2_S27 38,152,118.00  11,029,236.00  8,797,988.00
8_K3_3_S28 46,712,314.00  25,661,040.00  20,207,194.00
12_K2_1_S29 36,976,508.00  13,069,650.00  12,505,044.00
12_K2_2_S30 43,545,256.00  14,900,536.00  14,182,790.00
12_K2_3_S31 38,718,196.00  15,670,436.00  14,791,794.00
12_K3_1_S32 43,479,688.00  4,752,668.00  3,837,618.00
12_K3_2_S33 31,205,306.00  1,026,976.00  885,716.00
12_K3_3_S34 25,404,798.00  1,837,676.00  1,448,170.00
16_K2_1_S35 56,577,908.00  16,183,532.00  15,469,104.00
16_K2_2_S36 38,347,870.00  12,287,338.00  11,730,840.00
16_K2_3_S37 30,119,956.00  11,670,598.00  11,138,836.00
16_K3_1_S38 31,721,220.00  2,218,408.00  1,833,964.00
16_K3_2_S39 27,345,420.00  994,878.00  859,788.00
16_K3_3_S40 31,164,748.00  646,304.00  520,608.00
20_K2_1_S41 37,693,126.00  10,777,708.00  10,177,506.00
20_K2_2_S42 41,456,140.00  13,934,760.00  12,857,290.00
20_K2_3_S43 30,775,094.00  8,892,270.00  8,256,114.00
20_K3_1_S44 29,213,312.00  8,071,824.00  7,547,998.00
20_K3_2_S45 47,298,070.00  12,319,872.00  11,120,578.00
20_K3_3_S46 63,796,638.00  19,151,266.00  17,635,486.00
V_DH_1_S47 38,354,362.00  35,656,150.00  28,585,514.00
V_DH_2_S48 41,777,574.00  36,364,144.00  30,996,082.00
V_DH_3_S49 37,077,836.00  28,418,822.00  24,390,660.00
V_DS_1_S50 48,050,330.00  45,020,544.00  43,058,582.00
V_DS_2_S51 41,674,098.00  38,302,310.00  36,794,930.00
V_DS_3_S52 43,751,872.00  39,941,036.00  38,095,330.00
V_IgG_S56 31,749,078.00  7,375,664.00  5,897,488.00
```

```
V_WD_1_S53  21,798,158.00   8,825,170.00   7,582,322.00
V_WD_2_S54  19,607,102.00   6,052,022.00   4,929,324.00
V_WD_3_S55  32,500,474.00  11,400,794.00   9,744,182.00
Z_DH_1_S57  61,270,690.00  16,125,394.00  12,245,922.00
Z_DH_2_S58  64,077,002.00  18,341,110.00  14,615,612.00
Z_DH_3_S59  67,469,872.00  29,555,502.00  23,851,138.00
Z_DS_1_S60  60,843,838.00  51,695,172.00  49,020,160.00
Z_DS_2_S61  53,028,422.00  32,888,186.00  29,970,752.00
Z_DS_3_S62  53,540,328.00  41,497,946.00  37,406,712.00
Z_IgG_S66   52,145,422.00  15,006,002.00  11,835,444.00
Z_WD_1_S63  51,181,810.00  34,580,236.00  27,206,620.00
Z_WD_2_S64  59,028,068.00  33,287,442.00  26,431,774.00
Z_WD_3_S65  60,486,502.00  20,390,910.00  16,681,584.00
```

**Antibodies**

H3K4me2: AM 39141, H3K4me3:Epicypher 13-0041, H3K4me3Q5his: Millipore ABE2570, WDR5: CST D9E1I, H3K4me3Q5ser: Millipore ABE2580, Rabbit IgG: Abcam ab172730, H3Q5his: Millipore ABE2578
All used 1:50 in antibody buffer

**Peak calling parameters**

MACS2 was used for peak calling with the following parameters (except for WDR5, which did not use --broad mode): macs2 callpeak \
-t ZT#AntibodyX.merged.bam \
-c ZT#IgG.bam \
-n ZT#AntibodyY_IgG_broad \
-f BAMPE -g mm (or hh) --keep-dup all \
--broad \
-q 0.05 \
--broad-cutoff .05 \
--outdir out_directory

**Data quality**

MACS2 peak calling was used with the corresponding IgG control. Peaks were visually inspected in IGV and by heatmap. Peak calling parameters were identical except in some cases where q values of .01 were used to reduce background peaks. Number of called peaks with q-values <.05 or .01 were used and are as follows (Peak-calling .broadPeak and .narrowPeak are available in the GEO submission):
singleHisHela_.BROAD.05_peaks.broadPeak: 17,981 Peaks
DualHisHela_.BROAD.05_peaks.broadPeak: 20,181 Peaks
K4me2_ZT0_IgG.BROAD.01_peaks.broadPeak: 58,790 Peaks
K4me3_ZT0_IgG.BROAD.05_peaks.broadPeak: 18,362 Peaks
WDR5_16_IgG.05_peaks.narrowPeak: 4,873 Peaks

**Software**

Sequencing files were demultiplexed using bcl2fastq2 (Illumina, v2.20). Samples were aligned to the hg19 or mm10 genome using bowtie2 (2.5.0), with the following parameters: --local --very-sensitive-local --phred33 -I 10 -X 700 --dovetail --no-unal --no-mixed --no-discordant. Low quality reads were filtered out using Samtools (1.9) with a cutoff MAPQ score of 30, and only unique reads kept for further processing. Unique read files for each replicate/timepoint/antibody  were merged and used for peak calling using MACS2 (3.0.0a6) using the callpeaks function with the options -f BAMPE -q 0.05 --broad --broad-cutoff .05, using the corresponding IgG sample as the -c. For visualization, each sample was normalized by scaling the samples based off of the E. coli spike in, and genome coverage tracks (bigwig files) were produced using deepTools (3.5.1) bamCoverage function with the options –binSize 10 –smoothLength 30 --normalizeUsing None --scaleFactor # and using an ENCODE hg19 or mm10 blacklist file (https://doi.org/10.1038/s41598-019-45839-z) to discard regions with consistent non-specific signal. Additional code details can be found in the supplemental information

