## [Peer Review file · Nature]

Bidirectional histone monoaminylation dynamics regulate neural rhythmicity

Corresponding Author: Dr Ian Maze

Version 1:

Reviewer comments:

Referee #1

(Remarks to the Author)

In this manuscript titled "histone monoaminylation dynamics are regulated by a single enzyme and promote neural rhythmicity", Zheng and the colleagues reported the "eraser" function of a chromatin regulatory enzyme TGM2, which is critical for epigenetics states regulation in neural cells. The findings are somewhat interesting, however, in vivo NGS analysis could further strengthen the main conclusion of this manuscript and the knockin rescue is more desired in the in vivo model for neural rhythmicity. I have included a few shortcomings of the paper below that the authors should address as main controls for the study.

In Figure 1B, besides the two different small molecule inhibitors, the catalytic-dead TGM2 should be transfected in HEK293 cells as negative control. Similarly, a catalytic-dead knock-in should be conducted in HeLa cells, which express endogenous TGM2, to fully demonstrate the catalytic activity of TGM2 is necessary for H3 serotonylation.

In Figure 2, the cross-talks between different histone modifications are critical for their physiological functions. The quantification of the modifications with histone H3 N-terminal peptide looks convincing, however, I suggest the authors to quantify these different modifications in living cells by mass spectrometry analysis.

In Figure 4, genome-wide ChIP-seq studies of H3Q5his should be conducted with cells/tissues to further demonstrate the function of this specific histone modification involved in transcriptional regulation. In addition, TGM2 ChIP-seq is also critical for this model, and should also be included. This is a critical piece of information to link the direct role of H3Q5his to CLOCK gene expression as well.

In Figure 5, the authors try to understand the impact of H3Q5his on H3K4 methylation and H3K4 methyltransferases in cells. However, all the experiments and analysis were conducted in vitro. The authors need to determine MLL1-4/SET1A-B occupancy and the H3K4me1-3 modifications by ChIP-seq in cells with/without TGM2 or treated with/without TGM2 inhibitors.

Finally, the impact of H3Q5his on histone H3K4 demethylases should also be considered or discussed in the manuscript.

Referee #2

(Remarks to the Author)

In this manuscript, Zheng and colleagues explore the molecular mechanism by which the transglutaminase TGM2 catalyzes the exchange of posttranslational modifications (PTMs, derived from monoamine neurotransmitters) at H3Q5, report a novel PTM – histaminylation, and convincingly demonstrate that this new PTM has a circadian expression pattern and contributes to neural rhythmicity. The authors further show that H3Q5his inhibits binding of the MLL1 complex to histone H3 and affects the methyltransferase activity of this complex.

Overall, this is a very well executed study – it addresses an important biological question and will likely have a profound impact on neuronal epigenetics due to the discovery of (1) the first enzyme that functions as a 'PTM exchanger', expanding the fundamental set of established epi- players, such as PTM-writers, PTM-erasers, PTM-readers, (2) a new PTM, histaminylation, which is directly implicated in neuronal signaling, and (3) diurnal rhythmicity of H3Q5his, the only PTM known to have this capability. This study should be of interest to a wide scientific community, and I enthusiastically support publication.

A few comments:

1. Page 6, first two sentences describing the mechanism are unclear.
Better (or something like this): ... whether this mechanism involves a nucleophilic attack of Cys277 from/of TGM2 on the carbon atom of the γ -carboxamide group of Q5 from/of histone H3 (Fig. 2A). In this mechanism [I wouldn't say 'hypothesized' - as it's somewhat predictable- you have two nucleophiles- monoamines, similar in structure around the nucleophile center (i.e. all of them are primary amines), and the equilibrium will be shifted toward the formation of the amide product derived from the monoamine that is present in higher local concentration] the second nucleophilic attack of different/another monoamine neurotransmitter on the thioester intermediate results in the formation of
2. Throughout the text, 'the re-writing activity' doesn't sound as exciting as it should, I would simply use 'the catalytic activity' [of a PTM/monoamine exchanger?]
3. Page 8, unclear phrase: ... is provided as a donor...
4. Page 8, unclear sentence starting with "Our results not only...". To simplify, Our results show that TGM2 catalyzes transglutamination at H3Q5, leading to the replacement/substitution/switching of one PTM with/to another (or something similar).
5. Page 9, the phrase '...H3Q5his was efficiently converted to the alternative...' needs to be revised, as well as the next sentence, '...histone monoaminylation is a family..... and based upon...'
6. Overall, the data shown in Figs. 1-3 are novel, high quality, and exciting, but I would display them in one figure, combining Figs. 1a, 1b, 2a, 2c and 3a,b, and move other panels to a supplement. I would also condense the text describing these results. I feel the paper would benefit by shortening some explanations- it's a superbly interesting part, which needs to be presented in a concise, straight to the point way to be fully appreciated.
7. The data shown in Fig. 4 on the other hand, should be displayed in two separate figures- it's a very important part of this work.
8. Page 10, the sentence starting with "Such pattern..." could be clearer, say '...decarboxylase, which catalyzes the formation of histamine from histidine...
9. Page 11, incorrect phrase: 'other valence states of H3K4 methylation'
10. Page 11, the top paragraph of this page describing K4me dynamics is too long. The authors wanted to simply say that we tested me1/2/3 and found that while ... do not, me2 does. I understand the motivation to write a long text to link it to the following study re MLL activity, but I feel it is unnecessary here.
11. Instead, I would probably add a possible explanation re differences found in TMN vs SCN- this could be exciting and important to future studies.
12. The data shown in Fig. 5 stand alone, and this part reads more like a somewhat distraction rather than help to the first two parts of the manuscript. There are several issues with including these results. Of note, the data quality is excellent, I am not questioning it, I am troubled re cohesion with this part, feels more like an artificial addition.
13. One issue is that would the structure of WDR5 in complex with H3Q5his show how the his moiety is bound- it would have added the significance to this study (though it's still quite a long link to the first two parts of the manuscript). But modeling never looks overly attractive, especially if the study reports a breakthrough discovery.
14. The second issue is that the MLL1 (human COMPAS) complex is known to catalyze tri-methylation to a higher extent than it catalyzes di-methylation, therefore focusing on K4me2 and not showing K4me3 in Fig. 5 raises more questions than answers.
15. Third, the rationale to go after MLL1 is somewhat weak, why not to test other writers/readers/erasers specific for methylated K4? There is no evidence that other proteins do not play a role or greatly affect the behavior of Q5his.
16. Fig. 5d, the rationale to switch to MLL3 is unclear and weak, as the catalytic activity and genome localization of MLL3 and MLL1 differ considerably.
17. Fig 5i-k, the results shown are even further away from the first two parts of the manuscript than MLL1.
18. Page 14, unclear phrase: 'displayed abrogated binding', better: abrogates or decreases binding.
19. Page 15, unclear phrase: 'WDR5WD40-H3 complex in the presence of an H3Q5his peptide.'
20. Page 15, the phrase '...to act as writer, eraser and re-writer...' could be simplified to 'to act as a PTM exchanger' (or similar). best regards, Tatiana K.

Referee #3

(Remarks to the Author)

Manuscript by Zheng et al provides new evidence that TGM2 can utilize histamine produced in histaminergic neurons to introduce or erase histone H3Q5His modifications. The authors detected this modification in the hypothalamic tuberomammillary nucleus (TMN) in mouse brain in vivo and demonstrate that the modification exhibits circadian rhythmic pattern. Overexpression of mutant histone H3 that cannot be modified on Q5 results in an increase in H3K4me2, some deregulation of circadian gene expression, and a mild change in circadian motor behavior.

This is an interesting and novel extension of previous work pioneered by this group on monoamine H3Q5 modifications. The strongest point of the manuscript is the rigorous and convincing biochemical characterization of TGM2 dependent histamination of H3Q5. Where the study is less convincing, is in the analysis of H3Q5His functional role in the regulation of rhythmic gene expression in the TMN nucleus and its consequences for animal behavior (see major concerns below). Most importantly, the study does not offer any clarity on more fundamental mechanistic questions – what is the chromatin-wide distribution of histamine modifications in neurons, how does it change during the circadian cycle, does it correlate/anti-correlate with any of the other H3 modifications, and how do the modifications affect expression of associated genes.

Major concerns:

1. Punch collection of TMN nucleus from 1 mm thick sections can result in variable ratios of histaminergic and non-

histaminergic neurons per specimen. Therefore, it is necessary to normalize all western blots to the fraction of histaminergic neurons in each sample. This should be performed by comparing markers specific to histaminergic neurons and pan neuronal markers in protein lysates.

2. What are the levels of H3Q5Ser in the TMN nucleus over the circadian cycle? Are Ser modifications on H3 exchanged for His modifications and vice versa or non-modified histones are subjected to the circadian changes in His modifications?

3. It is not clear why were the analyses in Fig S7B interpreted as a proof that these modifications and proteins are non-rhythmic? Just showing that the data do not fit well with 3rd polynomial function (the authors should list the p-values for their analyses in this figure and Fig 4D) cannot be used as a proof, as other oscillatory patterns can be present in the data. Indeed, it appears that there are statistically significant differences across individual timepoints. The authors should perform ANOVA analysis to evaluate changes in gene expression across the time series.

4. The same applies to the analysis of H3Q5His in SCN. What does ANOVA statistical analysis of this dataset show? Is the increase at 16h significant compared to 4h? Does this suggest delayed accumulation of histamine modification outside of the TMN? The authors should extend the analysis of H3Q5His to other brain regions that are diffusely innervated by histaminergic neurons to bring more clarity how broadly the modifications might affect brain function.

5. Oscillation of H3Q5His in Fig 4D seems to be more complex than a typical circadian gene with one maximum and one minimum over the period. There appear to be two maxima (0/24h and 8-12h) and two minima (4h and 20h). Is it possible that technical differences in sample collection across the 24 h period could contribute to the reported observations (another reason to include quantification of histaminergic and generic neuronal markers in lysates)? If the pattern is real, how do the authors interpret this strange rhythmicity and how does it correlate with histamine levels in the brain?

6. The authors should take advantage of their antibodies to purify H3Q5His and H3Q5Ser from TMN and quantify the amount of K4me1/2/3 modifications associated with these two modifications. Such analysis would provide important support to the hypothesis that the two modifications lead to a differential recruitment of the MLL1 complex.

7. What are the anticipated concentrations of WDR5 in the nucleus? Micromolar Kd values seem high, raising a question whether the proposed WD40 based binding mechanism is physiological. Please address and discuss.

8. Effects of viral overexpression of mutant H3Q5A should be compared against the overexpression of WT H3, control non-transduced tissue should be separated out of the analysis. Looking at the data presented in figure 4H, I do not believe that there is any statistical difference in H3Q5His/H3 between the red and white dots. This is an important point, as overexpression of H3 will affect stoichiometry of H3 and its modifiers. Thus, the authors should quantify other H3 modifications following WT and mutant H3 expression (e.g. H3K4me1, H3K4me3, H3K27Ac).

9. The authors mention that expression of H3Q5A resulted in "significant overlaps with genes disrupted by H3Q5his manipulations in comparisons to those identified as being rhythmic". This should be supported by more detailed analysis and inclusion of the data. Are these genes increased or decreased? Are they deregulated in a specific phase of the cycle? More specifically, the authors should comment on the level of histidine decarboxylase expression; and whether they anticipate attenuated or enhanced histamine neurotransmission in the manipulated animals.

10. Peptide co-IP experiments (Fig 5F-H) are interpreted as: "H3Q5his displayed abrogated binding" to MLL1 complex. This is misleading as the histamine modification potentiated the interaction compared to unmodified H3 peptide. Importantly, the experiments need to be performed with K4me1,2,3 modified peptides and double modified peptides as well.

11. Despite the biochemical analysis, it is impossible to conclude which histones are histamine modified in cells (random, at promoters, within genes, in distal enhancers, insulators or other elements?) and how does the modification correlate with other H3 modifications. Without such analysis, it is difficult to arrive to a satisfactory interpretation of the data and a plausible model. The authors should include ChIP-seq (or CUT&RUN) analysis of H3 modifications in the TMN at 8h and 20h and compare these with patterns of H3K4me2 and H3K4me3. If this is technically not possible, the chromatin analysis combined with gene expression analysis should be performed in the HEK/TGM2 cell system, as it will help to establish how broad and strong effect do histamine modifications play in the control of gene expression.

Version 2:

Editorial Note

We have not received a review report from Reviewer #1, so we sought the opinion of another reviewer on the panel to assess the revisions made. This reviewer has confirmed that the revisions adequately address the concerns raised.

Reviewer comments:

Referee #2

(Remarks to the Author)

The authors have substantially revised the manuscript and went beyond and above to address the reviewers' comments and further strengthen this remarkable study.

A few remaining points:

1. Modeling of the structures in the revised version remains an issue that needs to be resolved by editing the text and revising Fig. 2. The authors have added a modeled PTM moiety on the crystal structure of the MLL1 SET domain (Fig. 2b), which I would discourage to do. Because the major scientific breakthrough reported here is the discovery and characterization of the new PTM, H3Q5his, any overstatement, especially regarding Q5his, would only damage the incredible work of this team and is unnecessary here. I would certainly avoid showing [published by others structure of MLL1] and manually manipulating it by adding a PTM (Fig. 2b).
2. Fig. 2g shows a top-quality [as many other impressive structures reported by the Haitao's group] structure of the complex, and I don't see a problem that the electron density of the his moiety is not observed. I would keep this panel as it is but add the label 'WDR5-H3Q5his complex' or "WDR5-H3Q5his" below the image to emphasize the presence of Q5his. This will be sufficient and clear - reporting experimentally derived data.
3. Please do not manually add the His moiety in Fig. 2h and 2i, it's sure would be perceived as a misleading representation by the scientific community. [I would remove Fig. 2i entirely]
4. I would suggest toning down the explanation as to how the his moiety is recognized. Is this moiety in fact positively charged here? what is the pKa value of it? if the average pKa of a His (residue, the side chain) is 6.0 in proteins, would it be positively charged at pH 7.4? or what percent will be positively charged? and it is shown as a neutral group in Fig. 1 of this manuscript.
5. The contribution of steric hindrance might be a valid explanation too, as the difference between Kds for K259A and K297E is quite small, 7 μ M and 16 μ M, and we cannot exclude favorable intramolecular contact(s) involving the side chain of E in the complex that are unrelated to the presence of the his moiety.
6. If I am not mistaken, comparison of the current structure with the structure of WDR5-H3Q5ser (Zhao, 2021) does not support the idea of the charge-charge repulsion- the binding pocket for ser is essentially neutral or even slightly negatively charged. Does this region of the domain (surface) differ in the current structure?
7. One possibility that might help to elaborate the uniqueness of Q5his is to compare the position/orientation of the Q5 side chain in the current structure and structures of the WDR5-H3unmod and WDR5-H3Q5ser complexes. If the side chain of Q5 is restrained in these two known complexes, then it's logical to say that the presence of the his moiety makes Q5 flexible. But nothing else could be drawn at this point.
8. To summarize, I would suggest presenting only experimentally obtained structural data in figures and avoiding unneeded speculations at this point. This manuscript has exceedingly large number of amazing results, and following studies could be focused on proper analysis of the his binding site (it's not as solid at this point and is not as essential for the original discovery report).
9. Fig. 2a, should be ASH2L. best regards, Tatiana K.

Referee #3

(Remarks to the Author)

The authors conducted a series of additional experiments and analyses that address the majority of my criticisms. They also explained that some requested experiments are beyond the scope of the current manuscript, while others are technically not feasible. I do not have any further concerns.

Version 3:

Reviewer comments:

Referee #2

(Remarks to the Author)

The authors have very well addressed all my comments.

We would very much like to thank the three Reviewers for their encouraging and helpful comments on our manuscript, for which we now provide an extensively revised version that includes considerable new data and analyses (please see the list of new data included in this resubmission below; >50 new data panels in total), along with restructuring of the text to address – to the best of our ability – all of the comments raised. We now feel that our manuscript is greatly improved from the previous version, and we look forward to receiving further feedback from the Reviewers and Editor regarding this work.

List of New Data Included in this Resubmission:

Supplementary Figure 1a-b (related to **Figure 1**): Data supporting that the TGM2 enzyme is the primary “writer” of histone monoamination in cells.

Supplementary Figure 4e (related to **Figure 1**): in cellulo LC-MS/MS identification of H3Q5 histamination (shown as a mirror plot aligned to a synthetic H3Q5his peptide).

Supplementary Figure 5a-f (related to **Figure 2**): ITC assessments of binding affinities between the monoaminylated histone H3 tail and known interactors of histone H3(K4me3) (e.g., WDR5, CHD1, TAF3, BPTF, JMJD2A and JARID1A). Comparing the affinities of the “reader” domains of these proteins to unmodified vs. monoaminylated peptides (with or without the adjacent H3K4me3) revealed that H3Q5his selectively antagonizes WDR5’s binding to the H3 tail compared to unmodified H3 or H3Q5ser.

Figure 2b: New structural modeling of H3Q5his bound to the MLL1 core complex, further supporting potential antagonistic relationships between the histaminyl modification and the catalytic activity of MLL1 owing to unfavorable electrostatics.

Figure 2d; Supplementary Figure 6c-h (related to **Figure 2**): Histone methyltransferase (HMT) activity assays with MLL1-4 and SETD1A/B, demonstrating that H3Q5his inhibits the activity of all H3K4 HMT complexes tested (note that H3Q5ser does not have major inhibitory effects, as demonstrated against MLL1; **Supplementary Figure 6a**). These data validate the predictions made from the results presented in **Figure 2b**.

Figure 2e: Peptide affinity pulldown assay demonstrating that H3Q5his antagonizes the binding of recombinant WDR5 (the only member of the MLL/SETD1 complexes to directly interact with the H3 tail in monomeric form; **Supplementary Figure 6i** – related to **Figure 2**) to the H3 tail, whereas H3Q5ser potentiates this interaction (all now appropriately normalized to peptide concentrations).

Figure 2f: Peptide affinity pulldown assay demonstrating that H3K4me3Q5his antagonizes the binding of recombinant WDR5 to the H3 tail in comparison to H3K4me3 and H3K4me3Q5ser (all now appropriately normalized to peptide concentrations).

Supplementary Figure 6j (related to **Figure 2**): Peptide affinity pulldown assay demonstrating that H3Q5his antagonizes the binding of recombinant WDR5 within the MLL1 complex to the H3 tail, whereas H3Q5ser potentiates this interaction (all now appropriately normalized to peptide concentrations).

Figure 2k: in cellulo assessment of tagged wild type vs. mutant WDR5-K259A’s chromatin binding at high salt (vs. soluble nuclear enrichment), demonstrating that WDR5-K259A displays significantly higher levels of chromatin enrichment in the presence of H3Q5his vs. wildtype WDR5. The WDR5-K259A mutant is not antagonized in its binding to the H3 tail by Q5his; biochemical validations to this effect are presented in **Figure 2g-j**.

Figure 2l: qPCR analyses of H3Q5his-enriched target genes in cells expressing wild type vs. mutant WDR5-K259A, demonstrating that the expression of these genes is induced following transfection with WDR5-K259A, which can bind to H3Q5his and displays greater chromatin enrichment vs. WT WDR5. These results validate the loss of H3Q5his’ antagonistic effect on WDR5 binding in the mutant (**Figures 2g-k**), leading to increased chromatin binding and gene activation in cellulo.

Supplementary Figure 7j-l (related to **Figure 3**): During the revision process, we generated new lots for both our H3Q5his and H3K4me3Q5his antibodies, which took nearly one year to generate and validate. These new

lots are amenable for CUT&RUN-seq and were used to address Reviewers' concerns regarding the genomic localization of the indicated modifications. In these Supplementary Figure panels, we provide the antibody validations, demonstrating that they both work efficiently in CUT&RUN-seq assays in HeLa cells, where histamine is provided by serum in the media (note that the H3Q5his antibody did not work for CUT&RUN-seq in brain tissues, and as such, we used the H3K4me3Q5his antibody for in vivo assessments; *vide infra*). We also show that both antibodies display similar patterns of enrichment genome-wide, where >85% of the peaks were found to be located at genic loci, and that such enrichment is TGM2-dependent.

Supplementary Figure 8a-c (related to **Figure 3**): Quantifications and statistical assessments of global levels of H3Q5his, H3K4me3Q5his and TGM2, evaluated by western blotting, across circadian time in tuberomammillary nucleus/TMN (where both marks were found to be rhythmic and TGM2 is not) and suprachiasmatic nucleus/SCN (where H3Q5his is not rhythmic).

Figure 3d-i; Supplementary Figure 9d-I (related to **Figure 3**); **Supplementary Figure 10a-d** (related to **Figure 3**): Using our newly validated antibodies, we performed CUT&RUN-seq for H3K4me3Q5his (where ~84% of the peaks were found to be located within gene promoters, exons and introns), H3K4me3Q5ser, H3K4me2, H3K4me3 and WDR5 (vs. IgG; *E. coli* DNA spike-in controls were included in all of these analyses) in TMN tissues across Zeitgeber Time, ZT (ZT0, ZT4, ZT8, ZT12, ZT16, ZT20). These data demonstrated that H3K4me3Q5his displays rhythmic patterns of genomic enrichment, consistent with global levels of the marks, as assessed via western blotting (**Figure S8a**), which anti-correlated with H3K4 methylation patterns genome-wide. Fluctuating levels/enrichment of the marks across ZT did not appear to be dependent on altered expression of the histamine-producing enzyme, *Hdc*, differences in punching efficiency, as assessed by comparisons of *Hdc* in TMN vs. SCN, or expression of *NeuN* (**Figure S9a** – related to **Figure 3**). In addition, these new data are consistent with the biochemical analyses presented in **Figure 2**, indicating that H3Q5his antagonizes WDR5 binding and MLL1-4/SETD1A-B complex activities, whereas H3Q5ser is permissive for WDR5 binding and HMT activities. Rhythmic fluctuations in H3K4me3Q5his and H3K4me3Q5ser (along with associated changes in WDR5 binding) also correlate with rhythmic patterns of gene expression that appear to be dependent on the transcription factor complex, *Clock/Bmal1*, based on ontology analyses.

Figure S9b-c (related to **Figure 3**): ELISA assays to evaluate both histamine and serotonin total levels in TMN across ZT, revealing that neither monoamine displays circadian fluctuations in terms of their total levels. One caveat of this analysis is that such assessments cannot provide accurate measurements of intracellular, or subcellular, pools of the monoamines under study, which may indeed display intra-nuclear rhythmic fluctuations, as might be predicted based upon the biochemical data presented in **Figure 1**. We addressed this caveat in the text.

Figure 3k-l: CUT&RUN-seq assessments of H3K4me3Q5his, H3K4me3Q5ser and WDR5 in mouse TMN tissues following treatments with Zolpidem, a sleep-inducing drug, which is administered during the mouse's active phase. The results indicated that Zolpidem alters genome-wide enrichment of H3K4me3Q5his, H3K4me3Q5ser and WDR5, and promotes patterns that phenocopy those observed during phases of inactivity, including at known circadian genes.

Figure 4b-d: RNA-seq analysis of mouse TMN tissues across ZT transduced with AAV-H3.3Q5A, demonstrating that direct attenuation of H3 monoaminylation levels in TMN results in a robust loss of rhythmic, *Clock*-associated gene expression (**Figure 4c-d**). **Figure 4b,e** show that H3.3Q5A overexpression reduces H3Q5his levels (vs. both GFP and H3.3 WT controls) and disrupts circadian locomotor behavior.

Supplementary Figure 10e: Western blotting analyses of H3K4me3 and H3K27ac in TMN following transduction with AAV-GFP vs. H3.3 WT vs. H3.3Q5A, demonstrating that transduction with either H3.3 WT or H3.3Q5A does not impact the relative levels of H3K4me3 (total) or H3K27ac in comparison to GFP controls.

Supplementary Figure 11: Mechanistic model describing the results outlined in our manuscript.

General note on the restructuring of our manuscript:

In accordance with the Reviewers' recommendations, we have now significantly restructured our manuscript to improve its general flow and cohesiveness. As such, we now present the data as follows – 1) demonstration that TGM2 is both the "writer" and "eraser/exchanger" of histone H3 monoaminylation, including H3Q5his, which displays relative enrichment in vivo in TMN and is presented in this manuscript for the first time (**Figure**

1; Supplementary Figures 1-4, 12/uncropped blots); 2) biochemical characterizations elucidating the negative impact of H3Q5his (vs. H3Q5ser) on H3K4 HMT activities and antagonism of WDR5 binding to the H3 tail, which highlights the potential importance of histone monoamination dynamics to the biochemical regulation of transcription (Figure 2; Supplementary Figures 5-6, 13/uncropped blots); 3) evidence that histone H3 monoamination modifications display dynamic, rhythmic patterns of enrichment across ZT (which can be disrupted by sleep-inducing drugs, such as Zolpidem) in TMN that are related to alterations in WDR5 binding, and anti-correlate with H3K4me2/3 at Clock-mediated genes (Figure 3; Supplementary Figures 7-10, 14/uncropped blots); and 4) in vivo demonstration that attenuation of H3 monoaminylations in TMN causally suppresses rhythmic gene expression patterns and results in disrupted circadian locomotor behavior (Figure 4; Supplementary Figure 15/uncropped blots).

Reviewers' comments:

Reviewer #1 (Remarks to the Author):

In this manuscript titled "histone monoamination dynamics are regulated by a single enzyme and promote neural rhythmicity", Zheng and the colleagues reported the "eraser" function of a chromatin regulatory enzyme TGM2, which is critical for epigenetics states regulation in neural cells. The findings are somewhat interesting, however, in vivo NGS analysis could further strengthen the main conclusion of this manuscript and the knockin rescue is more desired in the in vivo model for neural rhythmicity. I have included a few shortcomings of the paper below that the authors should address as main controls for the study.

We thank the Reviewer for their positive and insightful comments on our manuscript. In accordance with their suggestions, we now include ample new NGS data (both in cellulo and in vivo/brain), knockout/add-back studies (in cellulo) and biochemical studies (in vitro and in cellulo) to strengthen the manuscript's conclusions. Please find our point-by-point responses to the critiques raised below.

In Figure 1B, besides the two different small molecule inhibitors, the catalytic-dead TGM2 should be transfected in HEK293 cells as negative control. Similarly, a catalytic-dead knock-in should be conducted in HeLa cells, which express endogenous TGM2, to fully demonstrate the catalytic activity of TGM2 is necessary for H3 serotonylation.

*This is an excellent suggestion. To address this concern, we generated a TGM2 CRISPR knockout HeLa cell line (TGM2 KO validation is presented in **Supplementary Figure 1a**) and used this line (vs. WT cells) to demonstrate that: a) histone monoamination deposition is dependent on TGM2; and b) TGM2's catalytic activity is required for its "writer" functions (**Supplementary Figure 1b**), as elucidated through add-back/rescue experiments.*

In Figure 2, the cross-talks between different histone modifications are critical for their physiological functions. The quantification of the modifications with histone H3 N-terminal peptide looks convincing, however, I suggest the authors to quantify these different modifications in living cells by mass spectrometry analysis.

*While we full heartedly agree that it would be highly beneficial to monitor the stoichiometries of these monoamination modifications in cells, or even in brain (vs. in vitro), despite our best efforts, we are currently only able to quantify the H3Q5his in cells via LC-MS/MS (**Supplementary Figure 4e**). We are able to see the H3Q5ser/H3K4me3Q5ser (Farrelly et al., 2019, Nature) and H3Q5dop/H3K4me3Q5dop (manuscript in preparation) in cells/brain tissues via LC-MS/MS following antibody-enrichment or biorthogonal-labeling, however these enrichment strategies do not allow for accurate stoichiometry assessments. While we continue to work with experts in the field of histone mass spectrometry (e.g., Ben Garcia, Simone Sidoli, Jim Galligan, and others) to develop new analytical approaches that might allow for the quantification of all known histone monoamination marks simultaneously within a given sample, these do not currently exist. As such, we feel that the development of new analytical methodologies is well beyond the scope of the current manuscript, and we posit that not having these comparisons at this time does not affect our overall conclusions regarding roles for histone monoamination dynamics in the regulation of circadian plasticity.*

In Figure 4, genome-wide ChIP-seq studies of H3Q5his should be conducted with cells/tissues to further demonstrate the function of this specific histone modification involved in transcriptional regulation. In addition,

TGM2 ChIP-seq is also critical for this model, and should also be included. This is a critical piece of information to link the direct role of H3Q5his to CLOCK gene expression as well.

*We strongly agree with the Reviewer's suggestions and have now added significant new NGS data to the revised manuscript to fully address these concerns. As discussed above (see **List of New Data**), during the revision process, we generated new lots of both our H3Q5his and H3K4me3Q5his antibodies (which took nearly one year to generate and validate; note that our previous antibody lots were found to be not compatible with sequencing-based analyses) that are now amenable for CUT&RUN-seq. First, in **Supplementary Figure panels 7j-l**, we provide data showing that both antibodies work efficiently for CUT&RUN-seq in HeLa cells (where monoamines are provided through serum in the media; note that the H3Q5his antibody did not work for CUT&RUN-seq in brain tissues, and as such, we used the H3K4me3Q5his antibody for in vivo assessments; *vide infra*), that both the single and combinatorial histaminyl marks display similar patterns of enrichment genome-wide (>85% of the peaks for both marks were found to be located at genic loci, similar to that of our previous results for H3Q5ser and H3K4me3Q5ser; Farrelly et al., 2019, Nature & Lukasak et al., 2022, PNAS, Sardar et al., 2023, Science and Chan et al., 2024, JMB), and that such enrichment is indeed TGM2-dependent.*

*Next, using our newly validated antibodies, we performed CUT&RUN-seq for H3K4me3Q5his H3K4me3Q5his (where ~84% of the peaks were found to be located within gene promoters, exons and introns), H3K4me3Q5ser, H3K4me2, H3K4me3 and WDR5 (vs. IgG; E. coli DNA spike-in controls were included in all of these analyses and used for precise normalizations) in TMN tissues across Zeitgeber Time, ZT (ZT0, ZT4, ZT8, ZT12, ZT16, ZT20; **Figure 3d-i, k-l**; **Supplementary Figure 9d-l**; **Supplementary Figure 10a-d**). These data demonstrated that H3K4me3Q5his displays rhythmic patterns of enrichment (consistent with global levels of the marks, as assessed via western blotting; **Supplementary Figure 8a**), which anti-correlate with H3K4 methylation patterns genome-wide. These data are fully consistent with our updated biochemical analyses (now presented in **Figure 2**) indicating that H3Q5his antagonizes WDR5 binding and MLL1-4/SETD1A-B complex activities in vitro, whereas H3Q5ser is permissive for WDR5 binding and HMT activities. Rhythmic fluctuations in H3K4me3Q5his and H3K4me3Q5ser (along with associated changes in WDR5 binding) also correlate with rhythmic patterns of gene expression that appear to be dependent on the transcription factor complex, Clock/Bmal1, based upon ontology analyses. Our findings are in line with earlier studies that found that the WDR5-containing MLL1 complex binds to both CLOCK and BMAL1, the master regulators of circadian transcription, in 293T cells. Furthermore, MLL1 have been previously shown to be recruited to CLOCK/BMAL1 target genes in a rhythmic fashion (Katada and Sassone-Corsi, 2010, PMID: 21113167). In addition, previous genome-wide ChIP-seq analyses in mouse tissues (liver) across circadian time (Koike et al., 2012, PMID: 229365566) provided robust evidence indicating that while co-activator recruitment of chromatin regulators by Clock-Bmal1 precedes nascent transcription, there is a clear lag in enrichment of H3K4 methylation states (likely driven by MLL). This was shown to be relative to RNA PolII recruitment, suggesting that such changes may reflect processes involved in the consequences of transcription vs. transcriptional events themselves. While precise mechanistic roles for H3K4me2/3 in circadian transcriptional regulation remain to be fully elucidated (and are well beyond the scope of the current manuscript), our observations similarly indicate that H3K4 methylation states lag behind that of WDR5 (and presumably MLL/SETD1) recruitment. This recruitment to Clock/Bmal1 target genes in mouse TMN appears, at least in part, to be dependent on the H3 monoaminylation dynamics described above.*

*Additionally, we performed CUT&RUN-seq assessments of H3K4me3Q5his, H3K4me3Q5ser and WDR5 in mouse TMN tissues following treatments with Zolpidem, a sleep-inducing drug, which was administered during the mouse's active phase. The results indicated that Zolpidem alters genome-wide enrichment of H3K4me3Q5his, H3K4me3Q5ser and WDR5 and promotes patterns that phenocopy those observed during phases of inactivity, including at known circadian genes (**Figure 3k-l**).*

*Finally, RNA-seq analysis of mouse TMN tissues across ZT transduced with AAV-H3.3Q5A (which reduces H3Q5his levels and disrupts circadian locomotor behavior; **Figure 4b, e**) vs. AAV-GFP/AAV-H3.3 WT expressing controls, demonstrated that direct attenuation of H3 monoaminylation levels in TMN results in a robust loss of rhythmic, Clock-associated gene expression (**Figure 4c-d**).*

Unfortunately, despite numerous attempts, we were never able to purchase/generate suitable TGM2 antibodies that are amenable for ChIP-seq/CUT&RUN-seq, either in cells or tissues. To our knowledge, TGM2

ChIP-seq/CUT&RUN-seq datasets have not yet been published by any other groups. As such, data to this effect are not presented in our resubmission. While we plan to work diligently in future efforts to generate novel tools/reagents that might allow for these types of sequencing analyses in vivo (e.g., generation of new TGM2 antibodies or mouse lines expressing tagged forms of Tgm2), we feel strongly that such efforts are better suited for follow-up studies. However, of note, we do now provide convincing evidence to indicate that TGM2 is indeed the bona fide monoaminylase in cells (**Supplementary Figure 1a-b; Supplementary Figure 7j-l**), so we do not feel that the lack of such data negatively affect our overall conclusions of our manuscript.

In Figure 5, the authors try to understand the impact of H3Q5his on H3K4 methylation and H3K4 methyltransferases in cells. However, all the experiments and analysis were conducted in vitro. The authors need to determine MLL1-4/SETD1A-B occupancy and the H3K4me1-3 modifications by ChIP-seq in cells with/without TGM2 or treated with/without TGM2 inhibitors.

First, to include more comprehensive assessments of the roles that H3Q5his plays in the regulation/antagonism of MLL (and possibly SETD1) HMT complexes, we began by expanding our initial in vitro experiments to now include MLL1-4 and SETD1A/B in our analyses (**Figure 2d; Supplementary Figure 6c-h**). We performed histone peptide HMT activity assays, followed by LC-MS/MS quantification for all six complexes. Our results indicated that H3Q5his (vs. H3 unmodified) inhibits the activity of all H3K4 HMT complexes tested (note that H3Q5ser does not have such inhibitory effects on binding or activity, as demonstrated with MLL1; **Supplementary Figure 6a**, as well as Farrelly et al., 2019, Nature, Zhao et al. 2021, PNAS & Zhao et al., 2021, Science Advances).

Next, we utilized peptide binding assays to demonstrate that H3Q5his selectively antagonizes the binding of WDR5 (the only member of the MLL/SETD1 complexes to directly interact with the H3 tail in monomeric form; **Supplementary Figure 6i**) to the H3 tail, whereas H3Q5ser potentiates such interactions, as previously reported. It is noteworthy that both monomeric full-length recombinant WDR5 and full-length recombinant WDR5 bound to the MLL1 complex are inhibited in their binding to the H3 tail by H3Q5his, both in the presence and absence of adjacent H3K4me3 (**Figure 2e-f; Supplementary Figure 6j**). The mechanisms controlling this inhibition were further elucidated through structural, biophysical and cellular assessments (**Figure 2g-l**), which indicated that WDR5 interactions with H3Q5his are disfavored owing to electrostatic clash between WDR5 and positively charged Q5his, an antagonistic relationship that mediates WDR5 chromatin binding and H3Q5his-target gene expression.

H3Q5his' antagonism of MLL/SETD1 complex activities appears to occur through disrupted interactions with the WDR5 core subunit and possibly also through unfavorable electrostatics within the core catalytic domain of MLL when in complex with other subunits (**Figure 2b, g-l**). We thus performed CUT&RUN-seq for WDR5 in vivo (TMN brain tissues) across ZT to compare WDR5's enrichment patterns vs. that of H3K4me3Q5his, H3K4me3Q5ser, H3K4me2 and H3K4me3. Our results indicated that H3K4me3Q5his displays rhythmic patterns of enrichment genome-wide (consistent with global levels of the marks, as assessed via western blotting; **Supplementary Figure S8a-b**) that anti-correlate with H3K4 methylation patterns. These data are fully consistent with our updated biochemical analyses indicating that H3Q5his antagonizes WDR5 binding and MLL1-4/SETD1A-B complex activities in vitro, whereas H3Q5ser is permissive for WDR5 binding and HMT activities. Note that based upon our sequencing results, at times when both H3K4me3Q5ser and H3K4me3Q5his are enriched at rhythmic genes (e.g., ZT16), WDR5 is also enriched at those same loci. However, at other times when H3K4me3Q5his is enriched at rhythmic genes and H3K4me3Q5ser is lost at those same loci (e.g., ZT20), WDR5 enrichment was found to be negatively impacted. Together, these data suggest that the presence of H3Q5his in the absence of H3Q5ser is perhaps sufficient to antagonize WDR5 enrichment at Clock/Bmal1-associated genes. Furthermore, our sequencing analyses of WDR5 robustly implicated WDR5 as an important mediator of Clock/Bmal1-mediated rhythmic gene expression, which is the first evidence, to our knowledge, of such a role for WDR5 in brain.

Finally, with respect to the Reviewer's suggested experiment of exploring MLL/SETD1 complex members' enrichment genome-wide in cells in the presence or absence of TGM2 (or with TGM2 inhibitors onboard), we respectfully do not feel that such an experiment would yield conclusive results. Both H3Q5his and H3Q5ser are present in cells at baseline, and knockout of TGM2 would be expected to eliminate all histone monoaminylations (both permissive H3Q5ser and antagonistic H3Q5his with respect to WDR5 binding). This, in turn, will prevent the loss of either mark owing to TGM2's dual role as an "eraser" of H3 monoaminylations. As such, it is unclear what one might expect to happen with respect to WDR5 chromatin interactions. If both

marks are lost by TGM2 knockout, then we would effectively be eliminating both a positive influencer of WDR5 binding (H3Q5ser) and a negative influencer (H3Q5his) of WDR5 interactions. Alternatively, if we were to inhibit TGM2 in cells that already have both marks expressed, we would effectively be preventing the removal of both positive and negative influencers of WDR5 recruitment. As such, we feel that this experiment will likely not further our understanding of roles for these different H3 monoamination marks in cells. We posit that the CUT&RUN-seq analyses of WDR5 in TMN more robustly support roles for WDR5 in the regulation of circadian gene expression and its associations with rhythmic patterns of H3K4me3Q5ser and H3K4me3Q5his.

Finally, the impact of H3Q5his on histone H3K4 demethylases should also be considered or discussed in the manuscript.

This is a certainly interesting point raised by the reviewer. While we feel that the investigation of H3Q5his' impact on recruitment and activities of H3K4 demethylases are worth exploring, such experiments are beyond the scope of the current study and are better suited for follow-up investigations. However, in accordance with the Reviewer's suggestion, we have now added additional discussion regarding this point, highlighting the need for future studies to untangle such phenomena further.

Referee #2 (Remarks to the Author):

In this manuscript, Zheng and colleagues explore the molecular mechanism by which the transglutaminase TGM2 catalyzes the exchange of posttranslational modifications (PTMs, derived from monoamine neurotransmitters) at H3Q5, report a novel PTM – histaminylation, and convincingly demonstrate that this new PTM has a circadian expression pattern and contributes to neural rhythmicity. The authors further show that H3Q5his inhibits binding of the MLL1 complex to histone H3 and affects the methyltransferase activity of this complex.

Overall, this is a very well executed study – it addresses an important biological question and will likely have a profound impact on neuronal epigenetics due to the discovery of (1) the first enzyme that functions as a 'PTM exchanger', expanding the fundamental set of established epi- players, such as PTM-writers, PTM-erasers, PTM-readers, (2) a new PTM, histaminylation, which is directly implicated in neuronal signaling, and (3) diurnal rhythmicity of H3Q5his, the only PTM known to have this capability. This study should be of interest to a wide scientific community, and I enthusiastically support publication.

We very much appreciate the Reviewer's enthusiasm for our manuscript. We have significantly restructured the manuscript in accordance with the Reviewer's suggestions, along with providing ample new data to strengthen the major conclusions of this work.

A few comments:

1. Page 6, first two sentences describing the mechanism are unclear.

Better (or something like this): ... whether this mechanism involves a nucleophilic attack of Cys277 from/of TGM2 on the carbon atom of the g-carboxamide group of Q5 from/of histone H3 (Fig. 2A). In this mechanism [I wouldn't say 'hypothesized'- as it's somewhat predictable- you have two nucleophiles- monoamines, similar in structure around the nucleophile center (i.e. all of them are primary amines), and the equilibrium will be shifted toward the formation of the amide product derived from the monoamine that is present in higher local concentration] the second nucleophilic attack of different/another monoamine neurotransmitter on the thioester intermediate results in the formation of

We appreciate the Reviewer's suggestion for describing this mechanism in a manner that is clearer to the reader. The text describing this mechanism has now been revised accordingly.

2. Throughout the text, 'the re-writing activity' doesn't sound as exciting as it should, I would simply use 'the catalytic activity' [of a PTM/monoamine exchanger?]

This is an excellent suggestion. The descriptions of this activity have been updated in accordance with the Reviewer's suggestion.

3. Page 8, unclear phrase: ... is provided as a donor...

This phrase has been revised for clarity.

4. Page 8, unclear sentence starting with “Our results not only...”. To simplify, Our results show that TGM2 catalyzes transglutamination at H3Q5, leading to the replacement/substitution/switching of one PTM with/to another (or something similar).

This sentence has been revised for clarity.

5. Page 9, the phrase ‘...H3Q5his was efficiently converted to the alternative...’ needs to be revised, as well as the next sentence, ‘...histone monoaminylation is a family..... and based upon...’.

This phrase has been revised as suggested.

6. Overall, the data shown in Figs. 1-3 are novel, high quality, and exciting, but I would display them in one figure, combining Figs. 1a, 1b, 2a, 2c and 3a,b, and move other panels to a supplement. I would also condense the text describing these results. I feel the paper would benefit by shortening some explanations- it’s a superbly interesting part, which needs to be presented in a concise, straight to the point way to be fully appreciated.

7. The data shown in Fig. 4 on the other hand, should be displayed in two separate figures- it’s a very important part of this work.

(6-7): In accordance with the Reviewer’s recommendations, we now present the data as follows – 1) demonstration that TGM2 is both the “writer” and “eraser/exchanger” of histone H3 monoaminylations, including H3Q5his, which displays relative enrichment in vivo in TMN and is presented in this manuscript for the first time (Figure 1; Supplementary Figures 1-4, 12/uncropped blots); 2) biochemical characterizations elucidating the negative impact of H3Q5his (vs. H3Q5ser) on H3K4 HMT activities and antagonism of WDR5 binding to the H3 tail, which highlights the potential importance of histone monoaminylation dynamics to the biochemical regulation of transcription (Figure 2; Supplementary Figures 5-6, 13/uncropped blots); 3) evidence that histone H3 monoaminylation modifications display dynamic, rhythmic patterns of enrichment across ZT (which can be disrupted by sleep-inducing drugs, such as Zolpidem) in TMN that are related to alterations in WDR5 binding, and anti-correlate with H3K4me2/3 at Clock-mediated genes (Figure 3; Supplementary Figures 7-S10, 14/uncropped blots); and 4) in vivo demonstration that attenuation of H3 monoaminylations in TMN robustly suppresses rhythmic gene expression patterns and results in disrupted circadian locomotor behavior (Figure 4; Supplementary Figure 15/uncropped blots).

8. Page 10, the sentence starting with “Such pattern...” could be clearer, say ‘...decarboxylase, which catalyzes the formation of histamine from histidine...’

This sentence has been revised for clarity.

9. Page 11, incorrect phrase: ‘other valence states of H3K4 methylation’

This phrase has been revised accordingly.

10. Page 11, the top paragraph of this page describing K4me dynamics is too long. The authors wanted to simply say that we tested me1/2/3 and found that while ... do not, me2 does. I understand the motivation to write a long text to link it to the following study re MLL activity, but I feel it is unnecessary here.

This paragraph has been revised for brevity.

11. Instead, I would probably add a possible explanation re differences found in TMN vs SCN- this could be exciting and important to future studies.

Possibilities for these differences are now discussed in the Main Text.

12. The data shown in Fig. 5 stand alone, and this part reads more like a somewhat distraction rather than help to the first two parts of the manuscript. There are several issues with including these results. Of note, the data

quality is excellent, I am not questioning it, I am troubled re cohesion with this part, feels more like an artificial addition.

*Please see our response to comments #6-7 above. Given the manner in which the manuscript has been restructured, along with our new data exploring relationships between H3 monoamination enrichment genome-wide vs. WDR5 and H3K4me2/3, we feel that the data flow much better. We now link rhythmic alterations in H3Q5his vs. H3Q5ser to changes in WDR5/H3K4 methylation and rhythmic gene expression, data which are consistent with the biochemical analyses now presented in **Figure 2**.*

13. One issue is that would the structure of WDR5 in complex with H3Q5his show how the his moiety is bound- it would have added the significance to this study (though it's still quite a long link to the first two parts of the manuscript). But modeling never looks overly attractive, especially if the study reports a breakthrough discovery.

*We fully agree that it would have been ideal to present a structure of the H3Q5his peptide bound to WDR5_{WD40}, which more definitively resolves the Q5his moiety's position within the complex. However, despite our best attempts, we were never able to generate a complex structure that fully traces the electron density surrounding the histamination mark itself. Given that Q5 (which is partially traced in our complex) points outwards from the K4 binding pocket toward the surface of WDR5_{WD40}, its position appears to be flexible, which may contribute to the difficulties in tracing the histamination mark. In addition, since our biochemical data robustly indicate that the positive charge of histamine within this complex functions to antagonize WDR5_{WD40} binding to the H3 tail, such charge repulsion (which we posit occurs because of interactions with K259 on WDR5) may also contribute to this flexibility and a subsequent lack of Q5his tracing within the complex. We continue to feel that the structural modeling data presented in **Figures 2h-I** are critical to our manuscript, providing testable hypotheses/mechanisms for further exploring binding relationships between H3Q5his and WDR5, as well as the impact of such interactions on WDR5 chromatin binding and related gene expression. With respect to our previous lack of cohesiveness linking the biochemical results from **Figure 2** to the remainder of the data, we now feel that the new data presented in our revised manuscript – including the manner in which we have restructured the discussion of this work – have greatly improved its general flow to allow for more robust and cohesive conclusions.*

*First, to include more comprehensive assessments of the roles that H3Q5his plays in the regulation/antagonism of MLL (and possibly SETD1) HMT complexes, we began by expanding our initial in vitro experiments to include assessments of MLL1-4 and SETD1A/B (**Figure 2d; Supplementary Figure 6c-h**). Specifically, we performed histone peptide HMT complex activity assays, followed by LC-MS/MS quantifications for all six complexes. These analyses demonstrated that H3Q5his (vs. H3 unmodified) inhibits the activity of all H3K4 HMT complexes tested (note that H3Q5ser does not have such inhibitory effects on binding or HMT complex activities, as demonstrated with MLL1; **Supplementary Figure 6a**, as well as Farrelly et al., 2019, Nature, Zhao et al. 2021, PNAS & Zhao et al., 2021, Science Advances).*

*Next, we utilized peptide binding assays to demonstrate that H3Q5his selectively antagonizes the binding of WDR5 (the only member of the MLL/SETD1 complexes to directly interact with the H3 tail in monomeric form; **Supplementary Figure 6i**) to the N-terminal H3 tail, whereas H3Q5ser potentiates such interactions, as previously reported. It is noteworthy that both monomeric full-length recombinant WDR5 and full-length recombinant WDR5 bound to the MLL1 complex are inhibited in their binding to the H3 tail by H3Q5his, both in the presence and absence of adjacent H3K4me3; **Figure 2e-f; Supplementary Figure 6j**. The mechanisms controlling this inhibition were further elucidated through structural, biophysical and cellular assessments (**Figure 2g-I**), which indicated that WDR5 interactions with H3Q5his are not favored owing to electrostatics between WDR5 and positively charged Q5his, an antagonistic relationship that mediates WDR5 chromatin binding and H3Q5his-target gene expression.*

*H3Q5his' antagonism of MLL/SETD1 complex activities appears to occur through disrupted interactions with the WDR5 core subunit and possibly also through unfavorable electrostatics within the core catalytic domain of MLL when in complex with other subunits (**Figure 2b, g-I**). We thus performed CUT&RUN-seq for WDR5 in vivo (TMN brain tissues) across ZT to compare its enrichment patterns vs. that of H3K4me3Q5his, H3K4me3Q5ser, H3K4me2 and H3K4me3. Our results indicated that H3K4me3Q5his displays rhythmic patterns of enrichment genome-wide (consistent with global levels of the marks, as assessed via western blotting; **Supplementary Figure 8a**) that anti-correlate with H3K4 methylation patterns. These data are fully*

consistent with our updated biochemical analyses indicating that H3Q5his antagonizes WDR5 binding and MLL1-4/SETD1A-B complex activities *in vitro*, whereas H3Q5ser is permissive for WDR5 binding and HMT activities. Note that based upon our sequencing results, at times when both H3K4me3Q5ser and H3K4me3Q5his are enriched at rhythmic genes (e.g., ZT16), WDR5 is also enriched at those same loci. However, at other times when H3K4me3Q5his is enriched at rhythmic genes and H3K4me3Q5ser is lost at those same loci (e.g., ZT20), WDR5 enrichment was found to be negatively impacted. Together, these data suggest that the presence of H3Q5his in the absence of H3Q5ser is perhaps sufficient to antagonize WDR5 enrichment at Clock/Bmal1-associated genes. Furthermore, our CUT&RUN sequencing analyses of WDR5 robustly implicated WDR5 as an important mediator of Clock/Bmal1-mediated rhythmic gene expression, which is the first evidence, to our knowledge, of such a role for WDR5 in brain.

Our findings are in line with earlier studies that found that the WDR5-containing MLL1 complex binds to both CLOCK and BMAL1, the master regulators of circadian transcription, in 293T cells. Furthermore, MLL1 has previously been shown to be recruited to CLOCK/BMAL1 target genes in a rhythmic fashion (Katada and Sassone-Corsi, 2010, *Nat Struct Mol Biol*, PMID: 21113167). In addition, previous genome-wide ChIP-seq-based analyses in mouse tissues (liver) across circadian time (Koike et al., 2012, *Science*, PMID: 229365566) provided robust evidence indicating that while co-activator recruitment of chromatin regulators by Clock-Bmal1 precedes nascent transcription, there is a clear lag in enrichment of H3K4 methylation states (likely driven by MLL). This was shown to be relative to RNA PolII recruitment, suggesting that such changes may reflect processes involved in the consequences of transcription vs. transcriptional events themselves. While precise mechanistic roles for H3K4me2/3 in circadian transcriptional regulation remain to be fully elucidated (and are well beyond the scope of the current manuscript), our observations similarly indicate that H3K4 methylation states lag behind that of WDR5 (and presumably MLL/SETD1) recruitment. This recruitment to Clock/Bmal1 target genes in mouse TMN appears, at least in part, to be dependent on the H3 monoaminylation dynamics described above.

14. The second issue is that the MLL1 (human COMPASS) complex is known to catalyze tri-methylation to a higher extent than it catalyzes di-methylation, therefore focusing on K4me2 and not showing K4me3 in Fig. 5 raises more questions than answers.

It is certainly true *in vivo*, as the Reviewer points out, that the MLL1 (COMPASS) complex catalyzes H3K4me3 to a higher extent than H3K4me2. However, in both our MALDI-TOF (**Figure 2c**) and LC-MS/MS (**Figure 2d; Supplementary Figure 6a, c-h**) experiments using peptide substrates, we found that the purified MLL1 complex largely catalyzes mono- and dimethylated H3. In the MALDI-TOF experiment, we did not observe catalysis of H3K4me3 at all, and in the LC-MS/MS experiments, while we did observe some H3K4me3, the signal was <1% of the total peptides being measured (**Supplementary Figure 6c-h** provide quantifications for all methylation states observed on H3K4). It is important to note that our results are consistent with previous reports, which demonstrated that MLL1's catalysis of H3K4me3 specifically requires a nucleosomal context, with H3K4me3 unable to be catalyzed even in the context of monomeric H3 by MLL1. H3K4me1/2 can be efficiently catalyzed by MLL1 to H3 monomers, similar to our observations using peptides (Park et al., *Nature Communications*, 2019: PMID 31804488). Importantly, however, we do find that H3Q5his significantly attenuates the processive activities of the MLL1 complex to establish H3K4me1 and H3K4me2, which would be required to fully establish H3K4me3 *in vivo* on nucleosomes. As such, while future experiments perhaps using synthetic 'designer' H3Q5his-modified nucleosomes could certainly be useful in validating our peptide results, the effort required to generate and validate such reagents would, in our opinion, unnecessarily delay publication. As such, we continue to feel that the peptide data presented in our revised manuscript are of physiological relevance and nicely demonstrate the inhibitory activities of H3Q5his on the establishment of H3K4 methylation states.

15. Third, the rationale to go after MLL1 is somewhat weak, why not to test other writers/readers/erasers specific for methylated K4? There is no evidence that other proteins do not play a role or greatly affect the behavior of Q5his.

This is an excellent suggestion. ITC assessments of binding affinities between monoaminylated peptides and known interactors of histone H3 (e.g., WDR5, CHD1, TAF3, BPTF, JMJD2A and JARID1A) are now included in our revised manuscript (**Supplementary Figure 6a-f**). Comparing the affinities of these effector proteins to unmodified vs. H3(K4me3)Q5his vs. H3(K4me3)Q5ser peptides revealed that WDR5 is selectively antagonized in its binding to the H3 tail by H3Q5his compared to unmodified H3 or H3Q5ser. While we cannot conclude, of

course, that H3Q5his does not alter the binding of other H3 N-terminal tail interacting proteins that were not tested here, we feel confident – owing to these data, as well as the other validations presented throughout the paper (in vitro, in cellulo and in vivo) – that H3Q5his indeed antagonizes WDR5 binding and MLL/SETD1 complex activities, whereas H3Q5ser potentiate such binding interactions. In addition, we have now restructured the manuscript in a manner that better links our biochemical findings in **Figure 2** to the in vivo regulation of WDR5 recruitment to circadian genes (**Figure 3**), especially given that WDR5/MLL activities have been well-documented to play important roles in the regulation of circadian transcriptional plasticity. Furthermore, it is clear from our sequencing results in **Figure 3** that circadian fluctuations in H3K4me3Q5ser/H3K4me3Q5his may support mechanisms in addition to that of regulating WDR5 recruitment to chromatin, and as such, we now discuss this possibility in the text.

16. Fig. 5d, the rationale to switch to MLL3 is unclear and weak, as the catalytic activity and genome localization of MLL3 and MLL1 differ considerably.

*We thank the reviewer for pointing this out. Since the MLL1 complex structure with SAH was recently published [Sha et al., Nat Cell Biology 2023], we now model possible H3Q5his interactions with MLL1 using this structure, as well as the existing MLL3-RBBP5-ASH2L-H3 complex structure. Specifically, the MLL1-SAH structure was aligned to the MLL3-SAH-histone peptide structure, where we found that SAH and other conserved residues aligned very well. Based upon these structural modeling assessments (**Figure 2b**), we now predict that the H3Q5 residue would be ‘sandwiched’ between R3886 and K3945 of MLL1. While enough space would exist, in theory, to tolerate a large modification such as serotonylation (with stacking between R3886 and serotonylation possibly stabilizing the H3 peptide), the positively charged histaminylation moiety would not be favored owing to electrostatic clash. We now used these new modeling predictions to test whether H3Q5his (vs. H3 unmodified, or H3Q5ser in the context of MLL1) impacts HMT activities for all of the H3K4 HMT complexes (MLL1-4, SETD1A/B). Indeed, we found that H3Q5his attenuates the activities of all complexes tested (**Figure 2c-d**; **Supplementary Figure 6a**). Based upon these data, as well as experiments now presented in **Figure 2e-i** and **Supplementary Figure 6j**, we hypothesize that H3Q5his has two independent effects of MLL complex activity: 1) inhibit the activity of the catalytic center of MLL due to unfavorable electrostatics; and 2) affect the recruitment of MLL complexes to chromatin in a WDR5-dependent fashion.*

17. Fig 5i-k, the results shown are even further away from the first two parts of the manuscript than MLL1.

Please see our responses to comments #6-7 and #13 above.

18. Page 14, unclear phrase: ‘displayed abrogated binding’, better: abrogates or decreases binding.

This phrase has been revised for clarity.

19. Page 15, unclear phrase: ‘WDR5WD40-H3 complex in the presence of an H3Q5his peptide.’

This phrase has been revised for clarity.

20. Page 15, the phrase ‘...to act as writer, eraser and re-writer...’ could be simplified to ‘to act as a PTM exchanger’ (or similar).

This phrase has been simplified.

Referee #3 (Remarks to the Author):

Manuscript by Zheng et al provides new evidence that TGM2 can utilize histamine produced in histaminergic neurons to introduce or erase histone H3Q5His modifications. The authors detected this modification in the hypothalamic tuberomammillary nucleus (TMN) in mouse brain in vivo and demonstrate that the modification exhibits circadian rhythmic pattern. Overexpression of mutant histone H3 that cannot be modified on Q5 results in an increase in H3K4me2, some deregulation of circadian gene expression, and a mild change in circadian motor behavior.

This is an interesting and novel extension of previous work pioneered by this group on monoamine H3Q5 modifications. The strongest point of the manuscript is the rigorous and convincing biochemical characterization of TGM2 dependent histamination of H3Q5. Where the study is less convincing, is in the analysis of H3Q5His functional role in the regulation of rhythmic gene expression in the TMN nucleus and its consequences for animal behavior (see major concerns below). Most importantly, the study does not offer any clarity on more fundamental mechanistic questions – what is the chromatin-wide distribution of histamine modifications in neurons, how does it change during the circadian cycle, does it correlate/anti-correlate with any of the other H3 modifications, and how do the modifications affect expression of associated genes.

We greatly appreciate the Reviewer's positive and helpful comments on our manuscript. In accordance with the Reviewer's insightful suggestions, we now include substantial new data and analyses (including aforementioned sequencing analyses of histaminylation, both in cellulo and in vivo/brain across ZT), which we believe have greatly strengthened the conclusions that can be drawn from this work. Please find our point-by-point responses to the provided critiques below.

Major concerns:

1. Punch collection of TMN nucleus from 1mm thick sections can result in variable ratios of histaminergic and non-histaminergic neurons per specimen. Therefore, it is necessary to normalize all western blots to the fraction of histaminergic neurons in each sample. This should be performed by comparing markers specific to histaminergic neurons and pan neuronal markers in protein lysates.

*This is an interesting point that is raised by the Reviewer. While grossly informative, we feel that western blotting for the various marks in TMN served only to provide a global view of the overall abundance of these modifications at various time points across ZT but are not necessarily valuable in dissecting their molecular roles in the potential regulation of circadian gene expression. With all the new circadian sequencing data now presented in our revised manuscript (see our response to comment #2 below), we have thus chosen to de-emphasize western blotting analyses of the marks across ZT (originally presented as primary evidence for circadian regulation of the histaminyl marks in our first submission) in this resubmission. Instead, we now only show (via western blotting) circadian regulation of H3K4me3Q5his and H3Q5his in TMN (both of which are significantly rhythmic via ANOVA), along with H3Q5his in SCN (not found to be significantly rhythmic via ANOVA – **Supplementary Figure 8a, c**). Together, we make the overarching point that since these initial data indicated possible rhythmic fluctuations of the H3 histaminylation marks, we next chose to further explore their circadian fluctuations genome-wide (via CUT&RUN-seq) and their potential relationships with circadian gene expression (via RNA-seq).*

*To further address the Reviewer's concern, we performed additional western blotting to probe for Tgm2 (**Supplementary Figure 8b**), Hdc and NeuN across ZT in mouse TMN (note that SCN tissues were used as dissection controls, as Hdc was only found to be expressed within the TMN; **Supplementary Figure 9a**). These results robustly demonstrated the consistency and accuracy of our dissections, as well as the fact that neither Tgm2, Hdc nor NeuN levels are altered as a consequence of circadian time. While we continue to feel that total H3 is the most appropriate loading control for modified H3, such as H3K4me3Q5his and H3Q5his, we are comforted by the observation that neither Hdc nor NeuN levels display variance in expression across ZT. We also performed ELISAs for both histamine and serotonin in TMN across ZT and found that neither monoamine displays circadian fluctuations in terms of their total levels. One caveat of this analysis, however, is that such assessments cannot provide accurate measurements of intracellular, or subcellular, pools of the monoamines under study, which may indeed display intra-nuclear rhythmic fluctuations, as might be predicted based on the biochemical data presented in **Figure 1**. We cannot definitively state at this time why the monoaminyl marks display circadian regulation in TMN (e.g., alterations in monoamine biosynthesis/release, alterations in TGM2 activity, regulation of TGM2 function by PTMs that have yet to be identified, etc.). However, it is now clear that they are indeed rhythmic at the genome-wide level, that their enrichment patterns anti-correlate with H3K4 methylation (and correlate with regulation of WDR5 recruitment) and that their fluctuations appear to occur at rhythmic genic loci that are known Clock/Bmal1 targets. While beyond the scope of the current study, future experiments will indeed be necessary to further elucidate the mechanisms through which alterations in histone monoaminylation signatures are regulated. The need for such future studies is now discussed in the text.*

2. What are the levels of H3Q5Ser in the TMN nucleus over the circadian cycle? Are Ser modifications on H3 exchanged for His modifications and vice versa or non-modified histones are subjected to the circadian changes in His modifications?

*This is an outstanding question. during the revision process, we generated new lots of both our H3Q5his and H3K4me3Q5his antibodies (which took nearly one year to generate and validate; note that our previous antibody lots were found to be not compatible with sequencing-based analyses) that are now amenable for CUT&RUN-seq. First, in **Supplementary Figure** panels **7j-l**, we provide data showing that both antibodies work efficiently for CUT&RUN-seq in HeLa cells (where monoamines are provided through serum in the media; note that the H3Q5his antibody did not work for CUT&RUN-seq in brain tissues, and as such, we used the H3K4me3Q5his antibody for in vivo assessments; *vide infra*), that both the single and combinatorial histaminyl marks display similar patterns of enrichment genome-wide (>85% of the peaks for both marks were found to be located at genic loci, similar to that of our previous results for H3Q5ser and H3K4me3Q5ser; Farrelly et al., 2019, Nature & Lukasak et al., 2022, PNAS, Sardar et al., 2023, Science and Chan et al., 2024, JMB), and that such enrichment is indeed TGM2-dependent.*

*Next, using our newly validated antibodies, we performed CUT&RUN-seq for H3K4me3Q5his H3K4me3Q5his (where ~84% of the peaks were found to be located within gene promoters, exons and introns), H3K4me3Q5ser, H3K4me2, H3K4me3 and WDR5 (vs. IgG; E. coli DNA spike-in controls were included in all of these analyses and used for precise normalizations) in TMN tissues across Zeitgeber Time, ZT (ZT0, ZT4, ZT8, ZT12, ZT16, ZT20; **Figure 3d-i, k-l**; **Supplementary Figure 9d-l**; **Supplementary Figure 10a-d**). These data demonstrated that H3K4me3Q5his displays rhythmic patterns of enrichment (consistent with global levels of the marks, as assessed via western blotting; **Supplementary Figure 8a**), which anti-correlate with H3K4 methylation patterns genome-wide. These data are fully consistent with our updated biochemical analyses (now presented in **Figure 2**) indicating that H3Q5his antagonizes WDR5 binding and MLL1-4/SETD1A-B complex activities in vitro, whereas H3Q5ser is permissive for WDR5 binding and HMT activities. Rhythmic fluctuations in H3K4me3Q5his and H3K4me3Q5ser (along with associated changes in WDR5 binding) also correlate with rhythmic patterns of gene expression that appear to be dependent on the transcription factor complex, Clock/Bmal1, based upon ontology analyses. Our findings are in line with earlier studies that found that the WDR5-containing MLL1 complex binds to both CLOCK and BMAL1, the master regulators of circadian transcription, in 293T cells. Furthermore, MLL1 have been previously shown to be recruited to CLOCK/BMAL1 target genes in a rhythmic fashion (Katada and Sassone-Corsi, 2010, PMID: 21113167). In addition, previous genome-wide ChIP-seq analyses in mouse tissues (liver) across circadian time (Koike et al., 2012, PMID: 229365566) provided robust evidence indicating that while co-activator recruitment of chromatin regulators by Clock-Bmal1 precedes nascent transcription, there is a clear lag in enrichment of H3K4 methylation states (likely driven by MLL). This was shown to be relative to RNA PolII recruitment, suggesting that such changes may reflect processes involved in the consequences of transcription vs. transcriptional events themselves. While precise mechanistic roles for H3K4me2/3 in circadian transcriptional regulation remain to be fully elucidated (and are well beyond the scope of the current manuscript), our observations similarly indicate that H3K4 methylation states lag behind that of WDR5 (and presumably MLL/SETD1) recruitment. This recruitment to Clock/Bmal1 target genes in mouse TMN appears, at least in part, to be dependent on the H3 monoaminylation dynamics described above.*

*Next, we performed CUT&RUN-seq assessments of H3K4me3Q5his, H3K4me3Q5ser and WDR5 in mouse TMN tissues following treatments with Zolpidem, a sleep-inducing drug, which was administered during the mouse's active phase. The results indicated that Zolpidem alters genome-wide enrichment of H3K4me3Q5his, H3K4me3Q5ser and WDR5 and promotes patterns that phenocopy those observed during phases of inactivity, including at known circadian genes (**Figure 3k-l**).*

*Finally, RNA-seq analysis of mouse TMN tissues across ZT transduced with AAV-H3.3Q5A (which reduces H3Q5his levels and disrupts circadian locomotor behavior; **Figure 4b, e**) vs. AAV-GFP/AAV-H3.3 WT expressing controls, demonstrated that direct attenuation of H3 monoaminylation levels in TMN results in a robust loss of rhythmic, Clock-associated gene expression (**Figure 4c-d**).*

3. It is not clear why were the analyses in Fig S7B interpreted as a proof that these modifications and proteins are non-rhythmic? Just showing that the data do not fit well with 3rd polynomial function (the authors should list the p-values for their analyses in this figure and Fig 4D) cannot be used as a proof, as other oscillatory

patterns can be present in the data. Indeed, it appears that there are statistically significant differences across individual timepoints. The authors should perform ANOVA analysis to evaluate changes in gene expression across the time series.

*In accordance with the Reviewer's suggestion, we have now performed one-way ANOVAs for H3K4me3Q5his and H3Q5his in TMN (see comments above regarding the removal of additional histone PTM western blot analyses in this resubmission), as well as for H3Q5his in SCN, across ZT, which revealed significant regulation of both H3K4me3Q5his and H3Q5his in TMN, but not for H3Q5his in SCN (**Supplementary Figure 8a,c**).*

*In addition, our new CUT&RUN-seq data, as discussed above, as well as previously presented TMN RNA-seq data (**Figure 3a-b**), robustly demonstrate rhythmic patterns of enrichment for all of the marks (as well as WDR5) examined across ZT. These alterations display robust overlaps with gene expression patterns determined to be circadian (note that for these genomic analyses, JTKcycle – a non-parametric algorithm for detecting rhythmic components of genome-scale datasets – was used employed to statistically call circadian events).*

4. The same applies to the analysis of H3Q5His in SCN. What does ANOVA statistical analysis of this dataset show? Is the increase at 16h significant compared to 4h? Does this suggest delayed accumulation of histamine modification outside of the TMN? The authors should extend the analysis of H3Q5His to other brain regions that are diffusely innervated by histaminergic neurons to bring more clarity how broadly the modifications might affect brain function.

*As mentioned above, we have now performed one-way ANOVAs for H3K4me3Q5his and H3Q5his in TMN, as well as for H3Q5his in SCN, across ZT, which revealed significant regulation of both H3K4me3Q5his and H3Q5his in TMN, but not for H3Q5his in SCN. While we agree with the Reviewer that it could be interesting to explore additional brain regions that are more diffusely innervated by histaminergic neurons for regulation of H3Q5his across circadian time, we would argue that such studies are well beyond the scope of the current manuscript and would be better suited for follow-up studies. We feel that these assessments would be best explored using sequencing-based approaches vs. western blotting to determine potential circadian regulation of the marks (note that while H3Q5his is indeed enriched in TMN, it is also expressed in other brain regions, just to a lower extent – see **Supplementary Figure 7m**). We address this issue in the text commenting about the benefit of such assessments to be made in future studies.*

5. Oscillation of H3Q5His in Fig 4D seems to be more complex than a typical circadian gene with one maximum and one minimum over the period. There appear to be two maxima (0/24h and 8-12h) and two minima (4h and 20h). Is it possible that technical differences in sample collection across the 24 h period could contribute to the reported observations (another reason to include quantification of histaminergic and generic neuronal markers in lysates)? If the pattern is real, how do the authors interpret this strange rhythmicity and how does it correlate with histamine levels in the brain?

*Based on our CUT&RUN-seq data for H3K4me3Q5his in TMN (and in comparison to H3K4me3Q5ser and H3K4me2/3 levels.), it indeed appears that the histaminyl mark displays two maxima at the genome-wide level – one at ZT16/20 (active phase) and another at ZT8 (inactive phase), although at a significantly lower level vs. ZT16. Within the active phase, we see that H3K4me3Q5his initially co-enriches at circadian loci with H3K4me3Q5ser (although likely on different nucleosomes) and WDR5 (likely on the same nucleosomes based upon our biochemical data). But as the active phase proceeds to ZT20, we find that H3K4me3Q5ser enrichment falls off, with a corresponding loss of WDR5, while H3K4me3Q5his remains preserved in its enrichment at this time point. This may suggest that the combination of losing H3(K4me3)Q5ser (which is permissive to WDR5 binding) and maintenance of H3(K4me3)Q5his (which is antagonistic to WDR5 binding) contributes to the overall loss of WDR5 binding at circadian, Clock/Bmal1-associated targets. Following ZT20, and into the animal's inactive phase, when H3K4me3Q5his enrichment is reduced, we observed a corresponding increase in H3K4 methylation, which again is consistent with our biochemical data demonstrating that H3Q5his is antagonistic to HMT complex activities. Why H3K4me3Q5his would experience another maxima at ZT8 (towards the end of the inactive phase) remains unclear at this time, although perhaps such a large surge in H3Q5his enrichment simply functions to prevent subsequent increases in WDR5/H3K4 methylation enrichment during the transition back into the animal's active phase. While future studies will indeed be necessary to fully tease apart these complex patterns of regulation for H3Q5his across circadian time, these data – along with our biochemical findings in **Figure 2** and H3.3Q5A manipulation analyses in*

Figure 4 – robustly indicate that transitions in H3 monoaminylation states contribute to the regulation of circadian gene expression and behavior.

Moreover, as discussed above, we performed additional western blot analyses of Hdc and NeuN across ZT in TMN (note that SCN was used as a dissection control, as HDC was only found to be expressed within the TMN; **Supplementary Figure 9a**). These results robustly demonstrated the consistency and accuracy of our dissections, as well as the fact that neither Hdc nor NeuN levels are altered as a consequence of circadian time. While we continue to feel that total H3 is the most appropriate loading control for modified H3, such as H3K4me3Q5his and H3Q5his, we are comforted by the observation that neither Hdc nor NeuN levels display variance across the circadian time. In addition, we performed ELISAs for both histamine and serotonin in TMN across ZT and found that neither monoamine displays circadian fluctuations in terms of their total levels (**Supplementary Figure 9b-c**). One caveat of this analysis, however, is that such assessments cannot provide accurate measurements of intracellular pools of the monoamines under study, which may indeed display intranuclear rhythmic fluctuations, as might be predicted based on the biochemical data presented in **Figure 1**. As discussed above, we cannot determine at this time why the monoaminyl marks display circadian regulation in TMN (e.g., alterations in monoamine biosynthesis/release, alterations in TGM2 activity, etc.). However, it is clear that they are indeed rhythmic, that their levels anti-correlate with overall H3K4 methylation patterns (and correlate with regulation of WDR5 recruitment) and that their fluctuations appear to occur at rhythmic genic loci that are known Clock/Bmal1 targets.

6. The authors should take advantage of their antibodies to purify H3Q5His and H3Q5Ser from TMN and quantify the amount of K4me1/2/3 modifications associated with these two modifications. Such analysis would provide important support to the hypothesis that the two modifications lead to a differential recruitment of the MLL1 complex.

Despite our best efforts, owing to the limited amount of tissues that can be recovered from mouse TMN (even with pooling), we were only able to use these tissues for RNA-seq and/or CUT&RUN-seq, which require very low input, but not for biochemical assays. Our ideal experiment would include monitoring the stoichiometries of the different monoaminylations, in the presence or absence of H3K4 methylation events, in cells, or even in brain. However, the only mark that we are currently able to observe without enrichment is H3Q5his in cells overexpressing TGM2 and treated with histamine (**Supplementary Figure 4e**). We are able to detect the H3Q5ser/H3K4me3Q5ser (Farrelly et al., 2019, Nature) and H3Q5dop/H3K4me3Q5dop (manuscript in preparation) modifications in cells/brain tissues via LC-MS/MS using antibody-enrichment or biorthogonal-labeling, however, such enrichment assays unfortunately do not allow for accurate stoichiometry assessments. While we continue to work with experts in the field of histone mass spectrometry (e.g., Ben Garcia, Simone Sidoli, Jim Galligan, and others) to develop new analytical approaches that might allow for the quantification of all known histone monoaminylation marks simultaneously within a given sample, these do not currently exist. As such, we feel that the development of new analytical methodologies is well beyond the scope of the current manuscript, and we posit that not having these comparisons at this time does not affect our overall conclusions regarding roles for histone monoaminylation dynamics in the regulation of circadian plasticity.

7. What are the anticipated concentrations of WDR5 in the nucleus? Micromolar Kd values seem high, raising a question whether the proposed WD40 based binding mechanism is physiological. Please address and discuss.

To our knowledge, the absolute concentrations of WDR5 protein in the nucleus of cultured cells have not been determined and certainly have not been convincingly elucidated in brain tissues. However, WDR5 is indeed a highly expressed gene in brain, as demonstrated through our RNA-seq assessments. We however disagree with the Reviewer's assertion that the micromolar Kd values obtained for WDR5_{WD40} binding to the H3 tail are high in comparison to other known histone-"reader" interactions and may not be of physiological relevance (see **Supplementary Figure 5** for additional ITC assessments of other known and well validated H3 N-terminal tail binding proteins). WDR5 is well established to be the primary MLL/SETD1 core complex member that initially presents the H3 tail to the rest of the HMT complex to promote the catalytic addition of H3K4 methylation, and its binding affinity is in the range of other known histone binding proteins. These binding events (e.g., bromodomain protein interactions with histone acetylation marks) are now being targeted in the clinic as cancer treatments). In addition, our new sequencing data are fully consistent with our updated biochemical assessments (presented in **Figure 2**) indicating that H3Q5his antagonizes WDR5 binding and MLL1-4/SETD1A-B complex HMT activities in vitro, whereas H3Q5ser is permissive for WDR5 binding and

HMT activities. For example, rhythmic fluctuations in, and divergent patterns of, H3K4me3Q5his and H3K4me3Q5ser (along with associated changes in WDR5 binding) correlate with rhythmic patterns of gene expression that appear to be dependent on the transcription factor complex, Clock/Bmal1, based upon ontology analyses. Please also see our response to comment #5 above for additional discussion regarding potential roles for histone monoamination dynamics and WDR5 recruitment to neural chromatin in the regulation of circadian gene expression.

8. Effects of viral overexpression of mutant H3Q5A should be compared against the overexpression of WT H3, control non-transduced tissue should be separated out of the analysis. Looking at the data presented in figure 4H, I do not believe that there is any statistical difference in H3Q5His/H3 between the red and white dots. This is an important point, as overexpression of H3 will affect stoichiometry of H3 and its modifiers. Thus, the authors should quantify other H3 modifications following WT and mutant H3 expression (e.g. H3K4me1, H3K4me3, H3K27Ac).

We thank the Reviewer for bringing this to our attention. The indicated experiments have now been repeated with separated GFP and H3.3 WT controls vs. H3.3Q5A transduced animals (Figure 4b). In doing so, we now observe that H3.3Q5A transduced TMN displays statistically significantly lower levels of H3Q5his vs. both control groups (normalized to total H3 levels). However, with respect to our viral approach potentially altering the stoichiometry of histone H3 within neural chromatin, this is not predicted to be the case. For example, we do not observe alterations in the expression of total H3 levels with our viral manipulations, nor do we see global differences in other H3 modifications (e.g., H3K4me3 and H3K27ac; Supplementary Figure 10e), as requested by the Reviewer. Importantly, H3.3 is the only isoform of H3 expressed in neural chromatin in adult brain and the total abundance of H3 in chromatin cannot change in response to alterations in H3 expression. In fact, even knockdown of H3.3 in neurons results in a reduction in histone turnover rates to preserve overall chromatin structure and total H3 levels. Thus, overexpression of histone H3.3 (WT or mutant forms) will not change the total amount of histone H3 expressed in neural chromatin, but rather will alter the rates of histone H3 turnover to allow for competition against endogenous histones for incorporation (see Maze et al., Neuron, 2015). Therefore, our strategy using H3.3Q5A functions effectively as a dominant negative. Also, our viral approach has a distinct advantage for the circadian experiments, as presented in Figure 4. Specifically, it is non-selective for the different histone monoamination marks, thereby allowing for simultaneous reductions in both H3Q5his (Figure 4b), which is antagonistic to H3K4 methylation, and H3Q5ser (e.g., Al-Kachak et al., Nature Communications, 2024; Sardar et al. Science, 2023; Farrelly et al. Nature, 2019), which is permissive for H3K4 methylation, within a given tissue. Since we have now restructured our manuscript to better explore relationships between H3Q5his vs. H3Q5ser dynamics in the regulation of neural rhythmicity, we feel that it is advantageous to be able to disrupt both modification simultaneously in order to explore the impact of such manipulations on circadian gene expression and behavior.

While our data clearly indicate that attenuating H3 monoaminylations in TMN severely disrupts circadian gene expression and locomotor behavior, it is unclear at this time what impact such manipulations might have on adjacent K4 methylation states at the genome-wide level. The most appropriate experiment to examine this would be to perform ChIP/re-ChIP-seq assays, first IP'ing for exogenously expressed (and tagged) H3.3 WT vs. H3.3Q5A, followed by re-ChIP for the monoaminyl modifications or other marks, such as H3K4me3, to show that – in a locus-specific manner – incorporation of H3.3Q5A significantly alters enrichment for H3K4 methylations. Unfortunately, while we have used such approaches in recent studies (using primary neuronal culture systems, for example) to definitively demonstrate that H3.3Q5A incorporation into neuronal chromatin reduces the monoaminyl marks at affected loci (e.g., H3K4me3Q5ser; Al-Kachak et al., Nature Communications, 2024, PMID: 38871707), such strategies require ample input material, the likes of which can simply not be obtained from virally transduced in vivo brain samples. As such, while we fully agree that future studies aimed at exploring potential crosstalk interactions between H3Q5his/ser and other PTMs could be useful, we feel that such studies are well beyond the scope of the current manuscript and will require detailed follow-up investigations.

9. The authors mention that expression of H3Q5A resulted in “significant overlaps with genes disrupted by H3Q5his manipulations in comparisons to those identified as being rhythmic”. This should be supported by more detailed analysis and inclusion of the data. Are these genes increased or decreased? Are they deregulated in a specific phase of the cycle? More specifically, the authors should comment on the level of histidine decarboxylase expression; and whether they anticipate attenuated or enhancer histamine neurotransmission in the manipulated animals.

We very much appreciate the Reviewer's suggestion to further expand upon our previous viral manipulation gene expression analyses in this resubmission. To do so, additional RNA-seq data were collected from mouse TMN tissues across ZT. Briefly, mouse TMN were transduced with AAV-H3.3Q5A, which reduces H3Q5his levels and disrupts circadian locomotor behavior (**Figure 4b, e**), vs. AAV-GFP/AAV-H3.3 WT expressing controls. These data robustly demonstrated that direct attenuation of H3 monoamination levels in TMN results in a robust loss (~92 % of all rhythmic genes) of circadian, Clock/Bmal1-associated gene expression. Subsequent ontology analyses revealed, as expected, enrichment of dysregulated genes in pathways associated with circadian entrainment, as well as enrichment in pathways associated with neurotrophin signaling, LTP and glutamatergic synapses (**Figure 4d**). Regulation of histaminergic signaling, per se, was not observed among the enriched pathways, however, it is unclear – given the limited numbers of genomics datasets currently available from TMN – whether such ontology pathways are well established. Interestingly, however, the gene encoding Hdc – which is important for histamine biosynthesis – was found to be circadian in both our non-virally transduced RNA-seq dataset (**Figure 3a**), as well as in the merged GFP/H3.3WT dataset (**Figure 4c**). Note that the data from the two control groups were merged in these analyses, as direct comparisons between GFP and H3.3 WT-expressing animals did not reveal significant gene expression differences. We now provide additional discussion in the text regarding the possibility that our H3.3Q5A manipulation functions, at least in part, to disrupt histamine biosynthesis/histaminergic signaling, which may, in turn, feedback to disrupt circadian gene expression and behavior.

10. Peptide co-IP experiments (Fig 5F-H) are interpreted as: “H3Q5his displayed abrogated binding” to MLL1 complex. This is misleading as the histamine modification potentiated the interaction compared to unmodified H3 peptide. Importantly, the experiments need to be performed with K4me1,2,3 modified peptides and double modified peptides as well.

We very much thank the Reviewer for pointing this out. Upon a re-analysis of our peptides, we realized that they were of slightly different concentrations in solution, which led to the appearance of a slightly potentiated binding interaction between H3Q5his and WDR5. After adjusting our peptides' concentration for affinity pulldown experiments, we performed the following experiments to address the Reviewer's concerns:

Figure 2e: Peptide affinity pulldown assay demonstrating that H3Q5his antagonizes the binding of recombinant WDR5 (the only member of the MLL/SETD1 complexes to directly interact with the H3 tail in monomeric form; **Supplementary Figure 6i** – related to **Figure 2**) to the N-terminal H3 tail, whereas H3Q5ser potentiates this interaction (all now appropriately normalized to peptide concentrations).

Figure 2f: Peptide affinity pulldown assay demonstrating that H3K4me3Q5his antagonizes the binding of recombinant WDR5 to the N-terminal H3 tail in comparison to H3K4me3 and H3K4me3Q5ser (all now appropriately normalized to peptide concentrations).

Supplementary Figure 6j (related to **Figure 2**): Peptide affinity pulldown assay demonstrating that H3Q5his antagonizes the binding of recombinant WDR5 within the MLL1 complex to the N-terminal H3 tail, whereas H3Q5ser potentiates this interaction (all now appropriately normalized to peptide concentrations).

Regarding the additional studies mentioned by the Reviewer aimed at exploring the potential impact of H3K4me1/2 on H3Q5his antagonism of WDR5 binding (or H3Q5ser's potentiation of binding), we respectfully do not feel that such experiments are necessary at this time to support the overall conclusions drawn from this work. WDR5 has been shown previously to bind equally well to all valence states of H3K4 (unmodified vs. K4me1 vs. K4me2 vs. K4me3; Ruthenburg et al., Nat Struct Mol Biol, 2006), and the effort to generate these additional peptides might unnecessarily delay publication. As such, we hope that the addition of the data discussed above will be sufficient to address the Reviewer's concerns.

11. Despite the biochemical analysis, it is impossible to conclude which histones are histamine modified in cells (random, at promoters, within genes, in distal enhancers, insulators or other elements?) and how does the modification correlate with other H3 modifications. Without such analysis, it is difficult to arrive to a satisfactory interpretation of the data and a plausible model. The authors should include ChIP-seq (or CUT&RUN) analysis of H3 modifications in the TMN at 8h and 20h and compare these with patterns of H3K4me2 and H3K4me3. If this is technically not possible, the chromatin analysis combined with gene

expression analysis should be performed in the HEK/TGM2 cell system, as it will help to establish how broad and strong effect do histamine modifications play in the control of gene expression.

*As discussed above, using our newly validated antibodies, we performed CUT&RUN-seq for H3K4me3Q5his, H3K4me3Q5ser, H3K4me2, H3K4me3 and WDR5 (vs. IgG; E. coli DNA spike-in controls were included in all of these analyses and used for precise normalizations) in TMN tissues across Zeitgeber Time, ZT (ZT0, ZT4, ZT8, ZT12, ZT16, ZT20; **Figure 3d-i, k-l; Supplementary Figure 9d-l; Supplementary Figure 10a-d**). These data demonstrated that H3K4me3Q5his displays rhythmic patterns of enrichment (consistent with global levels of the marks, as assessed via western blotting; **Supplementary Figure 8a**), which anti-correlate with H3K4 methylation patterns genome-wide. These data are fully consistent with our updated biochemical analyses (now presented in **Figure 2**) indicating that H3Q5his antagonizes WDR5 binding and MLL1-4/SETD1A-B complex activities in vitro, whereas H3Q5ser is permissive for WDR5 binding and HMT activities. Rhythmic fluctuations in H3K4me3Q5his and H3K4me3Q5ser (along with associated changes in WDR5 binding) also correlate with rhythmic patterns of gene expression that appear to be dependent on the transcription factor complex, Clock/Bmal1, based upon ontology analyses. Our findings are in line with earlier studies that found that the WDR5-containing MLL1 complex binds to both CLOCK and BMAL1, the master regulators of circadian transcription, in 293T cells. Furthermore, MLL1 have been previously shown to be recruited to CLOCK/BMAL1 target genes in a rhythmic fashion (Katada and Sassone-Corsi, 2010, PMID: 21113167). In addition, previous genome-wide ChIP-seq analyses in mouse tissues (liver) across circadian time (Koike et al., 2012, PMID: 229365566) provided robust evidence indicating that while co-activator recruitment of chromatin regulators by Clock-Bmal1 precedes nascent transcription, there is a clear lag in enrichment of H3K4 methylation states (likely driven by MLL). This was shown to be relative to RNA PolII recruitment, suggesting that such changes may reflect processes involved in the consequences of transcription vs. transcriptional events themselves. While precise mechanistic roles for H3K4me2/3 in circadian transcriptional regulation remain to be fully elucidated (and are well beyond the scope of the current manuscript), our observations similarly indicate that H3K4 methylation states lag behind that of WDR5 (and presumably MLL/SETD1) recruitment. This recruitment to Clock/Bmal1 target genes in mouse TMN appears, at least in part, to be dependent on the H3 monoaminylation dynamics described above.*

*Further CUT&RUN-seq assessments of H3K4me3Q5his, H3K4me3Q5ser and WDR5 in mouse TMN tissues following treatments with Zolpidem, a sleep-inducing drug administrated during the animal's active phase, indicated that Zolpidem alters genome-wide enrichment patterns of H3K4me3Q5his, H3K4me3Q5ser and WDR5 to promote patterns that phenocopies those observed during phases of inactivity, including at known circadian genes (**Figure 3k-l**).*

*Finally, RNA-seq analysis of mouse TMN tissues across ZT transduced with AAV-H3.3Q5A (which reduces H3Q5his levels and disrupts circadian locomotor behavior; **Figure 4b, e**) vs. AAV-GFP/AAV-H3.3 WT expressing controls, demonstrated that direct attenuation of H3 monoaminylation levels in TMN results in a robust loss of rhythmic, Clock-associated gene expression (**Figure 4c-d**).*

With the addition of these new sequencing data, along with our robust new biochemical findings, we feel confident in our conclusions that histone H3 monoaminylation dynamics play critical roles in guiding rhythmic gene expression in TMN, in part, through mediation of WDR5 recruitment and H3K4 methylation activities. There is, of course, much still to be learned about these fascinating modifications and their mechanistic actions in the regulation of neural plasticity. However, we sincerely hope that our efforts, as presented in this resubmission, are considered by the Reviewer to be sufficient for publication of our current manuscript, as we feel that they offer important advances to the fields of chromatin biology and circadian gene regulation in brain.

Reviewers' comments:

Reviewer #1 (Remarks to the Author):

Never received, however, an additional Reviewer polled by the Editor confirmed that all previous critiques by Reviewer 1 were addressed in our previous submission.

Reviewer #2 (Remarks to the Author):

The authors have substantially revised the manuscript and went beyond and above to address the reviewers' comments and further strengthen this remarkable study.

We thank the Reviewer again for their enthusiastic appreciation of our study and continued helpful comments. Please see our responses to the remaining points raised below.

A few remaining points:

1. Modeling of the structures in the revised version remains an issue that needs to be resolved by editing the text and revising Fig. 2. The authors have added a modeled PTM moiety on the crystal structure of the MLL1 SET domain (Fig. 2b), which I would discourage to do. Because the major scientific breakthrough reported here is the discovery and characterization of the new PTM, H3Q5his, any overstatement, especially regarding Q5his, would only damage the incredible work of this team and is unnecessary here. I would certainly avoid showing [published by others structure of MLL1] and manually manipulating it by adding a PTM (Fig. 2b).

As suggested by the Reviewer, we have removed previous Figure 2b from the manuscript, along with related discussions in the text regarding these modeling efforts, in order to not overstate potential roles for H3Q5his in the antagonism of H3K4 methylation. Instead, we focus entirely on our experimentally obtained data to support potential mechanisms contributing to such phenomena.

2. Fig. 2g shows a top-quality [as many other impressive structures reported by the Haitao's group] structure of the complex, and I don't see a problem that the electron density of the his moiety is not observed. I would keep this panel as it is but add the label "WDR5-H3Q5his complex" or "WDR5-H3Q5his" below the image to emphasize the presence of Q5his. This will be sufficient and clear - reporting experimentally derived data.

We have added the label "WDR5-H3Q5his complex" to Figure 2f (previously Figure 2g).

3. Please do not manually add the His moiety in Fig. 2h and 2i, it's sure would be perceived as a misleading representation by the scientific community. [I would remove Fig. 2i entirely]

In accordance with the Reviewer's suggestion, we have removed the manually added His moiety from Figure 2g (previous Figure 2h) and have also removed previous Figure 2i entirely.

4. I would suggest toning down the explanation as to how the his moiety is recognized. Is this moiety in fact positively charged here? what is the pKa value of it? if the average pKa of a His (residue, the side chain) is 6.0 in proteins, would it be positively charged at pH 7.4? or what percent will be positively charged? and it is shown as a neutral group in Fig. 1 of this manuscript.

Histamine contains two basic centers – the aliphatic amino group and whichever nitrogen atom of the imidazole ring does not already have a proton. Under physiological conditions, histamine is normally protonated to a singly charged cation, with the aliphatic amino group (having a pKa of ~9.4) being protonated, whereas the second nitrogen of the imidazole ring (pKa ~5.8) is not protonated. Since in the absence of pathological states, the pH of the human body is slightly basic (ranging from 7.35 to 7.45, with an average of 7.4), the predominant form of histamine is monoprotic at the aliphatic nitrogen. This would also be predicted to be true with in vitro and in cellulo systems that are buffered to physiological pH. Having said this, we have done our best in the revised version of the manuscript to tone down the explanation of how precisely the His moiety is recognized. In addition, we now show the His moiety as monoprotic in Figure 1 for clarification..

5. The contribution of steric hindrance might be a valid explanation too, as the difference between Kds for K259A and K297E is quite small, 7 μ M and 16 μ M, and we cannot exclude favorable intramolecular contact(s) involving the side chain of E in the complex that are unrelated to the presence of the his moiety.

The Reviewer is correct that WDR5-K259E has a larger side chain compared to WDR5-K259A. As such, we now additionally discuss possible contributions of favorable intramolecular contacts involving the side chain of E in the complex, which may also contribute to the increased binding observed with WDR5-K259E vs. WDR5-K259A.

6. If I am not mistaken, comparison of the current structure with the structure of WDR5-H3Q5ser (Zhao, 2021) does not support the idea of the charge-charge repulsion- the binding pocket for ser is essentially neutral or even slightly negatively charged. Does this region of the domain (surface) differ in the current structure?

We thank the Reviewer for pointing this out. They are correct that the binding surface of H3Q5ser in the Zhao, 2021 WDR5-H3Q5ser structure is different from that of H3Q5his in our current structure. To assess this directly, we performed alignments of WDR5-H3Q5un (Schuetz et al., 2006, EMBO J) vs. WDR5-H3Q5ser (Zhao et al., 2021, Science Advances) vs. WDR5-H3Q5his (Figure 2f), which are now presented in Figures 2h-i. While H3A1-H3K4 in all structures could be aligned well, H3Q5un and H3Q5his were found to have the same binding orientation, both of which differ from that of H3Q5ser. As such, it does appear that the His moiety makes H3Q5 more flexible, allowing it to come into contact with WDR5-K259, which would support the idea of charge-charge repulsion (further supported by our mutational ITC assessments in Figure 2j). Having said this, in accordance with the Reviewer's comment #4 above, we have toned down our explanation of how precisely the His moiety is recognized in the revised manuscript, so as to not overstate the possible mechanisms involved.

7. One possibility that might help to elaborate the uniqueness of Q5his is to compare the position/orientation of the Q5 side chain in the current structure and structures of the WDR5-H3unmod and WDR5-H3Q5ser complexes. If the side chain of Q5 is restrained in these two known complexes, then it's logical to say that the presence of the his moiety makes Q5 flexible. But nothing else could be drawn at this point.

Please refer to our response to comment #6 above.

8. To summarize, I would suggest presenting only experimentally obtained structural data in figures and avoiding unneeded speculations at this point. This manuscript has exceedingly large number of amazing results, and following studies could be focused on proper analysis of the his binding site (it's not as solid at this point and is not as essential for the original discovery report).

We fully agree with the Reviewer's additional suggestions and have modified the Figures and text to address all of the concerns raised above.

9. Fig. 2a, should be ASH2L.

We thank the Reviewer for noticing this labeling error (also found in our Model presented in Supplementary Figure 11). This has been corrected in both Figure 2 and Supplementary Figure 11.

Reviewer #3 (Remarks to the Author):

The authors conducted a series of additional experiments and analyses that address the majority of my criticisms. They also explained that some requested experiments are beyond the scope of the current manuscript, while others are technically not feasible. I do not have any further concerns.

We very much thank the Reviewer for their continued positive comments on our manuscript, and their indication that we have adequately addressed all of their prior concerns.